# Targeting P2Y$_{14}$R protects against necroptosis of intestinal epithelial cells through PKA/CREB/RIPK1 axis in ulcerative colitis

Chunxiao Liu[1], Hui Wang[2], Lu Han[1], Yifan Zhu[2], Shurui Ni[1], Jingke Zhi[1], Xiping Yang[1], Jiayi Zhi[1], Tian Sheng[2], Huanqiu Li [2] ✉ & Qinghua Hu [1] ✉

Purinergic signaling plays a causal role in the pathogenesis of inflammatory bowel disease. Among purinoceptors, only P2Y$_{14}$R is positively correlated with inflammatory score in mucosal biopsies of ulcerative colitis patients, nevertheless, the role of P2Y$_{14}$R in ulcerative colitis remains unclear. Here, based on the over-expressions of P2Y$_{14}$R in the intestinal epithelium of mice with experimental colitis, we find that male mice lacking P2Y$_{14}$R in intestinal epithelial cells exhibit less intestinal injury induced by dextran sulfate sodium. Mechanistically, P2Y$_{14}$R deletion limits the transcriptional activity of cAMP-response element binding protein through cAMP/PKA axis, which binds to the promoter of *Ripk1*, inhibiting necroptosis of intestinal epithelial cells. Furthermore, we design a hierarchical strategy combining virtual screening and chemical optimization to develop a P2Y$_{14}$R antagonist **HDL-16**, which exhibits remarkable anti-colitis effects. Summarily, our study elucidates a previously unknown mechanism whereby P2Y$_{14}$R participates in ulcerative colitis, providing a promising therapeutic target for inflammatory bowel disease.

Inflammatory bowel diseases (IBD), classified into Crohn's disease (CD) and ulcerative colitis (UC), are chronic inflammatory intestinal disorders, which involve dynamic alterations in numerous cell types including epithelial, mesenchymal and immune cells[1]. In clinic, non-steroidal anti-inflammatory drugs (NSAIDs), glucocorticosteroids, immunosuppressants, and biological agents such as infliximab and adalimumab have been approved to treat IBD, nevertheless, long-term use of these drugs would lead to a train of side effects including autoimmune responses, viral infection, or tumorigenesis[2,3]. Thus, it remains an urgent need to discover novel efficient targets for IBD treatment as well as corresponding therapeutic strategies.

As the histological hallmark of IBD, disruption of the epithelial barrier during intestinal inflammation results from dysregulated cell death of intestinal epithelium cells (IECs), which might trigger more

IECs damage, leading to a vicious cycle[4]. Necroptosis, a kind of programmed cell death, is driven by receptor-interacting protein kinase 1 (RIPK1) and receptor-interacting protein kinase 3 (RIPK3), which phosphorylates the mixed-lineage kinase domain-like pseudo kinase (MLKL)[5]. Phosphorylated MLKL oligomerizes and translocates to the membrane, forming pores and finally disrupting cellular membrane integrity[6]. Previous studies have found necroptosis was active in children with IBD and contributed to heighten intestinal inflammation[7]. Consistently, IECs erosion and colitis induced with dextran sulfate sodium (DSS) were attenuated by knocking out MLKL, implicating that IECs necroptosis might act as a key constituent of experimental colitis[8]. Especially, necroptosis also actively participated in the inflammatory response by promoting cascade reactions through the leakage of cellular contents from damaged plasma membranes[9]. Based

[1]School of Pharmacy, China Pharmaceutical University, Nanjing 211198, China. [2]College of Pharmaceutical Sciences, Soochow University, Suzhou, China. ✉e-mail: huanqiuli@suda.edu.cn; huqh@cpu.edu.cn

on the above-mentioned findings, suppressing excessive necroptosis could be a promising therapeutic strategy for IBD. However, the role of dysregulated genes that contribute to necroptosis in IBD remains largely unexplored.

A growing body of evidences indicated that purinergic signaling and its receptor system were widely involved in the development of IBD[10]. An epidemiological study on purine gene dysregulation profiles in IBD showed that 59% of purine genes were dysregulation in IBD, but only the expression of *P2ry14* was positively correlated with acute inflammatory score in UC mucosal biopsies[11]. In addition, previous studies have established that P2Y$_{14}$R and its endogenous ligand UDP-Glucose play key roles in many inflammation diseases, including gout, acute kidney injury and hepatic fibrosis[12–14]. Nonetheless, how UDPG/P2Y$_{14}$R affects the progression of IBD still remains unknown and needs further investigation.

Most P2Y$_{14}$R-related studies in vivo have highlighted the pro-inflammatory functions of P2Y$_{14}$R distributed in immune cells[14–16], while P2Y$_{14}$R are also expressed in parenchymal cells including epithelial cells, endothelial cells, and fibroblasts[17,18], which might participate in cell fate determination. Given the importance of IECs homeostasis to IBD pathophysiology, we investigated the role of P2Y$_{14}$R in an animal model of experimental colitis characterized by IECs necroptosis. Furthermore, we examined the effect and mechanism of P2Y$_{14}$R in IECs necroptosis using intestinal organoids or intestinal epithelial cell lines. In addition, we designed and synthesized a potent small-molecule P2Y$_{14}$R antagonist with high activity and low toxicity, which diminished IECs necroptosis and DSS-induced colitis in the challenge phase through targeting P2Y$_{14}$R. These results suggested that inhibition of P2Y$_{14}$R might be an effective strategy to treat UC patients.

## Results

### The P2Y$_{14}$R of intestinal epithelial cells regulates human IBD and DSS-induced experimental colitis

To investigate the potential role of P2Y$_{14}$R in the pathogenesis of IBD, we first analyzed *P2ry14* expression in intestinal biopsies from UC patients from public datasets (using datasets from NCBI's Gene Expression Omnibus: GSE38713, GSE75214, GSE16879). As shown in Fig. 1a, *P2ry14* expression was significantly increased in UC specimens compared with those in healthy controls. To confirm the findings from GEO datasets, we performed immunolocalization experiments on the inflamed colon tissues from UC and CD patients, the non-IBD tissues were used as healthy control. The results showed intense co-expression of P2Y$_{14}$R and EpCAM staining in ileal and colonic specimens from patients with UC, which mainly localized to IECs. In contrast, specimens from patients with CD tissues and healthy control showed subtle expression (Fig. 1b). To validate these data in IBD, we further performed RT-qPCR and immunoblotting analysis to determine P2Y$_{14}$R level of colonic epithelia in DSS-treated mice (Fig. 1c, d). To expand upon these observations, we also executed immunolocalization to characterize the P2Y$_{14}$R expression in IECs of DSS-treated mice. Notably, we found that P2Y$_{14}$R was increased in IECs of the DSS-treated mice relative to that of the control mice (Fig. 1e). Besides, Rybaczyk and her colleagues found that, in UC patients, the P2Y$_{14}$R expression of intestinal mucosa but not of PBMC was positively correlated with acute inflammatory score[11]. Altogether, these results indicated that P2Y$_{14}$R in the IECs might play a critical role in the UC progression.

To further explore the involvement of P2Y$_{14}$R in the intestinal epithelium in greater detail, we generated IEC-specific P2Y$_{14}$R knockout (P2Y$_{14}$R$^{\Delta IEC}$) mice. Then we challenged P2Y$_{14}$R$^{fl/fl}$ and P2Y$_{14}$R$^{\Delta IEC}$ mice with 3% DSS and monitored for daily weight and disease onset (Fig. 1f). We found that IEC-specific P2Y$_{14}$R knockout did not affect the weight, disease activity index (DAI) and colon length of mice received standard drinking water (Fig. 1g, h). However, P2Y$_{14}$R$^{\Delta IEC}$ mice displayed

mitigated weight loss, diarrhea and rectal bleeding compared with P2Y$_{14}$R$^{fl/fl}$ mice after DSS treatment (Fig. 1h). Meanwhile, rectum shortening, increased intestinal permeability, spleen index as well as histologic damage were less in P2Y$_{14}$R$^{\Delta IEC}$ mice than P2Y$_{14}$R$^{fl/fl}$ mice (Fig. 1h–k).

Besides, we performed chronic DSS colitis model in P2Y$_{14}$R$^{\Delta IEC}$ or P2Y$_{14}$R$^{fl/fl}$ mice, (supplementary Fig. 1a). The results showed that IEC-P2Y$_{14}$R deficient also displayed significant improvement in weight loss, diarrhea, rectal bleeding as well as histologic damage in the chronic DSS colitis model especially during the feeding period. However, the improvement effect of rectum shortening in chronic DSS colitis model was not as obvious as the acute DSS model (supplementary Fig. 1 b–d). Additionally, given the functional expression of P2Y$_{14}$R in myeloid cells, especially in macrophages and neutrophils, we also generated myeloid cells-specific P2Y$_{14}$R knockout (P2Y$_{14}$R$^{fl/fl}$ *Lyz2-cre*) mice, and found that myeloid cells-specific P2Y$_{14}$R knockout did not lead to significant improvements in DSS-induced weight loss, diarrhea, rectal bleeding, colon shortening as well as tissue damage (supplementary Fig. 1e–h). Collectively, these results strongly suggested that P2Y$_{14}$R of IECs was involved in the development of IBD.

### The P2Y$_{14}$R of intestinal epithelial cell regulates epithelial necroptosis

As described above, IEC-specific P2Y$_{14}$R knockout conferred protection from DAI evaluation, colon length and histological damage upon challenge with DSS, the improved IBD phenotype of P2Y$_{14}$R$^{\Delta IEC}$ mice led us to investigate mechanistic insight into the protective action of P2Y$_{14}$R knockout in IECs. Notably, the immunofluorescence staining results of tight junction proteins (Claudin-1, Occludin, and ZO-1) showed a marked improvement of the gut barrier in DSS-treated P2Y$_{14}$R$^{\Delta IEC}$ mice (Fig. 2a), indicating that P2Y$_{14}$R mediated DSS-induced colitis maybe through affecting the death of IECs. Interestingly, the deficiency of P2Y$_{14}$R in IECs showed less influence on TUNEL positive cells and the expression of apoptosis-related proteins (Caspase-3, Bcl-2, and Bax) in IECs, suggestive of a non-apoptotic mode of IECs death (Fig. 2b, c). Meanwhile, the level of Caspase-1 p20 and GSDMD N-terminal (GSDMD N) indicated that deficiency of P2Y$_{14}$R in IECs did not influence the pyroptosis of IECs in DSS-induced experiment colitis (Fig. 2b). In contrast, the activation of MLKL was suppressed significantly by IEC-specific P2Y$_{14}$R knockout (Fig. 2b). Consistently with the western blot results, transmission electron microscope (TEM) analysis showed that typical necroptosis cell morphology, including swelling of organelles, translucent cytoplasm, and loss of plasma membrane integrity, was dramatically suppressed by IEC-specific P2Y$_{14}$R knockout (Fig. 2d), indicating that IEC-specific P2Y$_{14}$R knockout alleviated DSS-induced colitis probably through inhibiting the necroptosis of IECs.

To further probe the function of P2Y$_{14}$R in the regulation of IECs necroptosis, we employed an in vitro model by treating HT-29 cells, a human intestinal epithelial cell line, with TNF-α adding Smac mimetic and z-VAD (TSZ) as described previously[19]. To determine if diminishing the level of P2Y$_{14}$R could reduce IECs necroptosis, the cellular P2Y$_{14}$R level was decreased by transfecting HT-29 cells with siRNA. The results showed that P2Y$_{14}$R silence significantly inhibited TSZ-induced necroptosis in HT-29 cells as evidenced by increased cell viability and decreased PI-positive cells, respectively, when compared with cells transfected with control siRNA (Fig. 2e, f). The necroptotic cell death was further confirmed using H&E staining. Notably, we found that cells treated with TSZ displayed an altered morphology typical of necrotic cells with increased size, swelling of cytoplasm, and rupture of the plasma membrane. Whereas, the silence of P2Y$_{14}$R reversed the morphological changes of HT-29 cells (Fig. 2g). Subsequently, we examined the effect of P2Y$_{14}$R silence on necroptosis hallmark, including MLKL and its phosphorylated status. Immunoblotting results showed

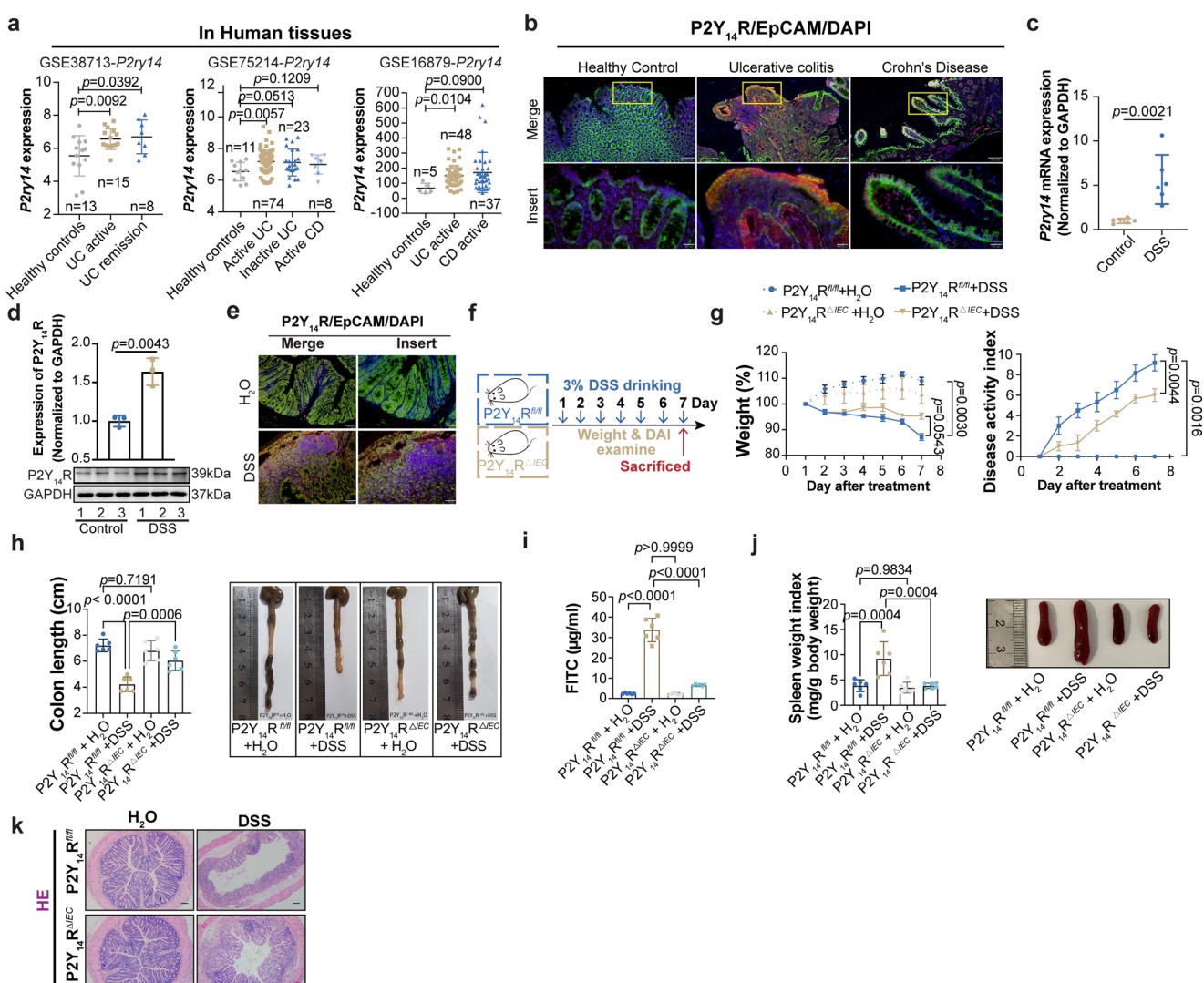

**Fig. 1 | The P2Y₁₄R of intestinal epithelial cell regulates human IBD and DSS-induced experimental colitis. a** *P2ry14* expression in the colonic mucosa of healthy individuals or IBD patients from GEO database (using datasets GSE38713, GSE75214, GSE16879). **b** The immunofluorescent images of colon tissues from healthy control and UC or CD patients stained with P2Y₁₄R (red), EpCAM (green), and nuclear (DAPI, Blue), Scale bar = 100 μm (up), Scale bar = 20 μm (down). **c** RT-qPCR analysis of *P2ry14* mRNA of intestinal epithelial cells in mice (*n* = 6 mice per group). **d** Western blot assay of P2Y₁₄R protein expression of intestinal epithelial cells in mice (*n* = 3 mice per group). **e** Representative colonic samples from control mice and DSS-treated mice were stained with P2Y₁₄R (red), EpCAM (green), and nuclear (DAPI, Blue), Scale bar = 50 μm (left), Scale bar = 20 μm (right).

**f** Experimental flow chart. **g** Body weight change and DAI evaluation during the disease process (*n* = 6 mice per group). **h** The length of colons from P2Y₁₄Rᶠˡ/ᶠˡ and P2Y₁₄Rᐃᴵᴱᶜ mice after DSS treatment (*n* = 6 mice per group). **i** Analysis of serum FITC-dextran assay (*n* = 6 mice per group). **j** The spleen index of P2Y₁₄Rᶠˡ/ᶠˡ and P2Y₁₄Rᐃᴵᴱᶜ mice after DSS treatment (*n* = 6 mice per group). **k** The H&E staining in colon tissues of DSS-treated mice, Scale bar = 200 μm. The data represent the mean ± SD for **a, c-d** and **h-j**, the data represent the mean ± SEM for **g**. The *p*-values were determined by two-tailed Unpaired T test for **a, c** and **d**, two-tailed paired Student's t -test for **g**, One-way analysis of variance (ANOVA) with Tukey multiple comparison test for **h, i, j**. For **b**, **e**, **and k**, each image was acquired independently three times, with similar results. Source data are provided as a Source Data file.

that HT-29 cells treated with TSZ showed an increase in p-MLKL level. While these necroptotic indicators were decreased when transfected with si-P2Y₁₄R (Fig. 2h). Consistently, the silencing of P2Y₁₄R in HCT-116 cells, another human intestinal epithelial cell line, also showed a suppression effect on necroptosis induced by TSZ, as characterized by the decreased PI positive cells and p-MLKL level (Fig. 2i, j). To further confirm the results of HT-29 and HCT-116 cells, we generated intestinal epithelial organoids from P2Y₁₄Rᶠˡ/ᶠˡ or P2Y₁₄Rᐃᴵᴱᶜ mice and then stimulated them with TSZ for 12 hours, and found that P2Y₁₄R deficient significantly inhibited TSZ-induced necroptosis in P2Y₁₄Rᐃᴵᴱᶜ organoids (Fig. 2k). Collectively, with the above observations, the P2Y₁₄R of IECs plays a key role in modulation of IECs necroptotic and DSS-induced colitis.

## P2Y₁₄R regulates necroptosis of IECs via RIPK1/RIPK3 pathway

The binding of RIPK1 and RIPK3 has been reported to initiate the formation of the necrosome complex[20]. Thus, together with MLKL, RIPK1/RIPK3 pathways have also been considered as the core components of the necroptotic programs and necroptosis-associated diseases[21]. More importantly, the co-immunoprecipitation analysis showed that silence of P2Y₁₄R disrupted the interaction between RIPK1 and RIPK3 (supplementary Fig. 2a). Based on that, we hypothesized that P2Y₁₄R regulated necroptosis of IECs via RIPK1/RIPK3 pathway. To probe this hypothesis, we overexpressed P2Y₁₄R by transfecting HT-29 cells with a hP2Y₁₄R-expression vector to enhance TSZ-induced necroptosis. As shown in supplementary Fig. 2b, c, overexpressed P2Y₁₄R aggravated the necroptosis of IECs, as demonstrated by cell viability analysis and

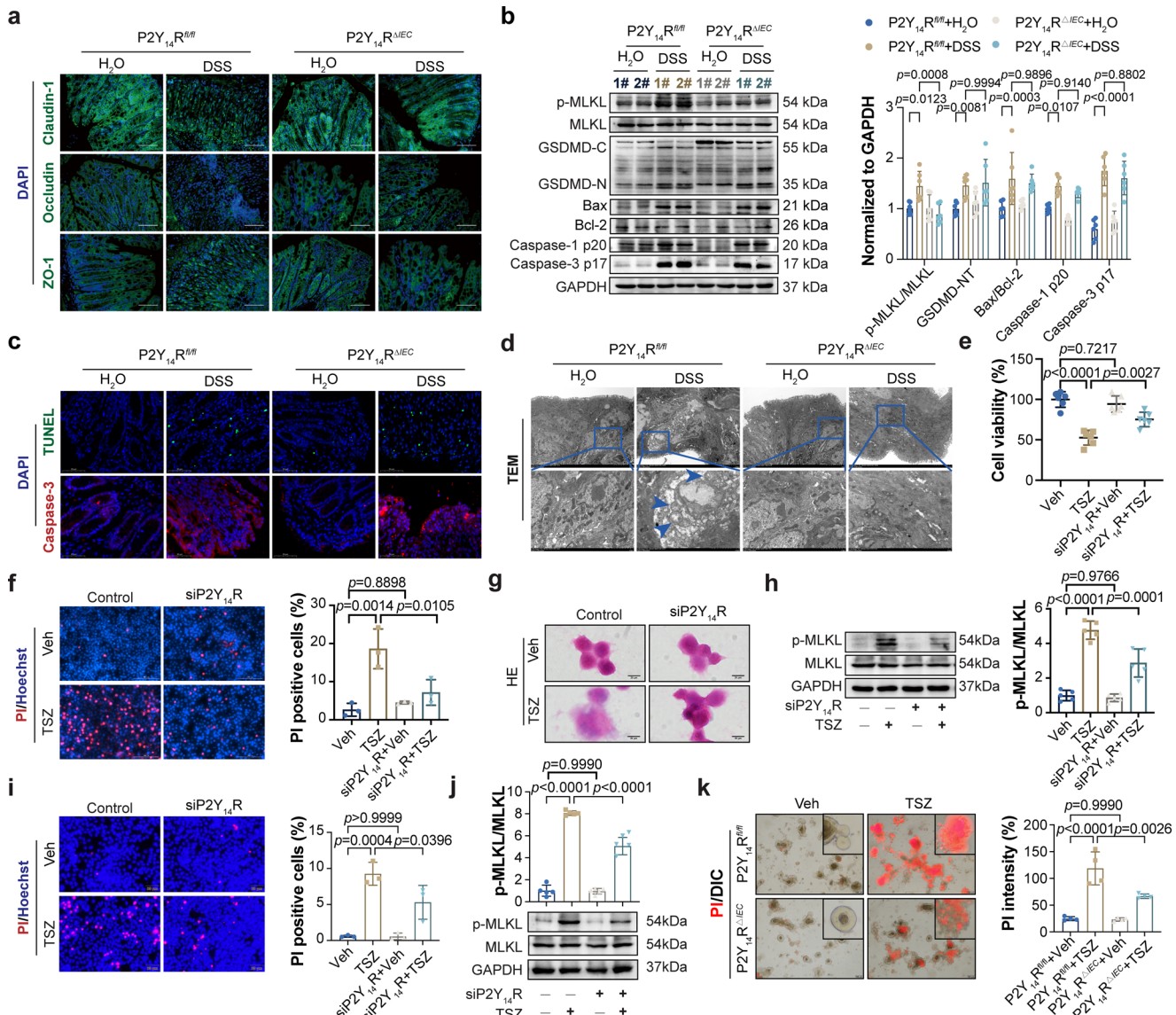

**Fig. 2 | The P2Y$_{14}$R of intestinal epithelial cell regulates epithelial necroptosis.**
**a** The immunofluorescent images of colon tissues stained with Claudin-1, Occludin and ZO-1, the principal components of tight junction (Scale bar = 200 μm). **b** The expression of Bcl-2, Bax, GSDMD-NT, Caspase-1 p20, Caspase-3 p17, MLKL, and phosphorylation of MLKL were analyzed by immunoblotting in the IECs from P2Y$_{14}$R$^{fl/fl}$ and P2Y$_{14}$R$^{\triangle IEC}$ mice after DSS treatment ($n$ = 6 mice per group). **c** The TUNEL staining of colon tissues from P2Y$_{14}$R$^{fl/fl}$ and P2Y$_{14}$R$^{\triangle IEC}$ mice after DSS treatment. The immunofluorescent images of colon tissues stained with cleave caspase-3 (Scale bar = 50 μm). **d** The TEM images showed the typical characteristics of necroptosis in the colon tissues of DSS-treated mice. **e** Cell viability was determined by CCK8 analysis in HT-29 cells ($n$ = 6 samples per group). **f** PI-positive cells were analyzed by PI/Hoechst staining in HT-29 cells ($n$ = 3 samples per group).

**g** H&E staining of HT-29 cells. **h** Phosphorylation of MLKL as well as its protein levels were analyzed by immunoblotting in HT-29 cells ($n$ = 5 samples per group). **i** PI-positive cells were analyzed by PI/Hoechst staining in HCT-116 cells ($n$ = 3 samples per group). **j** Phosphorylation of MLKL as well as its protein levels were analyzed by immunoblotting in HCT-116 cells ($n$ = 5 samples per group). **k** The PI staining and quantification of intestinal organoids from P2Y$_{14}$R$^{fl/fl}$ and P2Y$_{14}$R$^{\triangle IEC}$ mice treated as indicated with DMSO (veh) or TNF-α adding Smac mimetic and z-VAD (TSZ) for 12 h ($n$ = 3 samples per group). The data represent the mean ± SD, and $p$-values were determined by One-way ANOVA with Tukey multiple comparison test for **e**, **f**, **h**, **i**, **j**, and **k**, or two-way ANOVA with Sidak's multiple comparisons test for (**b**). For **a**, **c**, **d**, and **g**, each image was acquired independently three times, with similar results. Source data are provided as a Source Data file.

PI/Hoechst staining. In addition, through H&E staining, we observed that more necrotic cells were evident in cells transfected with hP2Y$_{14}$R-expression vector, whereas, cells transfected with the empty vector did not exhibit that behavior (supplementary Fig. 2d). Furthermore, the phosphorylated of MLKL were substantially increased in TSZ treated cells compared with the control cells transfected with the empty vector (supplementary Fig. 2e). As we thought, pre-treatment with Nec-1s (RIPK1 inhibitor) and GSK'872 (RIPK3 inhibitor), respectively, alleviated TSZ stimulated IECs necroptosis in P2Y$_{14}$R-overexpression cells (supplementary Fig. 2b–e). Similarly, over-expression of P2Y$_{14}$R in HCT-116 cells also amplified the necroptosis of

IECs while pre-treated with Nec-1s and GSK'872 alleviated this phenomenon (supplementary Fig. 2f, g). Collectively, these results indicated that P2Y$_{14}$R mediated TSZ-induced necroptosis in IECs relied on RIPK1/RIPK3 pathway.

## P2Y$_{14}$R contributes to necroptosis in IECs by inhibiting the DNA binding ability of CREB to the promoter of *Ripk1*

It was observed that the modulation of P2Y$_{14}$R expression, either through silence or overexpression, had an impact on the expression of RIPK1, but not RIPK3, in HT-29 and HCT-116 cells (Fig. 3a, b). This suggested that P2Y$_{14}$R may play a role in necroptosis by regulating the

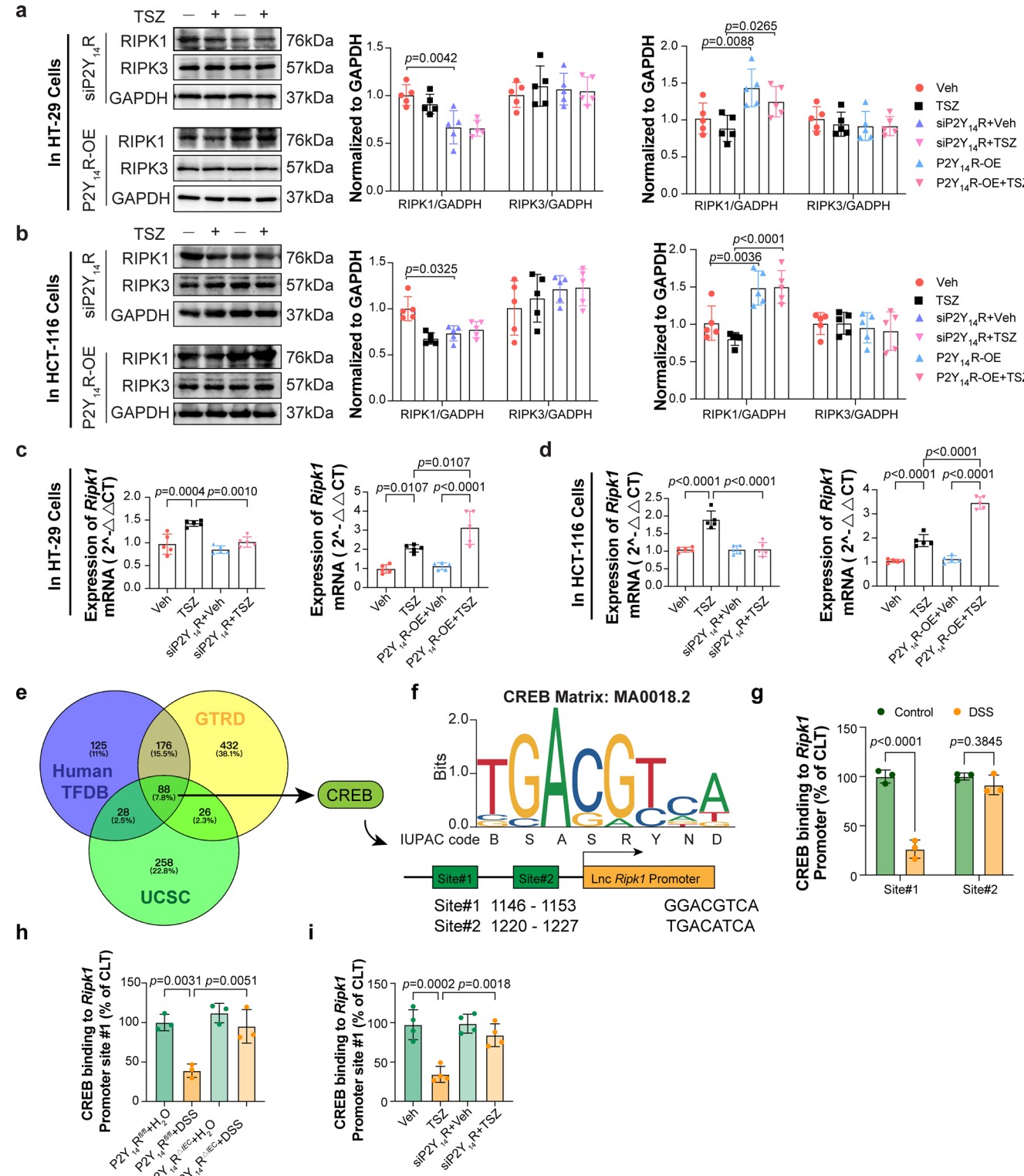

**Fig. 3 | P2Y₁₄R contributes to necroptosis in IECs by inhibiting DNA binding ability of CREB to the promoter of *Ripk1*. a** The expression of RIPK1 and RIPK3 were analyzed by immunoblotting in HT-29 cells (*n* = 5 samples per group). **b** The expression of RIPK1 and RIPK3 were analyzed by immunoblotting in HCT-116 cells (*n* = 5 samples per group). **c** The mRNA level of *Ripk1* in siP2Y₁₄R or P2Y₁₄R-OE HT-29 cells treated Veh or TNF-α adding Smac mimetic and z-VAD (TSZ) was analyzed by RT-PCR (*n* = 5 samples per group). **d** The mRNA level of *Ripk1* in siP2Y₁₄R or P2Y₁₄R-OE HCT-116 cells treated Veh or TSZ was analyzed by RT-PCR (*n* = 5 samples per group). **e** A Venn diagram showing potential transcription factors associated with RIPK1. **f** Partial sequences of the genomic *Ripk1* regions containing the underlined predicted CREB binding sites. **g** ChIP with anti-CREB of the regions

containing the CREB binding sites on the *Ripk1* gene promoter in DSS-treated IECs (*n* = 3 mice per group). **h** ChIP with anti-CREB of the regions containing the CREB binding site on the *Ripk1* gene promoter in DSS-treated P2Y₁₄R^*fl/fl* or P2Y₁₄R^*ΔIEC* IECs (*n* = 3 mice per group). **i** ChIP with anti-CREB of the regions containing the CREB binding sites on the *Ripk1* gene promoter in siP2Y₁₄R HT-29 cells 8 h after TSZ treatment (*n* = 4 samples per group). The data represent the mean ± SD. The *p*-values were determined by two-tailed Unpaired T test for **g**, One-way analysis of variance (ANOVA) with Tukey multiple comparison test for **c**, **d**, **h**, **i** and two-way analysis of variance (ANOVA) with Sidak's multiple comparisons test for **a**, **b**. Source data are provided as a Source Data file.

transcription of *Ripk1*. To verify this hypothesis, we examined the mRNA levels of *Ripk1* in HT-29 and HCT-116 cells. Based on the findings, the mRNA level of *Ripk1* showed a significant decrease in cells with silenced P2Y$_{14}$R. In contrast, overexpressing P2Y$_{14}$R resulted in an increase in *Ripk1* transcription 8 hours after TSZ treatment, both in HT-29 and HCT-116 cells (Fig. 3c, d). These results indicated that P2Y$_{14}$R contributes to necroptosis in IECs by inhibiting the transcription of *Ripk1*.

To explore the upstream molecular mechanisms of P2Y$_{14}$R to regulate the transcription of *Ripk1*, we used three online Transcription Factors (TFs) prediction websites (Human TFDB, GTRD, and UCSC) to predict potential TFs that could bind to the promoter of *Ripk1*. The results showed 88 common TFs between the three prediction websites, among them the cAMP-response element binding protein (CREB) attracted great attention of us (Fig. 3e). CREB was the downstream pathway of cAMP/PKA signal which has been reported to be activated by P2Y$_{14}$R deficiency in neutrophils in our previous study[15]. More importantly, Guida N and others reported that in neurons, the protein expression of the transcription factor CREB decreased in parallel with a reduction in binding to the RIPK1 gene promoter sequence, resulting in an increase of *Ripk1* expression[22]. Thus, we hypothesized that a similar process could occur in IECs. To investigate that hypothesis, we next searched for potential CRE sites in the *Ripk1* genomic regions. Interestingly, using the JASPAR databases (threshold score of 97.0), two putative CREB binding sequences in the genomic regions upstream of the *Ripk1* gene coding sequences were identified (Fig. 3f), which are located on the forward DNA strand from nucleotides −1146 to −1153 and −1220 to −1227, respectively. ChIP assays were performed to confirm the binding between CREB and the promoter of *Ripk1*. As shown in Fig. 3g, CREB could bind to the promoter of *Ripk1*. To further validate the influence of P2Y$_{14}$R on the binding between CREB and the promoter of *Ripk1*, we conducted ChIP assays in DSS-treated P2Y$_{14}$R$^{\Delta IEC}$ mice and TSZ-stimulated siP2Y$_{14}$R HT-29 cells. The results showed that DSS feeding in mice suppressed the binding between CREB and the promoter site#1 of *Ripk1*, whereas deficiency of P2Y$_{14}$R in IECs enhanced this binding (Fig. 3h). Similarly, silencing P2Y$_{14}$R in HT-29 cells improved the binding between CREB and the promoter of *Ripk1* under TSZ condition (Fig. 3i). Generally, P2Y$_{14}$R contributes to necroptosis in IECs by inhibiting the DNA binding ability of CREB to the promoter of *Ripk1*.

## P2Y$_{14}$R mediates the transcription of *Ripk1* through regulating cAMP/PKA/CREB pathway

Given the close connection between P2Y$_{14}$R, CREB, and cAMP/PKA pathways, we examined the expression patterns of cAMP, PKA, CREB in the presence or absence of cellular P2Y$_{14}$R after TSZ stimulating. As shown in Fig. 4a, P2Y$_{14}$R deficiency resulted in a significant elevation in the intracellular cAMP content in HT-29 cells when compared with control cells. Intriguingly, TSZ could not shift the intracellular level of cAMP anymore when HT-29 cells were transfected with si-P2Y$_{14}$R. Meanwhile, western blot showed that enhanced cAMP level under P2Y$_{14}$R silence further increased the phosphorylation of PKA and CREB (Fig. 4b). These changes were also observed in HCT-116 cells (Fig. 4c, d). Therefore, we mitigated cAMP/PKA/CREB pathway through treatment with SQ22536 (Adenylate cyclase inhibitor), H-89 (PKA inhibitor) or 666-15 (CREB inhibitor) respectively, which disrupted the siP2Y$_{14}$R-derived protective effect on TSZ-stimulated necroptosis (Fig. 4e–g). Besides, RT-PCR results showed that suppressed cAMP/PKA/CREB pathway upregulated the mRNA level of *Ripk1* in P2Y$_{14}$R silence HT-29 cells (Fig. 4h). Similar results were also observed in HCT-116 cells (Fig. 4i–k). In addition, we pre-treated P2Y$_{14}$R$^{\Delta IEC}$ organoids with SQ22536, H-89, or 666-15 and then induced necroptosis. As expected, inhibition of cAMP/PKA/CREB pathway disrupted the improvement of P2Y$_{14}$R deficiency on TSZ-mediated necroptosis (Fig. 4l). These results indicated that P2Y$_{14}$R regulated necroptosis in

IECs at least in part by altering the activation of cAMP/PKA/CREB pathway and further mediating the transcription of *Ripk1*.

## UDP-glucose promotes IECs necroptosis through activating P2Y$_{14}$R

UDP-glucose, the endogenous ligand of P2Y$_{14}$R, has been reported to be synthesized by UDP glucose pyrophosphorylase 2 (Ugp2) and then degraded by glycogen synthase 1 (Gys1) (Fig. 5a). Interestingly, the activity of P2Y$_{14}$R signaling cascade in UC could not due to the intracellular accumulation of UDPG, according to the analysis of GEO datasets, the expression of *Ugp2* and *Gys1* were decreased in IBD patient's intestinal mucosa (Fig. 5b), these changes were also observed in the IECs of DSS induced experiments colitis mice (Fig. 5c). It is notable that UDPG was reported to be released by necrotic hepatocytes, serving as a signaling molecule to promote the activation of hepatic stellate cells[12]. Therefore, we supposed that, in this study, the elevation of UDPG in the inflammatory microenvironment of UC was not directly related to the generation and degradation of UDPG but caused by dying IECs.

To examine this hypothesis, UDP-glucose was pre-administrated to the HT-29 cells for 12 h. As shown in Fig. 5d–i, we found that necroptosis induced with TSZ was increased after treatment with UDP-glucose, whether in HT-29 cells or HCT-116 cells. The necroptotic cell death was further confirmed using Nec-1s and GSK'872, where Nec-1s and GSK'872 treatment remarkably attenuated the UDP-glucose derived damage on TSZ-stimulated necroptosis. Furthermore, we observed that UDPG treatment increased necroptosis induced by TSZ in organoids from P2Y$_{14}$R$^{fl/fl}$ mice. The administration of Nec-1s and GSK'872 remarkably attenuated this damage (Fig. 5j). Additionally, UDPG enhanced the suppression of the cAMP/PKA/CREB pathway induced by TSZ treatment. Moreover, UDPG promoted the transcription of *Ripk1* and the inhibition effect on the binding between CREB and *Ripk1* promoter in both HT-29 cells and HCT-116 cells (Fig. 5k–p). These results strongly suggested that the accumulation of UDP-glucose might partly contribute to the activation of the P2Y$_{14}$R in regulating IECs necroptosis in IBD.

For better demonstrated the role of UDPG/P2Y$_{14}$R axis in mediating DSS-induced inflammation, we challenged P2Y$_{14}$R$^{fl/fl}$ and P2Y$_{14}$R$^{\Delta IEC}$ mice with 3% DSS, then rectal administration UDPG daily and monitored for weight and disease onset (supplementary Fig. 3a). We found that administration of UDPG aggravated DSS induced weight loss, diarrhea and rectal bleeding in P2Y$_{14}$R$^{fl/fl}$ mice. However, in P2Y$_{14}$R$^{\Delta IEC}$ mice, the administration of UDPG did not display obviously different with the mice administration blank vehicle. The results of colon length and histologic analysis also showed the similar results (supplementary Fig. 3b–d). These results indicated that UDPG regulation DSS-induced inflammation relied on the activity of P2Y$_{14}$R. In addition, we generated intestinal epithelial organoids from P2Y$_{14}$R$^{fl/fl}$ or P2Y$_{14}$R$^{\Delta IEC}$ mice and pretreated the organoids with UDPG 12 h before stimulating with TSZ. In P2Y$_{14}$R$^{fl/fl}$ organoids, the results of PI staining showed a dramatic increase of PI intensity in UDPG treated group. In contrast, UDPG treated in P2Y$_{14}$R$^{\Delta IEC}$ organoids did not display significant influence in TSZ-induced necroptosis (supplementary Fig. 3e). Collectively, these results strongly suggested that UDPG-P2Y$_{14}$R axis was involved in the regulation of DSS-induced inflammation by regulating the necroptosis of IECs.

## Discovery of P2Y$_{14}$R antagonist with expected potency and binding affinity

We next tried to find a P2Y$_{14}$R antagonist with excellent antagonistic activity and binding affinity to further verify the role of P2Y$_{14}$R in IECs necroptosis and IBD. Considering the unsolved crystal structure of P2Y$_{14}$R, the lack of possible key residues on P2Y$_{14}$R binding sites and high-throughput screening methods make it challenging to develop small-molecule antagonists targeting P2Y$_{14}$R. In our previous studies,

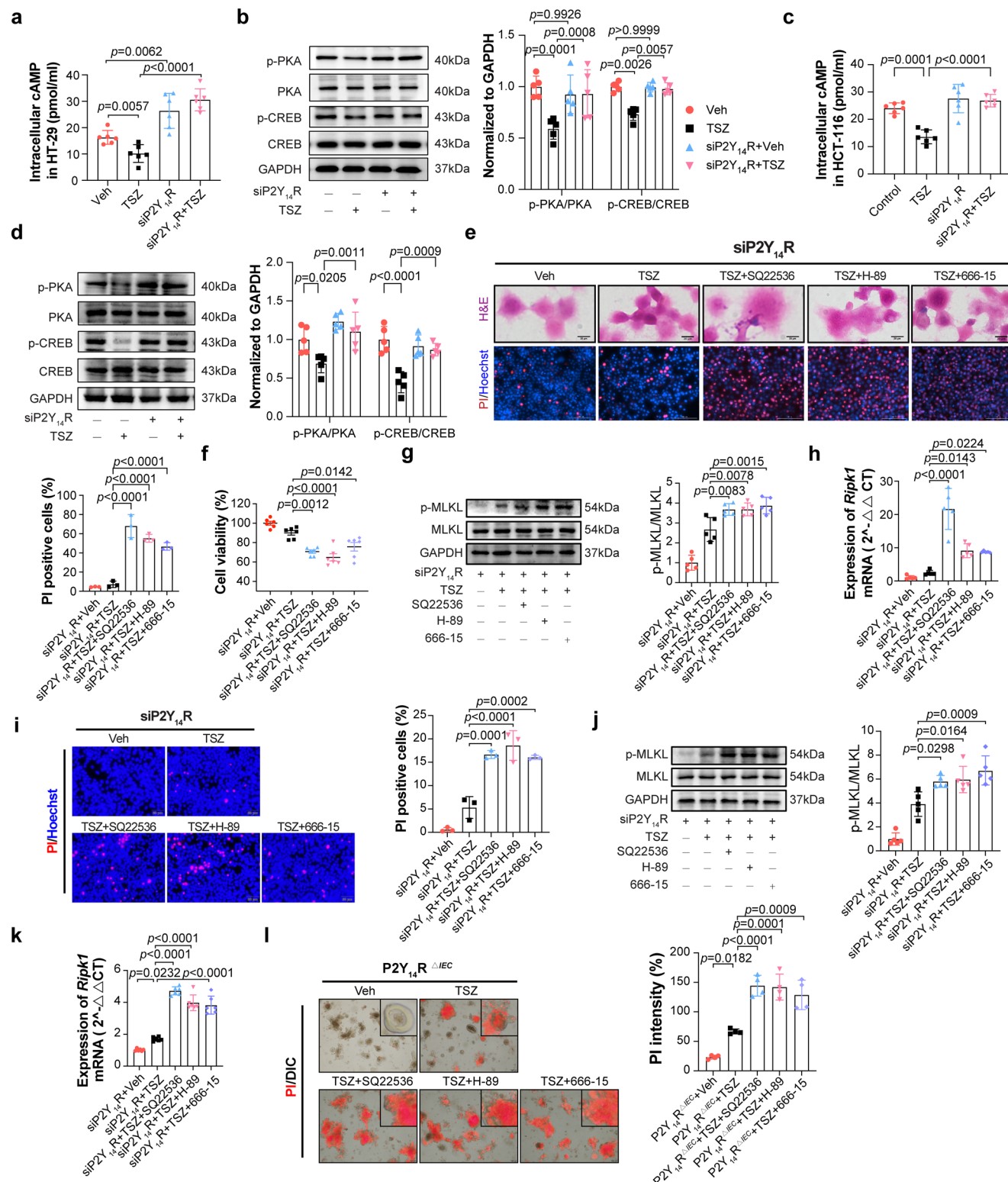

several potent P2Y$_{14}$R antagonists were discovered using the structure-based virtual screening (SBVS) and structure-activity relationship (SAR) research strategies[23]. A number of P2Y$_{14}$R antagonists with carboxyl group were identified by virtual screening protocol, however, the SAR research indicated that the carboxyl group was not an essential function group for receptor binding. These findings revealed that the key residues and explicit small-molecule binding sites to target P2Y$_{14}$R remained unknown[24]. Given the difficulties in small-molecule antagonists development, we first searched the most

probable conformation of the binding sites to find the critical residues on P2Y$_{14}$R structure by employing molecular dynamics (MD) simulations and MM/GBSA free energy calculations[25], and the results demonstrated that residues Tyr102, Val93, Cys172 and His184 were the most important binding determinants for small molecule-P2Y$_{14}$R interaction, and the favorable residue binding with P2Y$_{14}$R for small molecules with different chemical architectures varied significantly based on the inherent high flexibility of the P2Y$_{14}$R crystal structure (Fig. 6a).

**Fig. 4 | P2Y$_{14}$R mediates the transcription of *Ripk1* through regulating cAMP/PKA/CREB pathway. a** The intracellular cAMP level in siP2Y$_{14}$R HT-29 cells 8 h after TNF-α adding Smac mimetic and z-VAD (TSZ) treatment ($n = 6$ samples per group). **b** Phosphorylation of PKA and CREB as well as their protein levels were analyzed by immunoblotting in HT-29 cells 8 h after TSZ treatment ($n = 5$ samples per group). **c** The intracellular cAMP level in siP2Y$_{14}$R HCT-116 cells 8 h after TSZ treatment ($n = 6$ samples per group). **d** Phosphorylation of PKA and CREB as well as their protein levels were analyzed by immunoblotting in HCT-116 cells 8 h after TSZ treatment ($n = 5$ samples per group). **e** H&E cell staining and PI/Hoechst staining were used to analyze necroptotic cells in HT-29 cells ($n = 3$ samples per group). **f** Cell viability was determined by CCK8 analysis in HT-29 cells ($n = 6$ samples per group). **g** Phosphorylation of MLKL as well as its protein levels were analyzed by immunoblotting in HT-29 cells ($n = 5$ samples per group). **h** The mRNA level of *Ripk1*

in siP2Y$_{14}$R HT-29 cells was analyzed by RT-PCR ($n = 5$ samples per group). **i** PI/Hoechst staining were used to analyze necroptotic cells in HCT-116 cells ($n = 3$ samples per group). **j** Phosphorylation of MLKL as well as its protein levels were analyzed by immunoblotting in HCT-116 cells ($n = 5$ samples per group). **k** The mRNA level of *Ripk1* in siP2Y$_{14}$R HCT-116 cells was analyzed by RT-PCR ($n = 6$ samples per group). **l** The PI staining and quantification of intestinal organoids from P2Y$_{14}$R$^{\triangle IEC}$ mice treated as indicated ($n = 3$ samples per group). The data represent the mean ± SD, and *p*-values were determined by One-way analysis of variance (ANOVA) with Tukey multiple comparison test for **a, c, e–l** and two-way analysis of variance (ANOVA) with Sidak's multiple comparisons test for (**b**) and (**d**). For **e**, each image was acquired independently three times, with similar results. Source data are provided as a Source Data file.

Based on the binding determinants of the P2Y$_{14}$R structure, we designed a hierarchical strategy that combined virtual screening, bioassays, and chemical optimization to screen and found potential P2Y$_{14}$R antagonists. Firstly, a SBVS protocol based on binding area refinement, Glide docking with three precision levels of scoring (Glide HTVS, SP, and XP) and MM/GBSA free energy calculations (Fig. 6a) was employed[26]. Follow by drug-likeness prediction, REOS filtering and structural clustering, top-ranked 33 compounds were selected for biological assays. The P2Y$_{14}$R antagonistic activities of the compounds selected by in silico screening were evaluated based on production of cAMP in a HEK293 cell line stably expressing P2Y$_{14}$R, while compound **D4** showed satisfactory antagonistic activity of P2Y$_{14}$R (IC$_{50}$ = 25.4 nM). Through pharmacophore fusion strategy, we then designed a series of derivatives of compound **D4** to obtain **HDL-16** as a more potent antagonist (IC$_{50}$ = 0.3095 nM) with increased solubility for further structural biology and functional assays (Fig. 6b, d). **HDL-16** was docked into the binding site of the P2Y$_{14}$R homology model using the XP scoring mode of Glide, **HDL-16** and **D4** adopted a similar docking conformation in the binding sites of P2Y$_{14}$R. The N atom on benzoxazole ring of **HDL-16** forms a hydrogen bond with Tyr102 residue, and the amine group also forms a hydrogen bond interaction with His184 residue in the binding pocket of P2Y$_{14}$R (Fig. 6c), which was consistent with our previous MD simulations results. In addition, we detected the cytotoxicity of **HDL-16** in HT-29 cells, the results showed that **HDL-16** had almost no cytotoxicity under the concentration of 10 μM (Fig. 6e). Furthermore, we designed a **HDL-16**-FITC fluorescent conjugate probe suitable for imaging via confocal microscopy to confirm the interaction between **HDL-16** and the binding target P2Y$_{14}$R. Confocal-microscope images of **HDL-16**-FITC with HEK293 cells expressing P2Y$_{14}$R (hP2Y$_{14}$-HEK293) showed localized membrane fluorescence (Fig. 6f). In contrast, the location of **HDL-16**-FITC in HEK293 cells was mainly in intercellular space (Fig. 6f), indicated that the location of **HDL-16**-FITC in hP2Y$_{14}$-HEK293 cells relied on the expression of P2Y$_{14}$R in membrane. Besides, we designed a **HDL-16**-biotin conjugate probe, which could use to pull down the proteins interaction with **HDL-16**. The cell lysates of P2Y$_{14}$-HEK293 were incubated with 0.5 mM **HDL-16**-biotin probe, then the probe-protein complexes were collected with streptavidin-coupled Dynabeads. The P2Y$_{14}$R protein in probe-protein complexes was detected by western blot (Fig. 6g). The results showed that much more P2Y$_{14}$R protein was pulled down with **HDL-16** than the control group. Furthermore, cellular thermal shift assays (CETSA) showed that **HDL-16** increased the melting temperature (Tm) of P2Y$_{14}$R in a dose-dependent manner. The presence of 10 μM **HDL-16** increased the thermal stability of P2Y$_{14}$R by high temperature (50°C to 60°C) (Fig. 6h). Together, based on the binding site refinement, virtual screening, chemical optimization and in vitro assay, potent P2Y$_{14}$R antagonist **HDL-16** was developed through directly binding to the site of P2Y$_{14}$R.

### Pharmacological inhibition of P2Y$_{14}$R ameliorates in DSS-induced colitis

The above mentioned results suggested that decreased IECs necroptosis and intestinal inflammation were observed in DSS-treated

P2Y$_{14}$R$^{\triangle IEC}$ mice. Therefore, we tested whether the pharmacological blockade of P2Y$_{14}$R in mice treated with DSS could alleviate the necroptosis and inflammation (Fig. 7a). The results showed that daily treatment with **HDL-16** or **PPTN** significantly suppressed colitis symptoms and led to maintained gut barrier integrity in the mice. The mice treated with high-dose **HDL-16** (**HDL-16**-H) or **PPTN** showed dramatically less body weight loss and lower DAI than DSS-exposed mice (Fig. 7b). Meanwhile, the colons of the mice treated with high-dose **HDL-16** or **PPTN** were significantly longer than those of the mice treated with DSS only (Fig. 7c). In contrast, comparison of low-dose **HDL-16** (**HDL-16**-L) treatment with the DSS-treaded group revealed no significant difference. The results of H&E staining showed that administration with high-dose **HDL-16** or **PPTN** improved DSS-induced inflammatory infiltration and tissue damage (Fig. 7d). Additionally, the serum level of FITC-dextran in the DSS-treated mice was significantly higher than in the high-dose **HDL-16** or **PPTN** group (Fig. 7e). Consist with these, immunofluorescence results showed that treatment with high-dose **HDL-16** or **PPTN** stabilized the expression of tight junction protein Claudin-1, Occludin and ZO-1 to improve mucosal barrier function (Fig. 7f). Furthermore, both the level of p-MLKL and TEM imaging showed that treatment with either high-dose **HDL-16** or **PPTN** during DSS feeding resulted in a prominently suppression in necroptosis of the IECs in DSS-treated mice (Fig. 7g, h). In addition, the results of TUNEL staining and the expression of cleave caspase-3, Bcl-2, Bax, GSDMD-NT and caspase-1 p20 strong suggested that administration of **HDL-16** did not influence the apoptosis and pyroptosis of IECs during DSS-induced inflammation (supplementary Fig. 4a, b). Besides, we found that daily administration with **HDL-16** or **PPTN** showed barely improvement in DSS feeding P2Y$_{14}$R$^{\triangle IEC}$ mice, indicating that the improvement effect of **HDL-16** on DSS-induced experimental colitis relied on targeting P2Y$_{14}$R of IECs (supplementary Fig. 4c–i).

Mechanically, **HDL-16** treatment upregulated the activation of the cAMP/PKA/CREB pathway suppressed by DSS treatment (Fig. 7h, i). Moreover, it promoted the binding between CREB and *Ripk1* promoter in DSS-induced experiment colitis (Fig. 7j). Additionally, in vitro experiment showed that **HDL-16** treatment could inhibit the necroptosis of HT-29 cells according to the result of PI staining (Fig. 7k). Finally, we pre-treated P2Y$_{14}$R$^{\triangle IEC}$ organoids with **HDL-16** and then induced necroptosis. We found that administration with **HDL-16** also protected P2Y$_{14}$R$^{fl/fl}$ organoids from TSZ-induced necroptosis (Fig. 7l). These data suggested that treatment with P2Y$_{14}$R antagonist ameliorated DSS-induced colitis through suppressing necroptosis of IECs and protecting mucosal barrier function.

## Discussion

The present study focused on the role of P2Y$_{14}$R in the pathogenesis of UC, proving a potent target for related drug discovery. Since up-regulated P2Y$_{14}$R was found in the intestinal epithelium tissue of UC patients and DSS-treated mice, we systematically investigated the detailed mechanism by which P2Y$_{14}$R participated in the development

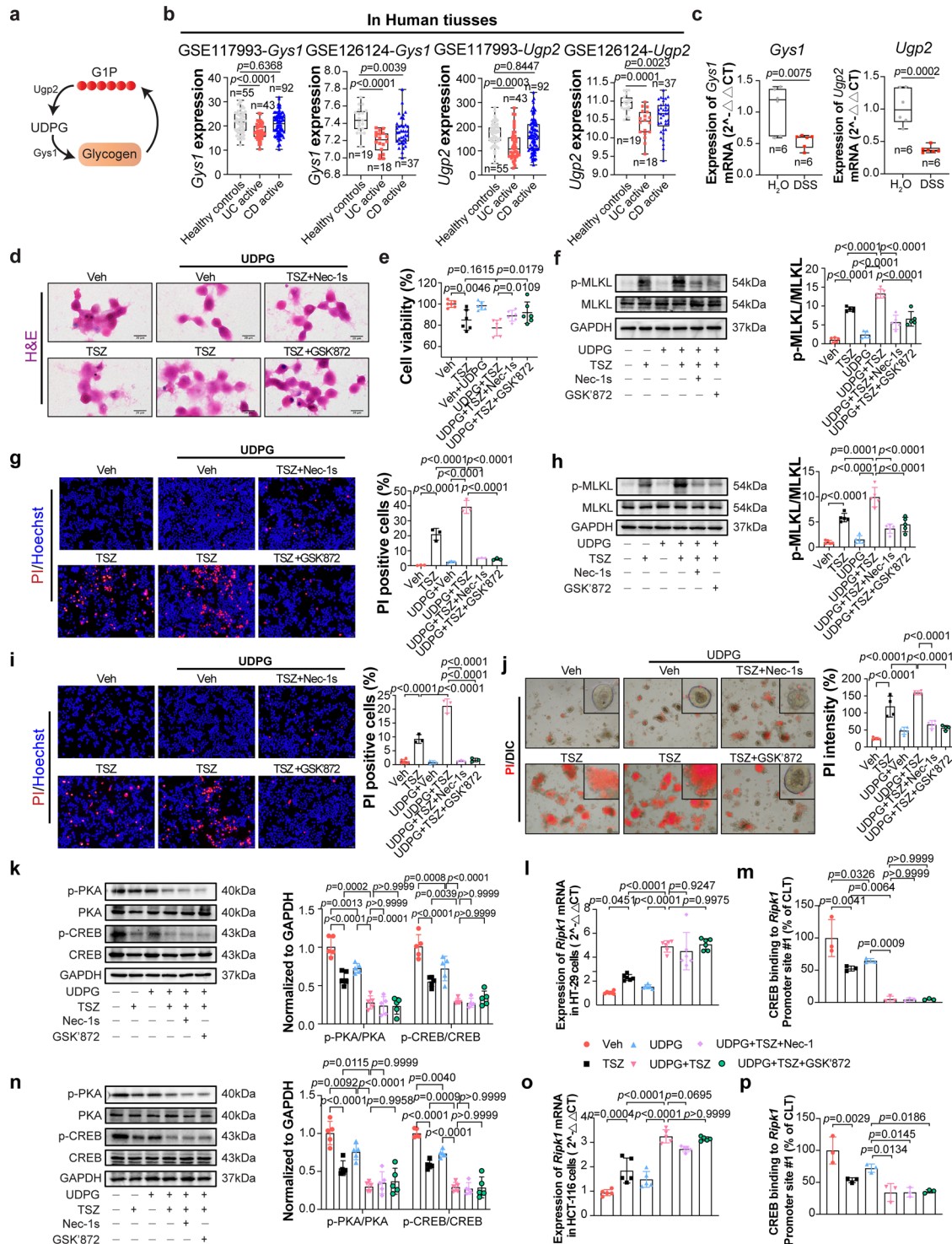

**Fig. 5 | UDP-glucose promotes IECs necroptosis through activating P2Y₁₄R.**
**a** Overview of UDP-glucose metabolic process. **b** *Ugp2* and *Gys1* expression in the colonic mucosa of healthy individuals or IBD patients from GEO database (GSE117993, GSE126124). **c** RT-qPCR analysis of *Ugp2* and *Gys1* mRNA of IECs in mice, the box plot appearance as box and whisker ($n = 6$ mice per group). **d** H&E cell staining was used to analyze necroptotic cells in HT-29 cells, each image was acquired independently three times, with similar results. **e** Cell viability was determined by CCK8 analysis in HT-29 cells ($n = 6$ samples per group). **f** Phosphorylation of MLKL and its protein levels in HT-29 cells ($n = 5$ samples per group). **g** The PI/Hoechst staining was used to analyze necroptotic cells in HCT-116 cells ($n = 3$ samples per group). **h** Phosphorylation of MLKL and its protein levels in HCT-116 cells ($n = 5$ samples per group). **i** The PI/Hoechst staining was used to analyze necroptotic cells ($n = 3$ samples per group). **j** The PI staining and

quantification of intestinal organoids from P2Y₁₄R$^{fl/fl}$ mice ($n = 3$ samples per group). **k** Phosphorylation of PKA and CREB and their protein levels in HT-29 cells ($n = 5$ samples per group). **l** mRNA level of *Ripk1* in HT-29 cells ($n = 6$ samples per group). **m** ChIP with anti-CREB of the regions containing the CREB binding sites on the *Ripk1* gene promoter ($n = 3$ samples per group). **n** Phosphorylation of PKA and CREB and their protein levels in HCT-116 cells ($n = 5$ samples per group). **o** mRNA level of *Ripk1* in HCT-116 cells ($n = 5$ samples per group). **p** ChIP with anti-CREB of the regions containing the CREB binding sites on the *Ripk1* gene promoter ($n = 3$ samples per group). The data represent the mean ± SD, and *p*-values were determined by One-way ANOVA with Tukey multiple comparison test for **b**–**j**, **l**, **m**, **o**, **p** or two-way ANOVA with Sidak's multiple comparisons test for **k** and **n**. Source data are provided as a Source Data file.

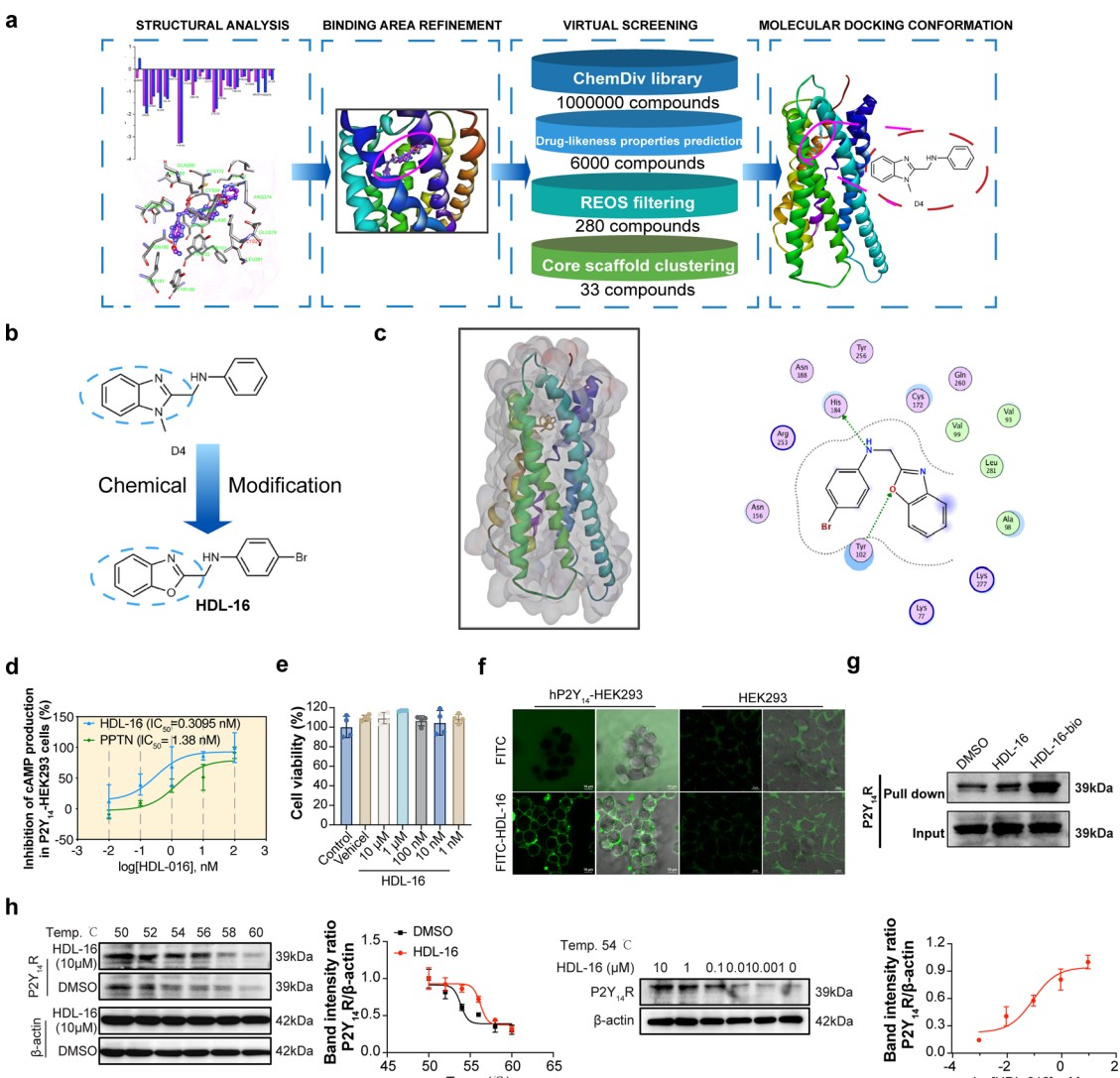

**Fig. 6 | Discovery of P2Y$_{14}$R antagonist. a** Workflow of SBVS. **b** Chemical modification of D4. **c** The predicted binding poses and interaction patterns of homology model of P2Y$_{14}$R. **d** Concentration–inhibition curves of **HDL-16** determined using the cAMP production assay with P2Y$_{14}$-HEK293 cells. **e** Cytotoxicity of **HDL-16** in HT-29 cells was determined by CCK8 analysis after 12 h treated ($n = 4$ samples per group). **f** Confocal imaging of **HDL-16**-FITC fluorescent conjugate probe in hP2Y$_{14}$-HEK293 cells or HEK293 cells. **g** **HDL-16**-biotin conjugate probe specifically interacted with P2Y$_{14}$R in P2Y$_{14}$-HEK293 cells, which was evaluated by a pull-down assay. **h** Cells were incubated with **HDL-16** or DMSO for 30 min, and cellular thermal shift assays (CETSA) analyzed the thermal stabilization of P2Y$_{14}$R protein at different temperatures and concentrations. Data points shown are the mean values of at three independent experiments, each performed in duplicate. For **f** and **g**, each image was acquired independently three times, with similar results. The data obtained from three independent experiments and represented as mean ± SD. Statistical significance was determined by One-way ANOVA with Tukey multiple comparison test. Source data are provided as a Source Data file.

of UC. Deficiency of P2Y$_{14}$R significantly improved intestinal injury through inhibiting necroptosis of IECs via typical cAMP/PKA/CREB signals, which might be attributed to the direct bonding between CREB and the promoter of *Ripk1*. On the other hand, as a potent antagonist of P2Y$_{14}$R synthesized by our group, **HDL-16** was used to further verify the feasibility of P2Y$_{14}$R-targeted therapy in the treatment of UC.

Purinergic receptors are implicated in the pathogenesis of gastrointestinal disorders and are being investigated as potential therapeutic targets[10]. Among these purinoceptors, P2X7 and its endogenous ligand ATP play important roles in both the initiation and exacerbation of intestinal inflammation, which have been widely reported[27]. P2Y$_6$R, another essential purinergic receptor, has also been proven to participate in IBD, whole-body P2Y$_6$R knockout protected mice against DSS-induced colitis[18,28]. P2Y$_{14}$R acknowledged as a G-protein-coupled receptors (GPCRs) combining with extracellular nucleoside, has attracted increasing attention for its immunomodulatory effects[16]. Our previous work revealed the role of P2Y$_{14}$R in

regulating caspase-1-mediated pyroptosis in macrophages and NETosis of neutrophils during the pathogenesis of acute gouty arthritis[14,15]. In addition, Mederacke and his colleagues demonstrated that UDPG/P2Y$_{14}$R signaling could link hepatocyte death to hepatic stellate cells activation, leading to fibrogenesis in the injured liver[12]. Nevertheless, there was no evidence showing the potential function of P2Y$_{14}$R in IBD, especially in the intestinal epithelial cell.

Compelling evidences from various experimental mouse models suggested that excessive cell death in the intestinal epithelium was sufficient to induce intestinal inflammation[8,29,30]. It is therefore tempting to speculate the dysregulation of cell death pathways in IECs involved in the pathogenesis of IBD in human. Necroptosis is a kind of programmed cell death, which plays an important role in the development of IBD[5,31]. Mechanistically, RIPK1 autophosphorylation initiated RIPK3 homo-oligomerization followed by MLKL recruited and ultimately causing necroptosis[32,33]. Interestingly, it was reported that IEC-specially caspase-8 knockout mice showed increased level of

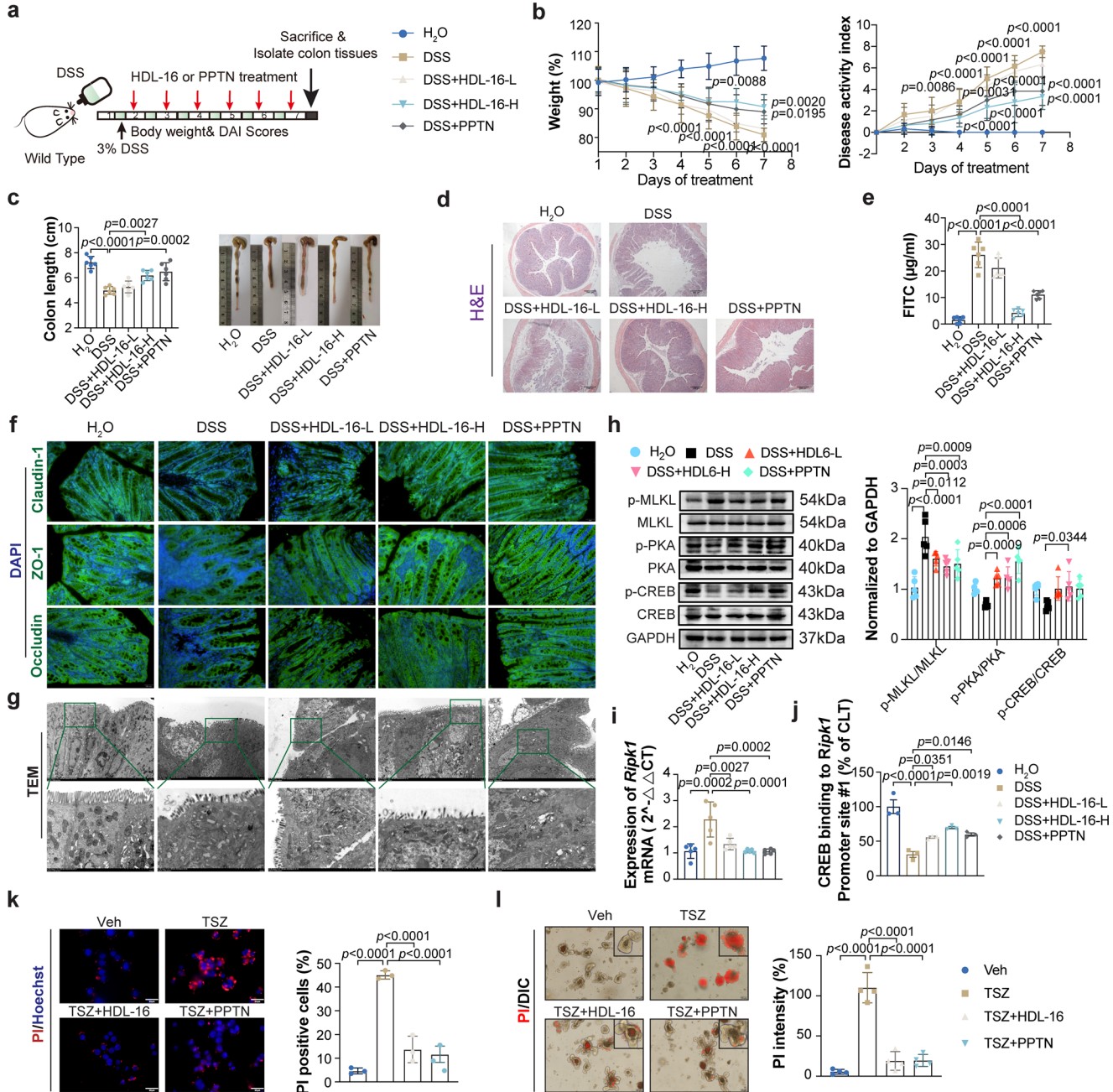

**Fig. 7 | Pharmacological inhibition of P2Y$_{14}$R ameliorates in DSS-induced colitis. a** Experimental flow chart. **b** Body weight and disease activity index evaluation of mice change during the disease process ($n = 6$ mice per group). **c** The length of colons from mice 7 days after DSS treatment ($n = 6$ mice per group). **d** The H&E staining in the colon tissues of DSS-treated mice (Scale bar = 200 μm). **e** Effects of P2Y$_{14}$R antagonists on mucosal barrier function as measured by serum level of FITC-dextran based on intestinal permeability method ($n = 6$ mice per group). **f** The immunofluorescent images of colon tissues stained with Claudin-1, Occludin and ZO-1, the principal components of tight junction (Scale bar = 200 μm). **g** The TEM images showed the typical characteristics of necroptosis in the colon tissues of DSS-treated mice. **h** Phosphorylation MLKL, PKA, CREB as well as its protein levels were analyzed by immunoblotting in colon tissues ($n = 5$ mice per group). **i** mRNA level of *Ripk1* was analyzed by RT-PCR in **HDL**-16-L, **HDL**-16-H or **PPTN**

administrated experimental colitis mice intestinal epithelial cell ($n = 5$ mice per group). **j** ChIP with anti-CREB of the regions containing the CREB binding sites on the *Ripk1* gene promoter in IECs of **HDL**-16-L, **HDL**-16-H or **PPTN** administrated experimental colitis mice ($n = 3$ mice per group). **k** PI positive cells were analyzed by PI/Hoechst staining in TNF-α adding Smac mimetic and z-VAD (TSZ) treated HT-29 cells after pre-administrated with **HDL**-16 ($n = 3$ samples per group). **l** The PI staining and quantification of intestinal organoids from P2Y$_{14}$R$^{fl/fl}$ mice treated as indicated ($n = 3$ samples per group). The data represent the mean ± SD for **c**, **e**, and **h–l**, the data represent the mean ± SEM for **b**. The *p*-values were determined by One-way ANOVA with Tukey multiple comparison test for **c**, **e**, **i–l** or two-way ANOVA with Sidak's multiple comparisons test for **b** and **h**. For **d**, **f**, and **g**, each image was acquired independently three times, with similar results. Source data are provided as a Source Data file.

RIPK3 and high susceptibility to colitis, expectedly, knocking down RIPK3 decreased intestinal inflammation in caspase-8 knockout mice to a certain degree[8]. Consistently, our observation exhibited that IEC-specially P2Y$_{14}$R knockout attenuated the severity of colitis through reversing typical of the necroptotic process in the colon tissues of DSS-

treated mice. Notably, as P2Y$_{14}$R specific ligand, UDP-glucose stimulated P2Y$_{14}$R-related signals like a damage-associated molecular pattern (DAMP) released after hepatocyte death, promoting HSC activation and fibrogenesis[12]. Similarly, in our present study, UDP-glucose might act as tissue messenger released from dying IECs to

aggravate necroptosis cascades through P2Y14R distributed in IECs, forming a feedback loop on the basis of the DAMP-DAMP receptor system.

The activation of P2Y14R is closely related to the content of intracellular cAMP, which is strongly involved in inflammation, diabetes, immune processes and other related complications[34]. Previous study implied that in patients with colitis or IBD, disease risk genes could result in insufficient cAMP activation in T cells and macrophages through cAMP-PKA-CREB-ATF2 signaling, targeting this pathway by the phosphodiesterase inhibitor dipyridamole restored immune homeostasis and improved colitis symptoms[35]. Additionally, cAMP-PKA signaling pathway was reported to regulate necroptosis of adipocytes in a CREB-dependent manner[36]. In our present study, we found that P2Y14R deletion inhibited TSZ-induced necroptosis in vitro attributed to activation of cAMP-PKA-CREB signals, as evidenced by SQ22536 (Adenylate cyclase inhibitor), H-89 (PKA inhibitor) or 666-15 (CREB inhibitor) pre-treatment, which restored the necroptosis of cells under P2Y14R silencing.

More importantly, we found that CREB reduced the transcription of *Ripk1* by binding to the promoter sequence of *Ripk1*, a crucial component involved in necroptosis, suggested that targeting P2Y14R could regulate RIPK1-mediated cell death in UC. Notably, a previous study showed that overexpression of CREB in SH-SY5Y cells could reduce the expression of RIPK1 by suppressing the promoter activity of the *Ripk1*, which was consistent with our findings[22].

Collectively, the evidences above indicated that P2Y14R was involved in the pathogenesis of UC, suggesting that P2Y14R might be an alternative therapeutic target for IBD. As members of GPCR, P2YRs have been regarded as effective therapeutic target for treatment of

disease, for instance, clopidogrel was widely used in the treatment of thrombosis P2Y12R inhibitor[37]. However, the lack of possible key residues on P2Y14R binding sites and P2Y14R-specific antagonist has hindered further pharmacological research and development. Among reported P2Y14R antagonists, **PPTN** was considered to be the most potent and selective P2Y14R antagonist ($IC_{50} = 0.3095$ nM), but **PPTN** suffers from poor solubility and low oral bioavailability due to its high lipophilicity and zwitterionic character[38]. In order to verify whether P2Y14R could be a potent therapeutic target for UC, we designed a screening workflow based on molecular dynamics simulations and MM/GBSA free energy calculations that identified **HDL-16** as a potent antagonist for further structural biology and functional assays. Interestingly, **HDL-16** was observed to ameliorated DSS-induced colitis through suppressing necroptosis of IECs, the beneficial effects of which were diminished in $P2Y_{14}R^{\triangle IEC}$ mice. These findings further proved that the causal role of P2Y14R in the pathogenesis of IBD, providing a promising therapeutic strategy for IBD (Fig. 8).

However, our study contains several limitations. Although we revealed the regulatory effects of UDPG/P2Y14R axis on the cell fate of IECs, it is likely that cell death promotes intestinal inflammation through additional receptors activated by the many different mediators secreted from dying cells. On the other hand, as P2Y14R ligand release has been described to occur in stressed cells[16], it is conceivable that activation of the P2Y14R-mediated necroptosis cascades in IECs could be activated by stress signals in addition to cell death, which would still be consistent with its function as DAMP-DAMP receptor system. Therefore, the exact mechanism by which P2Y14R in IECs is activated during acute phase of UC needs to be further investigated. Nevertheless, our findings suggested that P2Y14R-targeted

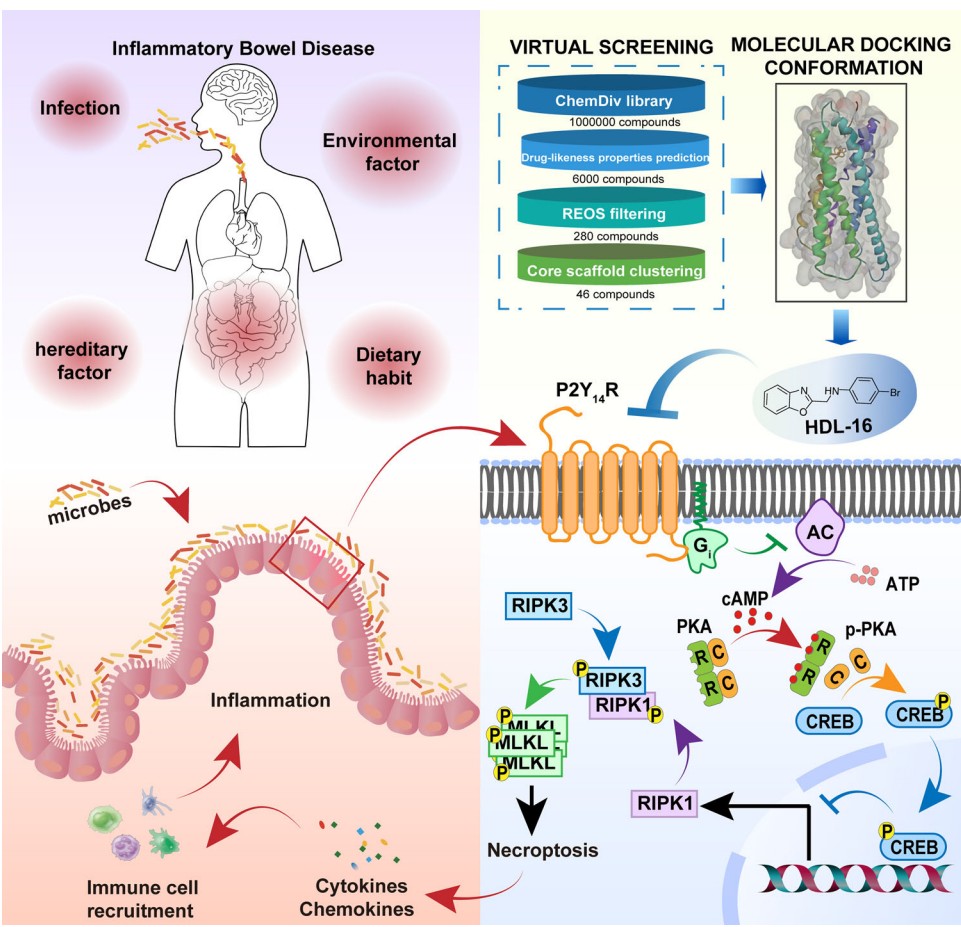

**Fig. 8 | Graph abstracts.** P2Y14R activation is involved in the pathogenesis of IBD by regulating necroptosis of IECs via PKA/CREB/RIPK1 axis. P2Y14R-targeted intervention by a potent antagonist **HDL-16** exhibits notable therapeutic effects on DSS-induced colitis in mice.

intervention inhibited aberrant transcription of *Ripk1* in IECs, which has emerged as a promising therapeutic target for the treatment of UC[39]. P2Y$_{14}$R antagonist **HDL-16** is expected to become a drug candidate to treat IBD through suppressing necroptosis in inflammatory bowel disease.

## Methods

### Ethics statement

Colon biopsy samples for non-IBD, UC, or CD were provided from Nanjing First Hospital. This study was approved by the Institutional Review Board of Nanjing First Hospital (KY20180604-05-KS-01) and informed consent was obtained from the study participants.

All animal experiments in this study were performed in conformity with the Guide for the Care and Use of Laboratory Animals (NIH publication No. 85-23, 1996 revision) and approved by the China Pharmaceutical University Committee for Laboratory Animal Use (2021-12-029).

### Animals

The P2Y$_{14}$R$^{fl/fl}$, *Villin-cre*, *Lyz2-cre*, and wild-type (WT) mice with C57BL/6 J background were purchased from Gempharmatech (GemPharmatech Co., Ltd) and bred onsite to generate animals for experimentation. All mice were housed in a controlled environment (20 ± 2 °C, 40–60% humidity, 12-hour/12-hour light/dark cycle), where they were maintained on a standard chow diet (1010088, 1010083, Jiangsu Xietong Pharmaceutical Bio-engineering Co., Ltd.) with free access to water.

### Cell culture

HT-29 cells and HCT-116 cells were purchased from the BeNa Culture Collection Co., ltd and HEK293 cells stably expressing the hP2Y$_{14}$R (P2Y$_{14}$-HEK293 cells) were purchased from Keygen Biotech Co, ltd. Cells were cultured in Dulbecco' s modified Eagle's medium supplemented with 1% penicillin/streptomycin and 10% FBS at 37 °C in a 5% CO$_2$ environment as previously described.

### Organoid culture and live imaging

Crypts were released from the murine small intestine by incubation for 30 min at 4°C in PBS containing 2 mM EDTA. Isolated crypts were counted and pelleted. A total of 500 crypts were mixed with 35 μl of Matrigel (356234, BD Bioscience) and grown in mouse Intesticult (06005, Stemcell Technologies). Organoid media were changed every 3 days. Necroptosis in organoid was induced by pretreatment with TNF-α (20 ng·mL$^{-1}$) (HY-P70426A, MCE) plus z-VAD-fmk (20 μM) (HY-16658B, MCE) and Smac mimetic (BV6, 2 μM) (HY-16701, MCE) for 12 h, then stained cells with 50 μg ml$^{-1}$ PI for 20 min. Dead cell imaging in live organoids was performed on the fluorescence microscope (BX53, Olympus Optical), and analyzed by image j.

### Overexpression construct and siRNA knockdown

The plasmid of human P2Y$_{14}$R and its control were purchased from Genepharma (Shanghai, China). Transfection was performed using Lipofectamine™ 2000 from Invitrogen. The infection rate was evaluated through the expression of green fluorescent protein after incubation with plasmid after 48 h. siRNA knockdown for human P2Y$_{14}$R and its control were synthesized by Genepharma (Shanghai, China). Transfection was performed using Lipofectamine™ 2000 from Invitrogen. Knockdown efficiency was determined by immunoblotting. The target sequences were as follows: 5'–3': CCUUAAGUCGGAAUTT. 3'–5': AUUCCGACUUGACUUAAGGTT. The results of transfection efficiency are shown in supplementary Fig. 8.

### Necroptosis induction and cell death analysis

Necroptosis was induced by pretreatment with TNF-α (20 ng·mL$^{-1}$) plus z-VAD-fmk (20 μM) and Smac mimetic (BV6, 2 μM) for 8 h, then

stained cells with hematoxylin/eosin. For PI/Hoechst staining, cells were stained with PI and Hoechst for 20 min, then photographed with a fluorescence microscope and at least 300 cells were counted. The ratio of PI positive cells (%) = (PI positive cells) / (Hoechst positive cells) × 100%. For cell viability assay, CCK8 was added to the well and incubated for 1 – 2 h and then OD450 was measured using a multi-mode microplate reader (Cytation5, BioTek).

### Induction and assessment of experimental (acute) colitis

Male, 7-8-week-old, P2Y$_{14}$R$^{fl/fl}$ *Vil-cre* (P2Y$_{14}$R$^{ΔIEC}$), P2Y$_{14}$R$^{fl/fl}$ *Lyz2-cre* and WT mice with C57BL/6 J background were used to establish experimental (acute) colitis model. P2Y$_{14}$R$^{fl/fl}$ C57BL/6 J mice were used as control for P2Y$_{14}$R$^{fl/fl}$ *Vil-cre* (P2Y$_{14}$R$^{ΔIEC}$) and P2Y$_{14}$R$^{fl/fl}$ *Lyz2-cre* mice. WT C57BL/6 J mice were used for the pharmacological study of **HDL-16**. Colitis was induced by dextran sodium sulfate (DSS; 36-50 kDa; 3% w/v; MP Biomedicals, Solon, Ohio, USA), added to the drinking water ad libitum for 7 days. Control mice received standard drinking water. The Disease Activity Index (DAI) was evaluated daily as described previously. After 7 days of DSS treatment, mice were euthanized by carbon dioxide inhalation and the colons were measured and then processed for histological analyses, homogenized to extract protein or RNA.

### Induction and assessment of experimental (chronic) colitis

Male, 7-8-week-old, P2Y$_{14}$R$^{fl/fl}$ *Vil-cre* (P2Y$_{14}$R$^{ΔIEC}$) mice with C57BL/6 J background were used to establish experimental (chronic) colitis model. P2Y$_{14}$R$^{fl/fl}$ C57BL/6 J mice were used as control for P2Y$_{14}$R$^{fl/fl}$ *Vil-cre* (P2Y$_{14}$R$^{ΔIEC}$) mice. To induce chronic colitis, a DSS concentration of 2% (w/v) in drinking water for 7 days, then replace the remaining DSS solution at day 8 by autoclaved drinking water without DSS for 14 days. Repeat this cycle (7 days DSS, 14 days water) three times[40]. Then the mice were euthanized by carbon dioxide inhalation, and the colons were measured and then processed for histological analyses, homogenized to extract protein or RNA.

### Administration

**HDL-16** and **PPTN** were dissolved in DMSO and diluted in saline to a final concentration before administration. The DSS group and the P2Y$_{14}$R$^{ΔIEC}$ + DSS group were rectally administered with 100 μl vehicle (saline with 0.1% DMSO). The low-dose group of mice was given 10 μM **HDL-16**, 100 μl per mouse, while the high-dose group was given 20 μM **HDL-16**, 100 μl per mouse. The group receiving **PPTN** administration was given 20 μM **PPTN**, 100 μl per mouse.

### Measuring intestinal permeability

The mice treated with DSS for 7 days were fasted for 6 h, given FITC-dextran (3-5 kDa, FD4, sigma) intragastrically at a dose of 60 mg kg$^{-1}$. After 3 h, hemolysis-free sera were collected and the fluorescence intensity of serum was read by a multi-mode microplate reader (excitation, 488 nm; emission, 520 nm).

### Transmission electron microscopy

For TEM, harvest fresh tissue blocks quickly within 1-3 minutes. Transferred the fresh tissue into fresh TEM fixative (2.5% glutaraldehyde in PBS) for fixation at 4°C overnight. Then tissues avoid light post fixed in 1% OsO4 in PBS for 2 h at room temperature. After fixation, the tissue samples were dehydrated through gradient ethanol as follows: 30% ethanol for 20 min, 50% ethanol for 20 min, 70% ethanol for 20 min, 80% ethanol for 20 min, 95% ethanol for 20 min, two changes of 100% ethanol for 20 min, two changes of acetone for 15 min, then progressively embedded in epon epoxy resin. The samples were moved into a 65°C oven to polymerize for more than 48 h. Ultrathin sections were cut to 60-80 nm thin with an ultramicrotome UCT6 (Leica Microsystems, Vienna) and placed on 150 meshes cuprum grids (formvar carbon coated Cu grids) with formvar film. The grids were further contrasted with 2% uranium acetate and 2.6% Lead citrate,

and dried overnight at room temperature. Micrographs were obtained with a Jeol JEM 1400 plus electron microscope (Jeol, USA) operating at 80 kV.

## Intestinal epithelial cell isolation

Colon tissues derived from the euthanized by carbon dioxide inhalation mice were opened longitudinally and washed in pre-cooling PBS for three times. Then, the tissues were cut into 3 mm$^2$ pieces and incubated with 5 mM EDTA solution in PBS on a constant temperature shaker at 200 rpm for 25 min at 37°C. Then, let it stand at room temperature for 30 seconds and collected the supernatant. Next, repeated the above process once. Supernatants from the above step were combined and centrifuged at 500 g for 8 min. Abandon the supernatants and obtain IECs.

## Immunoblotting

Tissues and cells were lysed using RIPA lysis buffer (Beyotime Biotechnology) containing protease and phosphatase inhibitor, homogenated, and centrifuged. The protein concentration was determined by BCA protein assay (Beyotime Biotechnology). Equal concentrations of proteins were separated in a 10% SDS-PAGE and transferred to PVDF membrane. The membrane was blocked using 5% milk in TBST, probed with primary antibody, RIPK1 (17519-1-AP, Proteintech, 1:1000), MLKL (GTX107538-100, GeneTex, 1:1000), RIPK3 (17563-1-AP, Proteintech, 1:1000), p-MLKL (EPR9514, Abcam, 1:1000), PKA (bs-0520R, Bioss, 1:1000), p-PKA (#9621, Cell signaling, 1:1000), CREB (AF6188, Affinity, 1:1000), p-CREB (AF3189, Affinity, 1:1000), P2Y$_{14}$R (bs-12028R, Bioss, 1:1000), GSDMD (ab209845, Abcam, 1:1000), Caspase1 p20 (bs-10442R, Bioss, 1:1000), Bcl-2 (bs-0032R, Bioss, 1:1000), Bax (AF0120, Affinity, 1:1000), Caspase-3 p-17 (Cell Signaling Technology, #9664, 1:1000), β-actin (bs-0061R, Bioss, 1:1000), GAPDH (BS-2188R, Bioss, 1:1000) and corresponding HRP-conjugated antibodies Goat Anti-Rabbit IgG H&L Antibody (bioss, Bs-0295G, 1:8000) or Goat Anti-Mouse IgG H&L Antibody (bioss, Bs-0296G, 1:8000). Densitometry was quantified using ImageJ software.

## RT-qPCR and ChIP-qPCR assays

Total RNA was extracted from cells or colonic tissues using Trizol (Thermo) under the manufacturer's instructions. First strand cDNA was synthesized using the PrimeScript RT Reagent Kit (480, Abm). SYBR Green Master Mix reagents (Q331-02, Vazyme) and primer mixtures (Table S1) were used for the real-time PCR. The ChIP assays were performed using EZ ChIP kit (17-371, Millipore). The procedure was as described in the kit provided by the manufacturer. Briefly, isolated IECs were fixed by 1% formaldehyde, and fragmented by sonication. CREB was then used for immunoprecipitation. After washing and reverse-crosslinking, the precipitated DNA was amplified by primers and quantified by the qPCR. Primer sequences can be found in Supplementary Table 1.

## Histopathological analysis

For histological assessment of colitis, colon specimens were fixed in 4% paraformaldehyde (PFA) and embedded in paraffin. Five-micrometer tissue sections were stained with H&E and examined for evidence of colitis as described previously. After the paraffin sections conventional dewaxing to water, put the sections into Hematoxylin solution for 3-5 min and treat the sections with Hematoxylin Differentiation solution. Next, place the sections in 95% ethanol for 1 min, Eosin dye for 15 s. Finally, dehydrated the sections with gradient alcohol and xylene, and sealed with neutral gum. Images were captured with a microscope (BX53, Olympus).

## Immunofluorescence

Immunofluorescence staining was performed using paraffin-embedded tissues as mentioned before. For immunofluorescence,

the paraffin slides were deparaffinized, rehydrated, blocked and treated according to a standard protocol. The expression of intestinal barrier associated tight junction protein, P2Y$_{14}$R and EpCAM was evaluated by probing the tissues with primary antibody against Claudin-1 (bioss, bs-10011R, 1:300), Occludin (bioss, DF6919, 1:300), ZO-1 (Affinity, AF5145, 1:300), EpCAM (AiFang, AF04654, 1:300), Caspase-3 p-17 (Cell Signaling Technology, #9664, 1:1000) and P2Y$_{14}$R (Invitrogen, PA5-103202, 1:300) overnight at 4°C followed by 1 h incubation with the corresponding secondary antibody Goat Anti-Mouse IgG H&L (Alexa Fluor® 555) (ab150114, Abcam, 1:500) or Goat Anti-Rabbit IgG H&L (Alexa Fluor® 488) (ab150077, Abcam, 1:300). All slides were incubated with DAPI (S2110, Solarbio) for 10 min to show the location of the nucleus. Images were captured with a confocal microscope (LSM 800, Zeiss).

## Immunoprecipitation

Cells were lysed in cell lysis buffer (P0013, Beyotime Biotechnology) with protease/phosphatase inhibitor cocktail. For the interaction of RIPK1 and RIPK3, anti-RIPK1 antibody (1:200) and control IgG antibody were added separately to each aliquot, and samples were rotated with protein A/G agarose beads (Sc-2003, Santa Cruz) at 4 °C overnight. The complex was washed for 3 times using the same lysis buffer and then subjected to immunoblotting.

## TUNEL staining

TUNEL staining was performed using paraffin-embedded tissues as mentioned before. The staining was performed with apoptosis kit (11684817910, Roche) to detect apoptotic cells in colon sections according to the instructions of manufacturer. After the paraffin sections conventional dewaxing to water, performed the antigen recovery with protease K working solution at 37°C for 22 min. Then incubated the sections with permeabilize working solution at room temperature for 20 min. Next, take appropriate amount of TDT enzyme, dUTP and buffer in the tunel kit according to the number of slices and tissue size and mix at 1:5:50 ratio, and used to incubate the tissues at 37°C for 1 h. Finally, the DAPI dye (S2110, Solarbio) was used to stain nucleus at room temperature for 10 min. Images were captured with NanoZoomer S60 (C13210-01, Hamamatsu).

## Cellular thermal-shift assay

CETSA was performed as described previously reported. P2Y$_{14}$-HEK293 cells were divided into 2 groups, one group hatched with 10 µM **HDL-16**, and the control group added DMSO of the same volume, and two groups of cells were collected after 30 min. Cells were loaded into a PCR tube and heated with 6 temperature gradients (48, 50, 52, 54, 56, and 58 °C) for 10 min. Then the samples were resuspended by ice-cold PBS with 0.4% NP-40 PBS and repeatedly frozen and melted for twice[41,42]. Then the samples were centrifuged (10000 g) at 4°C for 15 min. The supernatants were boiled with 1/4 times volume of 5×loading buffer for 5 min, and then perform a western blot analysis.

## Bioinformatics analysis

The expression level of *P2ry14* in intestinal mucosal tissues from IBD patients and healthy controls were obtained from expression profiling array of GEO dataset GSE38713, GSE75214, GSE16879, GSE117993, GSE126124 in GEO database (www.ncbi.nlm.nih.gov/gds). A GEO2R online program (www.ncbi.nlm.nih.gov/geo/geo2r/) was applied to detect differentially expressed genes between normal control, UC inflamed, and UC uninflamed tissues.

## Statistics and reproducibility

GraphPad Prism V.8 was used for statistical analysis. Data are presented as mean ± SD or SEM. Tests used include One-way ANOVA with Tukey multiple comparison test, two-way ANOVA with Sidak's multiple comparisons test, and Unpaired T test. $P < 0.05$ was considered

statistically significant. All data are representative of at least three independent experiments. The box in the box plot indicates the upper and lower quartiles, with the line inside the box indicating the median. The whiskers extending from the box represent the range of data, where the lower whisker reaches the minimum, and the upper whisker extends to the maximum.

## Reporting summary

Further information on research design is available in the Nature Portfolio Reporting Summary linked to this article.

## Data availability

All data generated or analyzed during this study are included in this published article (and its supplementary information files). Source data are provided with this paper. All the datasets in this study are existing published and are available via the NCBI website, including Gene Expression Omnibus (GSE) accession number: GSE38713, GSE75214, GSE16879, GSE117993, GSE126124. Any additional information is available upon request to the corresponding author (Qinghua Hu, huqh@cpu.edu.cn). Source data are provided with this paper.

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

## Acknowledgements

We gratefully thank Dr. Qianming Du of Nanjing First Hospital for supporting the clinical samples. This work was supported by the National Key Research and Development Program of China (2023YFC2812500 to Q.H.), the National Natural Science Foundation of China (No. 82373887 to Q.H., No. 82373725 to H.L.), the Natural Science Foundation of Jiangsu Province (No. BK20211223 to Q.H.) and the Priority Academic Program Development of the Jiangsu Higher Education Institutes (PAPD to H.L.).

## Author contributions

Q.H. did the concept and study design. C.L. performed the experiments. H.W. and Y.Z. did the molecular docking-based Virtual Screening. L.H. and S.N. performed the revision work. X.Y. and Jing-ke.Z. performed the pharmacodynamic experimental design for HDL-16. Jiayi.Z. did the statistical analysis and interpreted the results. H.L. and S.T. provided essential material and conceptual advice. C.L. and H.W. drafted the manuscript. Q.H. and H.L. revised the manuscript. All authors reviewed and commented on the manuscript and approved its final submission.

## Competing interests

The authors declare no competing interests.
