## [Peer Review File · Nature Communications]

REVIEWER COMMENTS

Reviewer #1 (Remarks to the Author):

In their manuscript "Targeting P2Y14R protects against necroptosis of intestinal epithelial cells through PKA/CREB/RIPK1 axis in ulcerative colitis", Liu et al. show a previously unknown function of the purinoceptor P2Y14R in regulating IEC necroptosis. Although the experiments describing meta-analysis of IBD cohorts for the receptor expression, mouse colitis experiments, and the screening and optimization of the receptor antagonist HDL16 are largely convincing, the evidence presented for the inhibition of necroptosis and its mechanism needs improvement.

Specific comments:

1. All findings regarding necroptosis regulation are performed in one cell line (HT29). The authors need to recapitulate some of their key findings in a more physiological epithelial model, like epithelial organoids generated from control and P2Y14R deficient mice.
2. Conceptually it is difficult to realize that the proposed mechanism behind RIPK1 repression at transcript level can take effect in the short time frame of the TSz experiments in cells shown by the authors. The mechanism needs to be proven more directly by measurement of RIPK1 protein and RNA levels in multiple experiments to directly show that it is reduced when P2RY14 is either inhibited, deleted or repressed via siRNA. In addition, protein half-life experiments for RIPK1 should be performed after inhibition of protein synthesis to show how long the residual RIPK1 persists in the cells. Please also provide qPCR results of RIPK1 in Fig 3, 4 and 5.
3. The rationale of cell death and in particular necroptosis in Fig. 2a is rather weak. No cell death analyses were shown from in vitro experiments. Please provide TUNEL staining and cleaved caspase-3 staining in all in vivo experiments.
4. Overall, the baseline pMLKL levels that the authors report in control cells is of concern.
5. For siRNA related experiments, the authors must provide evidence that a) HT29 express the P2Y14R and b) siRNA and OE treatment reduces/increases the expression levels of P2Y14R. Data can be shown as supplemental information.
6. The authors neglect the implications of their finding that the P2Y14 receptor modulates necroptosis signaling even in the absence of any addition of the ligand UDP-glucose to the cultures (Figs 2,3,4). Does this mean that sufficient UDP-glucose is already present in these cultures? In that case, levels of UDP-Glucose have to be analyzed in these cultures. The authors need to show what triggers the receptor in the absence of administration of UDP-glucose.
7. For Figure 4, the authors should provide CREB binding to RIPK1 promoter site1 data for mice lacking P2y14r in IEC and provide better western blots for pMLKL in Fig 4h. Moreover, in 4f, they claim that 90% of siP2Y14R+TSZ cells are viable, while PI staining of these cultures (4g) shows more than half of the cells are actually dead (PI positive).
8. The CREB mediated regulation is the molecular link between P2Y14R and necroptosis in this manuscript but unexpectedly neglected in Fig. 5 and 7. Since CREB binding to the RIPK1 promoter is suggested as a key mechanism of necroptosis regulation by P2Y14R, is

CREB-phosphorylation and promoter binding regulated by UDP-Glucose or antagonist treatment of epithelial cells.

9. The immunohistochemistry for Occludin, ZO-1 and Claudin-1 in figure 7f are of rather low quality and do not allow any interpretation. Show cell death in mice treated with/out the antagonist.

Minor:

1. It's confusing that the results description doesn't always follow the order of the subfigures.

2. Better stainings for Fig. 1d are required. Preferably for epithelial markers like E Cadherin or Epcam.

3. There is an obvious error in the labelling in Fig 2f.

4. The reduction in MLKL phosphorylation shown in Fig 2g. are not representative to the quantification. Please show all pMLKL blots used for analysis as supplement.

5. In the HT29 experiments showing TSz induced pMLKL levels, how can the authors explain pMLKL already at baseline (control cells).

6. The Fig 3 does not seemingly add any value to the main findings. It just verifies the well-established paradigm of RIP1 RIP3 involvement in necroptosis. Therefore, Fig3 can be moved to the supplement. The authors should however discuss in a brief statement that the P2Y14R mediated effect on necroptosis involves the canonical RIP1, RIP3 signaling mechanism to activate MLKL.

7. How can the authors justify only modest increase in PI in the control TSz group in Fig. 5c? TSz treatment is a powerful killer of HT29 cells.

8. Figure legend in 5b incorrectly states P2Y14R expression levels, when in fact these are Ugp2 expression levels.

9. Gene and protein nomenclatures do not follow standard nomenclature.

10. Line 36: Avoid using the term UC for experimental colitis in mice.

11. Line 53: Necroptosis has been discovered more than 10 years ago and not newly.

12. Line 109: colon shortening

13. Line 221: 84 common TFs, Figure shows 88

14. Professional help with English language editing is required.

Reviewer #2 (Remarks to the Author):

In this manuscript, the authors focused on the physiological and pathological functions of P2Y14R in the intestinal epithelium. The epithelial cell-specific deficiency of P2Y14R showed severer colitis induced by DSS treatment compared to control mice. More

specifically, the authors showed suppressive roles of P2Y14R in epithelial necroptosis through the RIPK pathway during inflammation. Furthermore, UDP-glucose which acts as the ligand of P2Y14 potentially involved in the necroptosis. Finally, a novel P2Y14R antagonist DL-16 was discovered which specifically bound the receptor and inhibited the onset of inflammation in DSS treated mice. The authors provided promising results supporting that the P2Y14R targeting therapy could be one of the unique methods for the control of UC.

The authors provide a novel aspect of our understanding on the P2Y14R mediated intestinal epithelial necroptosis during inflammation (e.g., ulcerative colitis: UC) and a potential target of P2Y14R for the control of colitis. The manuscript consisted of a large volume of data from basic (e.g., P2Y14R mediated signaling cascade) to clinical application (e.g., discovery of the receptor antagonist DL-16). Because of variety of the interesting data, the current form of the study suffers because of the lack of focus. Thus, it was difficult to figure out what is a major emphasis of their study. Are they focusing on the molecular and cellular physiological and pathological understanding of the P2Y14R mediated signaling in epithelial cell necroptosis and its involvement in UC? Alternatively, are they highlighting the discovery of the receptor antagonist DL-16 and its possible clinical application for the control of UC? The manuscript can be separated into two studies with a clearer message of the interesting research of the authors on the basic and clinical aspects of P2Y14R-mediated epithelial cell necroptosis.

It is also known that in addition to epithelial cells, P2Y14R is expressed in various cells of the mucosa, including immune cells, mesenchymal cells, and neural cells in the gut. Although this point was also raised by the authors (e.g., Introduction) in their manuscript, the current form of the study did not address or discuss another possibility of the role of the P2Y14R-mediated necroptosis regulation system on gut immune cells, mesenchymal cells, and /or neural cells. Considering the nature of inflammation, it is important address whether P2Y14R positive immune cells, mesenchymal cells and/or neural cells are involved in the UC inflammation or not? Since the authors are focusing on the P2Y14R epithelial cell mediated inflammation in UC, it would be useful to provide additional evidence supporting their view. What is the definitive evidence for the specific association of the P2Y14R epithelial cell mediated inflammation with UC but not with other inflammatory diseases?

Specific Comments

1. The author predicted that P2Y14R in the intestinal epithelium plays a central role in the control of inflammation in UC based on the previous report indicating the association of the inflammatory score and the expressions of P2Y14R in the mucosa. However, P2Y14R expresses in the various cell types including macrophage, mesenchymal cells and neutrophils and these cells are also involved in the development of intestinal inflammation and considered as initial targets of IBD therapy. These important points should not be ignored and should experimentally addressed and discussed.

2. It is critical to include the detailed data showing the molecular cellular mechanisms of upregulation of P2Y14R expression on colonic epithelial cells. How is the receptor expression elevated in UC epithelium? What is molecular and cellular mechanisms for the receptor induction?

3. In related to above comment, the authors emphasized the correlation between the P2Y14R expression and inflamed epithelial cells in UC. Does only the colon epithelium enhance the expression of P2Y14R in the inflammatory condition? The expression of P2Y14R in the colon mucosa of patients with Crohn's disease (CD) should be compared with UC. In addition to the data generated by the public depository (e.g., NCBI's Gene Expression Omnibus: GSE38713, GSE75214, GSE6879), this issue needs to be directly demonstrated by the related tissue specimens from different forms of inflammation.

4. The authors somehow did not provide any data related to the phenotypical assessment of IEC-P2Y14R deficient mice (P2Y14R Δ IEC) in the steady state. Is there a difference in epithelial cell integrity? It has been shown that goblet cells express functional P2Y14R. It is also important to study whether the P2Y14R Δ IEC influence on goblets cells or not? If so, P2Y14 deficiency could alter commensal bacteria and mucin secretion, which might influence the outcome of intestinal inflammation.

5. The authors emphasized the critical role of UDP-G-P2Y14R mediated signaling cascade for the epithelia cell necroptosis and inflammation. It is useful to provide additional data related to how does the UDP-G involve in the epithelial integrity or necroptosis during inflammation?

6. The author showed the expressions of Ugp2 and Gys1 in the mucosal compartment from the public data base and concluded the accumulation of UDP-G in patients with UC. This is very important point of the study and thus the authors should provide additional in vivo and/or in vitro data supporting their claim. In related to the issue, UDP- glucose is synthesized by UDP glucose pyrophosphorylase 2 (Ugp2) and then degraded by glycogen synthase 1 (Gys1) (Fig. 5a). As the authors recognized this conflicting result, this is an important part of the present study, however the authors somehow did not directly address and discuss the interpretation of simultaneous elevation of Ugp2 and Gys1 in intestinal inflammation.

7. The discovery of the novel inhibitor of P2Y14R "DL-16" is one of the strong advances of this manuscript. However, the current study is lacking the detailed pharmacological study including the detailed experimental protocols (e.g., concentration, administration route and frequency and the vehicle control). Administration of the inhibitor candidate to IEC P2Y14R deficient mice would strengthen the author's conclusion.

8. The authors stated in their Introduction, "An epidemiological study purine gene dysregulation profiles in IBD showed that 59% of purine genes was dysregulation in IBD, but only the expression of P2Y14R was positively correlated with acute inflammatory score in UC mucosal biopsies." (lines 66-69). Based on this remark, it is important to elucidate and compare the P2Y14R-mediated necroptosis regulation in the acute and chronic phases of the DSS induced inflammation using wild type and their unique P2Y14R Δ IEC mouse models.

9. The authors demonstrated the elevated P2Y14R expression in UC specimens compared with those in healthy controls based on the public datasets (using 91 datasets from NCBI's Gene Expression Omnibus: GSE38713, GSE75214, GSE6879) (lines 89-91 and Figure 1a). It is useful to include actual clinical and histological data showing the increase in the receptor expression in the human colon epithelium of UC but not CD in addition to the murine data.

10. To demonstrate the role of the P2Y14R-mediated signaling cascade for necroptosis, the authors used HT-29 cells, a human intestinal epithelial cell line. Although HT-29 cells have been used extensively in in vitro study, unfortunately it is a carcinoma-derived cell line that may not reflect the actual situation in vivo. However, recent progress using human iPS cells and/or tissue-derived organoid and epithelial cells is now available, and thus the authors should consider the use of more physiologically relevant in vitro models.

11. The authors extensively investigated the molecular aspect of the P2Y14R mediated signaling cascade using the HT-29 in vitro system and indicated that P2Y14R regulated necroptosis in IECs at least in part by altering the activation of the cAMP / PKA / CREB pathway and further mediating the transcript of RIPK (lines 246-248). However, it was not clear how these in vitro data directly related and/or contributed to our understanding of pathological aspects of inflammation mediated by the P2Y14R signaling cascade in the intestinal epithelium.

12. In terms of role of the UDP-glucose-P2Y14R mediated promotion of necroptosis of IECs, the data were provided by the HT-29 in vitro system. Since the authors have established a unique in vivo model of the P2Y14R Δ IEC mouse DSS model, it would be beneficial to adopt the system and compared to wild-type mice for a greater understanding of the UDP-glucose-P2Y14R mediated regulation of inflammation in vivo.

13. The authors' efforts on the discovery of P2Y14R antagonist (e.g., DL-16) with their potent binding affinity to the receptor and their ability to inhibit necroptosis of epithelial cells both in vivo and in vitro must be recognized and congratulated. Based on the amount of and biomedical significance of the data, it might be better to publish as a separate study focusing on the newly identified P2Y14R antagonists with additional pharmacological and clinically applicable data.

14. Related to the discovery of a P2Y14R antagonist (e.g., DL-16), it is critical to elucidate whether the antagonist specifically controls epithelial cell inflammation associated with UC or other forms of intestinal inflammation? Is it specific for inflammation that occurs in colon epithelial cells? Further, it is also important to demonstrate whether the antagonist influence on other inflammatory cells such as macrophages, mesenchymal cells and neutrophils in UC and/or other forms of intestinal inflammation, since the authors' group have previously shown the role of P2Y14R in regulating caspase-1-mediated pyroptosis in macrophages and NETosis of neutrophils.

15. The authors emphasized the increased expression of P2Y14R in the intestinal mucosa of UC patients but not CD patients. An obvious question is "why is such a specificity of P2Y14R associated with UC but not with CD inflammation?", however, this important point has been

neglected and seems to be a critical issue for the clinical application of the newly discovered P2Y14R antagonist DL-16.

Minor

16. Actual data from the evaluation of intestinal permeability assessment (e.g., FITC assay) should be shown in addition to the pictures of the mice to support the important findings of the authors (Figure 1).

17. The reference(s) should be cited for the sentence 'Previous studies have found increased necroptosis in the colon tissues of patients with CD and UC' in the Introduction (lines 56-57), as it is critically related to the scope of this study.

18. For introducing one of the authors' rationales of the study, they have indicated that "IECs erosion and colitis induced with dextran sulfate sodium (DSS) are attenuated by knocking out MLKL, implicating IECs necroptosis as a key constituent of experimental colitis. Especially, necroptosis also actively participated in the inflammatory response by promoting cascade reactions through the leakage of cellular contents from damaged plasma membranes." (lines 57-62). It will be more persuasive if the authors can include its related human IBD cases.

Reviewer #3 (Remarks to the Author):

In the manuscript entitled "Targeting P2Y14R protects against necroptosis of intestinal epithelial cells through PKA/CREB/RIPK1 axis in ulcerative colitis", the authors present a finding that P2Y14R deletion in UC can slow down intestinal damage. Then they investigated the mechanism of action and designed a new P2Y14R antagonist with significant anti-UC effect. The novelty is good.

The reviewer asks to address the following issues:

(1) In Figure 1d, the histology suggests that no good care was taken as to where along the colon the samples were collected, which is very important when interpreting DSS colitis. The samples showing the control mice clearly show proximal colon. In contrast the DSS treated sample shows severe inflammation in the distal colon. As in the DSS-induced colitis model colitis is usually severe in the distal colon while much less activity occurs in the proximal colon, the authors must present samples from the distal colon.

(2) This manuscript shows that P2Y14R may mediate DSS-induced colitis by affecting the death of IECs, and further studies have only clarified that it is not associated with apoptosis. The phenomena of cell membrane rupture and cell swelling also appeared when

the cell pyrosis. In addition, studies have shown that pyroptosis is associated with colitis, and P2Y14R is involved in the regulation of pyroptosis. The authors need to show whether there is a relationship between the effect of P2Y14R and pyroptosis in colitis.

(3) There is a problem with Figure 2f, which is inconsistent with the description in the manuscript.

(4) There was no evidence that P2Y14R was knocked down or overexpressed in HT-29 cells.

(5) In Figure 7f-h, normal HT-29 cells need to be added as controls.

(6) This manuscript suggests that P2Y14R prevents intestinal epithelial cell necrosis through the PKA/CREB/RIPK1 axis, but this has not been fully elucidated in experiments.

a) When P2Y14R endogenous ligand UDP-glucose acts on HT-29 cells, do the contents of PKA, CREB and RIPK1 and their phosphorylation levels change?

b) When treating DSS-induced colitis with P2Y14R antagonist HDL-16, do the contents of PKA, CREB and RIPK1 and their phosphorylation levels change in colon?

(7) The P2Y14R antagonist HDL-16 in this manuscript has excellent activity, but there are too few data to characterize the activity, only two methods (test cAMP production and CETSA) are used. GPCR is a kind of complex target, which needs more support of activity test data. In addition, P2Y14R is a membrane protein, which does not seem to be suitable for direct testing by CETSA method. Whether special treatment methods have been adopted? The paper does not describe any methods for testing HDL-16 activity in vitro.

(8) The proposal is correct and the results are well-founded. There are some fragments of the Discussion that seem more like a summary. I suggest that the Discussion be rewritten to better reflect the quality of the work. The statement and description of future works are needed.

Reviewer #1 (Remarks to the Author):

Dear Reviewers,

Thank you very much for your time involved in reviewing the manuscript and your very encouraging comments on the merits.

In their manuscript “Targeting P2Y₁₄R protects against necroptosis of intestinal epithelial cells through PKA/CREB/RIPK1 axis in ulcerative colitis”, Liu et al. show a previously unknown function of the purinoceptor P2Y₁₄R in regulating IEC necroptosis. Although the experiments describing meta-analysis of IBD cohorts for the receptor expression, mouse colitis experiments, and the screening and optimization of the receptor antagonist HDL16 are largely convincing, the evidence presented for the inhibition of necroptosis and its mechanism needs improvement.

We also appreciate your clear and detailed feedback and hope that the explanation has fully addressed all of your concerns. In the remainder of this letter, we discuss each of your comments individually along with our corresponding responses.

To facilitate this discussion, we first retype your comments and then present our responses to the comments.

Specific comments:

1. All findings regarding necroptosis regulation are performed in one cell line (HT29). The authors need to recapitulate some of their key findings in a more physiological epithelial model, like epithelial organoids generated from control and P2Y₁₄R deficient mice.

Response: We appreciate your valid suggestion and have supplemented intestinal epithelial organoids experiments and HCT-116 cells to further verify our findings in HT-29 cells, the detailed results are summarized as follows.

First, we generated intestinal epithelial organoids from P2Y₁₄R^{fl/fl} or P2Y₁₄R^{ΔIEC} mice and then stimulated them with TSZ for 12 hours (Patankar JV, et al. *Nat Cell Biol.* 2021). We found that, similar to the results in HT-29 cells, P2Y₁₄R deficient significantly inhibited TSZ-induced necroptosis in P2Y₁₄R^{ΔIEC} organoids (**Fig. #1a**). Following, we pre-treated P2Y₁₄R^{ΔIEC} organoids with SQ22536, H-89 or 666-15 and then induced necroptosis. As expected, inhibition of cAMP/PKA/CREB pathway disrupted the protective effect of P2Y₁₄R deficiency in TSZ mediated necroptosis (**Fig. #1b**). Furthermore, we found UDPG treatment could increase TSZ induced necroptosis in organoids from P2Y₁₄R^{fl/fl} mice, while administration with Nec-1s and GSK’872

remarkably attenuated the UDP-glucose derived damage on TSZ-stimulated necroptosis (**Fig. #1c**). More importantly, administration with P2Y₁₄R inhibitor **HDL-16** also protected P2Y₁₄R^{fl/fl} organoids from TSZ induced necroptosis (**Fig. #1d**). These results were all included in the **revised Fig 2k, Fig 4l, Fig 5j and Fig 7m**. Related methods have been included in the method part of the revised manuscript.

Fig.#1 a. Intestinal organoids from P2Y₁₄R^{fl/fl} or P2Y₁₄R^{ΔIEC} mice treated as indicated with DMSO (veh), or TNF- α adding Smac mimetic and z-VAD (TSZ) for 12 h and stained with PI (red). **b.** The PI staining of intestinal organoids from P2Y₁₄R^{ΔIEC} mice treated as indicated with DMSO (veh), TSZ, TSZ adding SQ22536, TSZ adding H-89 or TSZ adding 666-15. **c.** The PI staining of intestinal organoids from P2Y₁₄R^{fl/fl} mice treated as indicated with DMSO (veh), TSZ, TSZ+UDPG, TSZ+UDPG+Nec-1s or TSZ+UDPG+GSK'872. **d.** The PI staining of intestinal organoids from P2Y₁₄R^{fl/fl} mice treated as indicated with DMSO (veh), TSZ, TSZ + HDL-16 or TSZ+PPTN.

Next, we repeated our *in vitro* experiments in HCT-116 cells. Similar with the result in HT-29 cells, P2Y₁₄R silencing inhibit TSZ-induced necroptosis according to the results of PI staining and western blot results of p-MLKL (**Fig. #2a-b**). The lack of P2Y₁₄R in HCT-116 cells resulted in a decrease in the expression and mRNA level of *Ripk1*, while the overexpression of P2Y₁₄R led to an upregulation of *Ripk1* under TSZ condition, indicating that P2Y₁₄R participate in TSZ-induced necroptosis through influence the transcript of RIPK1 (**Fig. #2c-d**). The detection of intracellular cAMP, p-PKA and p-CREB indicating that cAMP/PKA/CREB pathway could involve in the regulation of P2Y₁₄R on TSZ-induced necroptosis of HCT-116 (**Fig. #2e-f**). Then, we pre-treatment P2Y₁₄R silencing HCT-116 cells with SQ22536, H-89 and 666-15 to further confirm that P2Y₁₄R

regulated TSZ-induced necroptosis relying on cAMP/PKA/CREB pathway. The results showed that, pre-treatment with SQ22536, H-89 and 666-15 in HCT-116 cells disrupted the protective effect on TSZ-induced necroptosis and suppressed the transcript of *Ripk1* (Fig. #2g-i). Finally, we pre-administrated HCT-116 cells with UDPG for 12 h and found that UDPG exacerbated TSZ-induced necroptosis according to the upregulation of PI positive cell and p-MLKL level. The necroptotic cell death was further confirmed using Nec-1s and GSK'872, where Nec-1s and GSK'872 treatment remarkably attenuated the UDP-glucose derived damage on TSZ-stimulated necroptosis (Fig. #2j-k). Besides, UDPG treatment aggravated the inhibition of TSZ on cAMP/PKA/CREB pathway according to the level of p-PKA, p-CREB, the mRNA level of *Ripk1* as well as the binding between CREB and *Ripk1* promoter (Fig. #2l-n). These results further demonstrated that UDP-glucose partly contribute to the activation of the P2Y₁₄R in regulating IECs necroptosis in IBD. These results were all included in the revised Fig 2i-j, Fig 3b and d, Fig 4c-d and i-k, Fig 5h-i and n-p. Related methods have been included in the method part of the revised manuscript. The primer sequence has been included in the supplementary file of the revised manuscript.

Fig.#2 a. PI/Hoechst staining were used to analyzed PI positive cells in HCT-116 cells treated as indicated with veh or TSZ. **b.** Phosphorylation of MLKL as well as its protein levels were analyzed

by immunoblotting in HCT-116 cells. **c.** The expression of *Ripk1* were analyzed by immunoblotting in HCT-116 cells. **d.** The mRNA level of *Ripk1* in siP2Y₁₄R or P2Y₁₄R-OE HCT-116 cells treated with Veh or TSZ was analyzed by RT-PCR. **e.** The intracellular cAMP level in HCT-116 cells. **f.** Phosphorylation of PKA and CREB as well as their protein levels were analyzed by immunoblotting in HCT-116 cells. **g.** PI/Hoechst staining were used to analyzed PI positive cells in siP2Y₁₄R HCT-116 cells treated as indicated. **h.** Phosphorylation of MLKL as well as its protein levels were analyzed by immunoblotting in siP2Y₁₄R HCT-116 cells. **i.** The mRNA level of *Ripk1* in siP2Y₁₄R HCT-116 cells treated with Veh, TSZ, TSZ+SQ22536, TSZ+H-89, TSZ+666-15 was analyzed by RT-PCR. **j.** PI/Hoechst staining were used to analyzed PI positive cells in HCT-116 cells treated as indicated. **k.** Phosphorylation of MLKL as well as its protein levels were analyzed by immunoblotting in HCT-116 cells. **l.** Phosphorylation of PKA and CREB as well as their protein levels were analyzed by immunoblotting in HCT-116 cells. **m.** The mRNA level of *Ripk1* in HCT-116 cells treated with Veh, TSZ, UDPG, UDPG+TSZ, UDPG+TSZ+Nec-1, UDPG+TSZ+GSK'872 was analyzed by RT-PCR. **n.** CHIP with anti-CREB of the regions containing the CREB binding sites on the RIPK1 gene promoter in HCT-116 cells treated with Veh, TSZ, UDPG, UDPG+TSZ, UDPG+TSZ+Nec-1, UDPG+TSZ+GSK'872. The data represent the mean ± SD, and statistical significance was determined by two-way analysis of variance (ANOVA) with Sidak's multiple comparisons test for **c, f and l**, and One-way analysis of variance (ANOVA) with Tukey multiple comparison test for **a-b, d-e, g-k, m-n**. **P* < 0.05, ***P* < 0.01, and ****P* < 0.001.

2. Conceptually it is difficult to realize that the proposed mechanism behind RIPK1 repression at transcript level can take effect in the short time frame of the TSz experiments in cells shown by the authors. The mechanism needs to be proven more directly by measurement of RIPK1 protein and RNA levels in multiple experiments to directly show that it is reduced when P2RY14 is either inhibited, deleted or repressed via siRNA. In addition, protein half-life experiments for RIPK1 should be performed after inhibition of protein synthesis to show how long the residual RIPK1 persists in the cells. Please also provide qPCR results of RIPK1 in Fig 3, 4 and 5.

Response: We appreciate your valuable feedback and constructive suggestions. To further prove that P2Y₁₄R involved in TSZ-induced necroptosis through influence the transcript of *Ripk1*, we detected mRNA level of *Ripk1* in HT-29 and HCT-116 cells. According to the results, the mRNA level of *Ripk1* significantly decreased in P2Y₁₄R silenced cells, while overexpression of P2Y₁₄R led to an increase in *Ripk1* transcription under TSZ condition. (**Fig.#3 a-b**). Further, inhibition cAMP/PKA/CREB pathway with SQ22536, H-89 or 666-15 greatly up-regulated the level of *Ripk1*

mRNA in siP2Y₁₄R cells (Fig.#3 c-d.). Besides, UDPG treatment promoted the transcript of *Ripk1*, while pre-treatment with Nec-1s or GSK'872 showed no effect in UDPG treated cells (Fig.#3 e-f.). These results were all included in the revised Fig 3 c-d, Fig 4 h, k and Fig 5 l, o. The method of Tunal staining and primer sequence have been included in the method section and Supplementary table 1 of the revised manuscript.

Fig.#3 a. The mRNA level of *Ripk1* in siP2Y₁₄R or P2Y₁₄R-OE HT-29 cells treated DMSO (Veh), or TSZ was analyzed by RT-PCR. **b.** The mRNA level of *Ripk1* in siP2Y₁₄R or P2Y₁₄R-OE HCT-116 cells treated with Veh or TSZ was analyzed by RT-PCR. **c.** The mRNA level of *Ripk1* in siP2Y₁₄R HT-29 cells treated with Veh, TSZ, TSZ+SQ22536, TSZ + H-89 or TSZ + 666-15 was analyzed by RT-PCR. **d.** The mRNA level of *Ripk1* in siP2Y₁₄R HCT-116 cells treated with Veh, TSZ, TSZ+SQ22536, TSZ + H-89 or TSZ + 666-15 was analyzed by RT-PCR. **e.** The mRNA level of *Ripk1* in HT-29 cells treated with Veh, TSZ, UDPG, TSZ+UDPG, TSZ+UDPG+Nec-1 or TSZ+UDPG+GSK'872 was analyzed by RT-PCR. **f.** The mRNA level of *Ripk1* in HCT-116 cells treated with Veh, TSZ, UDPG, TSZ+UDPG, TSZ+UDPG+Nec-1 or TSZ+UDPG+GSK'872 was analyzed by RT-PCR. The data represent the mean ± SD, and One-way analysis of variance (ANOVA) with Tukey multiple comparison test. **P* < 0.05, ***P* < 0.01, and ****P* < 0.001.

In addition, we performed protein half-life experiments for RIPK1, the results showed that treatment of HT-29 cells with cycloheximide (CHX) resulted in the rapid decrease (within 6 hours) of approximately 60-70% of the initial RIPK1 level. Although, a sizable fraction of the initial RIPK1

level remained even after 24 hours of CHX treatment, regulation of transcript level was enough to influence the downstream pathway within 6 hours (**Fig.#4**).

Fig.#4 The expression of RIPK1 in HT-29 cells 0 hours, 1.5 hours, 3 hours, 6 hours, 12 hours and 24 hours after cycloheximide (CHX) treatment was analyzed by western bolt. The data represent the mean \pm SD.

3. The rational of cell death and in particular necroptosis in Fig. 2a is rather weak. No cell death analyses were shown from in vitro experiments. Please provide TUNEL staining and cleaved caspase-3 staining in all in vivo experiments.

Response: We appreciate your valuable feedback and constructive suggestions. In our revised manuscript, we added the TUNEL staining and cleaved caspase-3 staining results to further investigate the relationship between P2Y₁₄R and apoptosis in IECs. As expected, deficiency of P2Y₁₄R in IEC showed less influence on TUNEL positive cell number and cleaved caspase-3 rate in DSS-induced experimental colitis (**Fig.#5a**). Meanwhile, administration of **HDL-16** improved DSS-induced IECs necroptosis according to the result of TEM but showed less improvement on apoptosis in the IECs of experimental colitis mice (**Fig.#5c**). We also supplement the western blot analysis of Bax, Bcl-2, p-MLKL, GSDMD and cleaved caspase-1 to investigate the role of IEC P2Y₁₄R on pyroptosis and found that deficiency of P2Y₁₄R in IEC or administration of **HDL-16** did not influence pyroptosis under DSS condition (**Fig.#5b and d**). We incorporated these section in the **revised Fig 2b and c** and **Supplement Figure 4a** of the revised manuscript. The method of TUNEL staining and the information of primary antibodies have been included in the method section of the revised manuscript.

Fig.#5 a. The TUNEL staining of colon tissues from $P2Y_{14}R^{fl/fl}$ and $P2Y_{14}R^{\Delta IEC}$ mice after DSS treatment. The immunofluorescent images of colon tissues stained with cleave caspase-3 (Scale bar = 200 μ m). **b.** The expression of Bcl-2, Bax, MLKL and phosphorylation of MLKL were analyzed by immunoblotting with corresponding antibodies in the IECs from $P2Y_{14}R^{fl/fl}$ and $P2Y_{14}R^{\Delta IEC}$ mice after DSS treatment. **c.** The TEM images showed the typical characteristics of necroptosis in the colon tissues of DSS-treated mice with **HDL-16** or **PPTN** administration. The TUNEL staining of colon tissues from mice with **HDL-16** or **PPTN** administration. The immunofluorescent images of colon tissues stained with cleave caspase-3 (Scale bar = 200 μ m). **d.** The expression of Bax, Bcl-2, GSDMD as well as caspase-1 p20 were analyzed by immunoblotting with corresponding antibodies in the IECs from DSS treated mice with **HDL-16** or **PPTN** administration. The data represent the mean \pm SD, and statistical significance was determined by two-way analysis of variance (ANOVA) with Sidak's multiple comparisons test. * $P < 0.05$, ** $P < 0.01$, and *** $P < 0.001$.

4. Overall, the baseline pMLKL levels that the authors report in control cells is of concern.

Response: We are very grateful to you for pointing out this problem. To clarify this problem, we repeated this experiment several times, but the baseline p-MLKL was still in a high level. Then, we changed the primary antibody of p-MLKL and compared with the previous blots. We found that the level of p-MLKL in control cells significantly dropped after the primary antibody was changed. Thus, we re-tested and switched all the p-MLKL and its corresponding total MLKL appeared in this manuscript (**Fig.#6**). Specifically, it included the western blots results in **Fig 2b, h and j, Fig 4g**

and j, Fig 5f and h, Fig 7h, supplement Fig 2e and g, and supplement Fig 4i of the revised manuscript. It's worth noting that this change would not affect our conclusion. All the original images of western blots have been submitted in a supplement file named "raw data". Finally, we sincerely thank the reviewer for helping us to improve our research.

Fig.#6 All the p-MLKL and its corresponding total MLKL appeared in this manuscript

5. For siRNA related experiments, the authors must provide evidence that a) HT29 express the P2Y14R and b) siRNA and OE treatment reduces/increases the expression levels of P2Y14R. Data can be shown as supplemental information.

Response: We greatly thank you for catching this lack of information. We have incorporated additional results into the **supplement Figure 7** showing the reduces and increases levels of P2Y₁₄R in HT-29 cells and HCT-116 cells after siRNA or over-expression plasmid treatment (**Fig.#7**).

Fig.#7 The expression of P2Y₁₄R in HT-29 cells and HCT-116 cells after siRNA or over-expression plasmid treatment. The data represent the mean \pm SD, and statistical significance was determined by One-way ANOVA with Tukey multiple comparison test. * $P < 0.05$, ** $P < 0.01$, and *** $P < 0.001$.

6. The authors neglect the implications of their finding that the P2Y₁₄ receptor modulates necroptosis signaling even in the absence of any addition of the ligand UDP-glucose to the cultures (Figs 2,3,4). Does this mean that sufficient UDP-glucose is already present in these cultures? In that case, levels of UDP-Glucose have to be analyzed in these cultures. The authors need to show what triggers the receptor in the absence of administration of UDP-glucose.

Response: Thank you for your valuable comment. It is well known that one key feature of cell death is the release of damage-associated molecular patterns (DAMPs). Mederacke I et al has reported that, in liver fibrosis, dying cell released UDPG promoted HSC activation in a P2Y₁₄-dependent manner (Mederacke I et al., *Sci Transl Med.* 2022). The increased UDPG in the urine of AKI patients activates the P2Y₁₄R expressed by intercalated cells, thereby promoting the initial inflammatory steps of AKI (Battistone MA, et al. *J Clin Invest.* 2020). Meanwhile, several *in vitro* investigations showed that UDPG could be released by macrophage, neutrophil and astrocyte under stress condition (Ma J, et al. *Nat Commun.* 2020; Kreda SM, et al. *Br J Pharmacol.* 2008; Sadiku P, et al. *Cell Metab.* 2021). Thus, we supposed that UDPG could be exist in the culture environment of TSZ-stimulated HT-29 cells. To examine this hypothesis, the UDPG level in supernatant of TSZ-treated HT-29 cells was measured by the UDPG Detection Kit (RY-11791, Shanghai RunYu Biotech; Ma J, et al. *Nat Commun.* 2020). As the results showed, the level of UDPG in the supernatant culture medium of HT-29 cells was dramatically increased by TSZ stimulation (**Fig.#8**), suggesting that under TSZ condition, P2Y₁₄R was triggered by UDP-glucose generated by HT-29 cells.

Fig.#8 HT-29 cells were simulated with TSZ for 6 h, UDPG in supernatants was determined by UDPG Detection Kit (n = 4). The data represent the mean ± SD, and statistical significance was determined by Unpaired T test. **P* < 0.05, ***P* < 0.01, and ****P* < 0.001.

7. For Figure 4, the authors should provide CREB binding to RIPK1 promoter site1 data for mice lacking P2y14r in IEC and provide better western blots for pMLKL in Fig 4h. Moreover, in 4f, they claim that 90% of siP2Y14R+TSZ cells are viable, while PI staining of these cultures (4g) shows more than half of the cells are actually dead (PI positive).

Response: We appreciate you for your valuable suggestion and supplement the binding data between CREB and *Ripk1* promoter site1 through CHIP-PCR analysis. As the results showed, DSS-fed in mice suppressed the binding between CREB and *Ripk1* promoter site1, while deficiency of P2Y₁₄R in IEC promoted the binding between CREB and *Ripk1* promoter (**Fig. #9 a**), this result had been incorporated in **revised Figure 3 h**.

For the western blots of Fig 4h, as we mentioned in the above response, we have re-tested all the p-MLKL and its corresponding total MLKL appeared in this manuscript, hope these new western blot results could meet your requirements (**Fig. #9 b-c**). All the information of primary antibodies has been included in the method section of the revised manuscript.

For Fig 4g, we checked all raw data of PI staining results in the manuscript, re-analyzed the PI positive cells and replaced the less representative PI staining pictures with more representative ones (**Fig. #9 d**). Additionally, we have also included all the original images used for PI positive analysis in the “raw data” file. We sincerely thank you for your helpful feedback, which greatly contributes to the improvement of the rigor of our research.

Fig.#9 a. ChIP with anti-CREB of the regions containing the CREB binding sites on the *Ripk1* gene promoter in DSS-treated P2Y₁₄R^{fl/fl} or P2Y₁₄R^{ΔIEC} intestinal epithelial cell. **b.** Phosphorylation of MLKL as well as its protein levels were analyzed by immunoblotting with corresponding antibodies in HT-29 cells. **c.** Phosphorylation of MLKL as well as its protein levels were analyzed by immunoblotting with corresponding antibodies in HCT-116 cells. **d.** PI/Hoechst staining were used to analyze necroptotic cells in siP2Y₁₄R HCT-116 cells treated with Veh, TSZ, TSZ+SQ22536, TSZ+H-89, TSZ+666-15. The data represent the mean ± SD, and statistical significance was determined by One-way ANOVA with Tukey multiple comparison test. **P* < 0.05, ***P* < 0.01, and ****P* < 0.001.

8. The CREB mediated regulation is the molecular link between P2Y₁₄R and necroptosis in this manuscript but unexpectedly neglected in Fig. 5 and 7. Since CREB binding to the RIPK1 promoter is suggested as a key mechanism of necroptosis regulation by P2Y₁₄R, is CREB-phosphorylation and promoter binding regulated by UDP-Glucose or antagonist treatment of epithelial cells.

Response: We appreciate your constructive suggestions and have incorporated the phosphorylation level of CREB and PKA in revised **Fig 5** and **Fig 7**, as well as detected the binding between CREB and *Ripk1* promoter. For **Fig 5k-p**, pre-treated with UDPG exacerbated TSZ-induced disrupted of PKA/CREB pathway and promoted the transcription of *Ripk1*, yet treatment with Nec-1s and GSK'872 did not affect the phosphorylation level of PKA/CREB pathway or the mRNA level of *Ripk1* (**Fig #10 a-b and d-e**). More important, the results of CHIP-PCR showed a significant decrease of the binding between CREB and *Ripk1* promoter, pre-treatment with Nec-1s and GSK'872 did not affect this process as well (**Fig #10 c and f**). In **Fig 7i-k**, the administration of HDL-16 or PPTN significantly increased the level of p-PKA and p-CREB, as well as the mRNA level of *Ripk1* (**Fig #10 g-h**). The results of CHIP-PCR showed that treatment with HDL-16 or PPTN promoted the binding between CREB and *Ripk1* promoter in the IECs of experimental colitis mice (**Fig #10 i**). These results suggested that P2Y₁₄R involved in DSS-induced inflammation and IEC necroptosis through PKA/CREB/RIPK1 pathway.

Fig.#10 a. Phosphorylation of PKA and CREB as well as their protein levels were analyzed by immunoblotting with corresponding antibodies in HT-29 cells (n = 5). **b.** mRNA level of RIPK1 in HT-29 cells treated with Veh, TSZ, UDPG, UDPG+TSZ, UDPG+TSZ+Nec-1, UDPG+TSZ+GSK'872. **c.** ChIP with anti-CREB of the regions containing the CREB binding sites on the *Ripk1* gene promoter in HT-29 cells treated with Veh, TSZ, UDPG, UDPG+TSZ, UDPG+TSZ+Nec-1, UDPG+TSZ+GSK'872. **d.** Phosphorylation of PKA and CREB as well as their protein levels were analyzed by immunoblotting with corresponding antibodies in HCT-116 cells. **e.** mRNA Level of *Ripk1* in HCT-116 cells treated with Veh, TSZ, UDPG, UDPG+TSZ, UDPG+TSZ+Nec-1, UDPG+TSZ+GSK'872. **f.** ChIP with anti-CREB of the regions containing the CREB binding sites on the RPK1 gene promoter in HT-29 cells treated with Veh, TSZ, UDPG, UDPG+TSZ, UDPG+TSZ+Nec-1, UDPG+TSZ+GSK'872. **g.** Phosphorylation of PKA and CREB as well as their protein levels were analyzed by immunoblotting with corresponding antibodies in HDL-16-L, HDL-H or PPTN administrated experimental colitis mice intestinal epithelial cell. **h.** mRNA level of RPK1 were analyzed by immunoblotting with corresponding antibodies in HDL-16-L, HDL-H or PPTN administrated experimental colitis mice intestinal epithelial cell. **i.** ChIP with anti-CREB of the regions containing the CREB binding sites on the *Ripk1* gene promoter in intestinal epithelial cell of HDL-16-L, HDL-H or PPTN administrated experimental colitis mice. The data represent the mean \pm SD, and statistical significance was determined by two-way analysis of variance (ANOVA) with Sidak's multiple comparisons test for **a, d** and **g**, and One-way analysis of variance (ANOVA) with Tukey multiple comparison test for **b-c, e-f, h-i**. * $P < 0.05$, ** $P < 0.01$, and *** $P < 0.001$.

9. The immunohistochemistry for Occludin, ZO-1 and Claudin-1 in figure 7f are of rather low quality and do not allow any interpretation. Show cell death in mice treated with/out the antagonist. Response: We appreciate your constructive suggestions. According to your suggestion, we repeated the immunohistochemistry for Occludin, ZO-1 and Claudin-1 in figure 7f and detected the results by confocal microscopy (**Fig #11a**). Besides we added the TEM analysis, TUNEL staining, cleaved caspase-3 staining to analysis the influence of HDL-16 on necroptosis and apoptosis in mice experimental colitis model. We also supplement the western blot analysis of the N-terminal of GSDMD (GSDMD-NT) and cleaved caspase-1 (caspase-1 p20) to investigate the effect of HDL-16 on pyroptosis. As the results shown, HDL-16 treatment suppressed DSS-induced necroptosis in IECs according to the results of TEM analysis (**Fig #11b**). On the other hand, the results of TUNEL staining, cleaved caspase-3 staining, and western blot analysis of Caspase-1 p20 and the GSDMD-NT indicated that HDL-16 administration had minimal impact on apoptosis and pyroptosis (**Fig #11c-d**). We have also incorporated these results in **Fig. 7f-g** and **supplement Fig. 4 a-b** of the revised manuscript. The method of TUNEL staining and the information of primary antibodies have been included in the method section of the revised manuscript.

Fig.#11 a. The immunofluorescent images of colon tissues stained with Claudin-1, Occludin and ZO-1, the principal components of tight junction (Scale bar = 200 μm). **b.** The TEM images showed the typical characteristics of necroptosis in the colon tissues of DSS-treated mice with HDL-16 or PPTN administration. **c.** The TUNEL staining of colon tissues from mice with HDL-16 or PPTN administration. The immunofluorescent images of colon tissues stained with cleave caspase-3 (Scale bar = 200 μm). **d.** The expression of Bax, Bcl-2, GSDMD as well as caspase-1 p20 were analyzed by immunoblotting with corresponding antibodies in the IECs from DSS treated mice with HDL-16 or PPTN administration. The data represent the mean ± SD, and statistical significance was

determined by two-way analysis of variance (ANOVA) with Sidak's multiple comparisons test. * $P < 0.05$, ** $P < 0.01$, and *** $P < 0.001$.

Minor:

1. It's confusing that the results description doesn't always follow the order of the subfigures.

Response: Thank you for pointing out this problem for us. We have adjusted the order of the subfigures to ensure that they correspond with the description of the results.

2. Better staining for Fig. 1d are required. Preferably for epithelial markers like E Cadherin or Epcam.

Response: Thank you for your constructive suggestion. In order to better illustrate the improvement of P2Y₁₄R in IECs, we conducted immunofluorescence staining to examine the colocalization of P2Y₁₄R and Epcam in the colon tissues of DSS-fed mice. We have also incorporated these results in **Fig. 1e** of the revised manuscript.

3. There is an obvious error in the labelling in Fig 2f.

Response: Thank you for pointing out this error for us. We have made the necessary correction in the revised manuscript. We appreciate your kind reminder.

4. The reduction in MLKL phosphorylation shown in Fig 2g. are not representative to the quantification. Please show all pMLKL blots used for analysis as supplement.

Response: Thank you for pointing out this problem for us. We have re-test all the p-MLKL and its corresponding total MLKL data presented in this manuscript and have made the necessary corrections. In addition, we have reviewed all the western blot results and their corresponding quantification data to ensure consistency. All original western blot images have been included as a supplement file titled "raw data."

5. In the HT29 experiments showing TSz induced pMLKL levels, how can the authors explain pMLKL already at baseline (control cells).

Response: Thank you for pointing out this problem for us. Through experiments, we confirmed that the abnormal elevation of p-MLKL in untreated HT-29 cells was caused by an unsuitable primary antibody. Therefore, we repeated the tests and replaced all blots of p-MLKL and its corresponding

total MLKL in this manuscript. All original western blot images have been included as a supplement file titled "raw data."

6. The Fig 3 does not seemingly add any value to the main findings. It just verifies the well-established paradigm of RIP1 RIP3 involvement in necroptosis. Therefore, Fig3 can be moved to the supplement. The authors should however discuss in a brief statement that the P2Y₁₄R mediated effect on necroptosis involves the canonical RIP1, RIP3 signaling mechanism to activate MLKL.

Response: Thank you for your valuable suggestion. We have restructured the manuscripts and modified the results of **Fig 3a-d** to **supplement Fig 2**. This change has made our study more focused.

7. How can the authors justify only modest increase in PI in the control TSz group in Fig. 5c? TSz treatment is a powerful killer of HT29 cells.

Response: Thank you for your point out this problem. We have repeated this experiment and replaced the original result with the new results in the **Fig 5g (Fig #11)**. All the original images used for PI positive analysis have been included in the "raw data" file and submitted with the revised manuscript.

Fig #11. PI/Hoechst staining were used to analyzed PI positive cells in siP2Y₁₄R HT-29 cells treated as indicated with Veh, TSZ, TSZ+SQ22536, TSZ+H-89, TSZ+666-15. The data represent the mean \pm SD, and One-way analysis of variance (ANOVA) with Tukey multiple comparison test. * $P < 0.05$, ** $P < 0.01$, and *** $P < 0.001$.

8. Figure legend in 5b incorrectly states P2Y₁₄R expression levels, when in fact these are Ugp2 expression levels.

Response: Thank you for pointing out this error for us. We have corrected this error in the revised manuscript and marked this change in red.

9. Gene and protein nomenclatures do not follow standard nomenclature.

Response: Thank you for your valuable suggestion, which is highly appreciated. We have corrected all the name of gene and protein appeared in the manuscript according to the standard nomenclature, words in red are the changes we have made in the manuscript.

10. Line 36: Avoid using the term UC for experimental colitis in mice.

Response: Thank you for your suggestion. We have modified this point in the manuscript and marked the change in red.

11. Line 53: Necroptosis has been discovered more than 10 years ago and not newly.

Response: Thank you for pointing out this problem. We have changed “a newly defined” into “a kind of” and marked this change in red in the revised manuscript.

12. Line 109: colon shortening

Response: Thank you for pointing out this problem. We have changed “rectum shorten” into “colon shortening” and marked this change in red in the revised manuscript.

13. Line 221: 84 common TFs, Figure shows 88

Response: Thank you for pointing out the problem. I would like to apologize for any confusion caused. We have revised the original sentence from "84" to "88" in the revised manuscript, which has been marked in red. Thank you again for bringing the problem to our attention.

14. Professional help with English language editing is required.

Response: Thank you for your suggestion. We have asked for help from a native English speaker to correct any possible grammar problems in the manuscript. We appreciate your feedback and understand the importance of precise expression.

Reviewer #2 (Remarks to the Author):

Dear Reviewers,

Thank you very much for your time involved in reviewing the manuscript and your very encouraging comments on the merits.

In this manuscript, the authors focused on the physiological and pathological functions of P2Y₁₄R in the intestinal epithelium. The epithelial cell-specific deficiency of P2Y₁₄R showed severer colitis induced by DSS treatment compared to control mice. More specifically, the authors showed suppressive roles of P2Y₁₄R in epithelial necroptosis through the RIPK pathway during inflammation. Furthermore, UDP-glucose which acts as the ligand of P2Y₁₄ potentially involved in the necroptosis. Finally, a novel P2Y₁₄R antagonist DL-16 was discovered which specifically bound the receptor and inhibited the onset of inflammation in DSS treated mice. The authors provided promising results supporting that the P2Y₁₄R targeting therapy could be one of the unique methods for the control of UC.

The authors provide a novel aspect of our understanding on the P2Y₁₄R mediated intestinal epithelial necroptosis during inflammation (e.g., ulcerative colitis: UC) and a potential target of P2Y₁₄R for the control of colitis. The manuscript consisted of a large volume of data from basic (e.g., P2Y₁₄R mediated signaling cascade) to clinical application (e.g., discovery of the receptor antagonist DL-16). Because of variety of the interesting data, the current form of the study suffers because of the lack of focus. Thus, it was difficult to figure out what is a major emphasis of their study. Are they focusing on the molecular and cellular physiological and pathological understanding of the P2Y₁₄R mediated signaling in epithelial cell necroptosis and its involvement in UC? Alternatively, are they highlighting the discovery of the receptor antagonist DL-16 and its possible clinical application for the control of UC? The manuscript can be separated into two studies with a clearer message of the interesting research of the authors on the basic and clinical aspects of P2Y₁₄R-mediated epithelial cell necroptosis.

We appreciate your clear and detailed comments. Our study focused on discovering potential targets for the treatment of UC. In the revised manuscript, we have included several experiments to further support our hypothesis. Our group discovered HDL-16, a novel P2Y₁₄R antagonist with excellent inhibitory activity. Administering HDL-16 in DSS-induced experimental colitis and TSZ-induced necroptosis models not only confirmed the feasibility of P2Y₁₄R as a therapeutic target for

UC, but also provided a potential for the development of UC therapeutic drugs.

It is also known that in addition to epithelial cells, P2Y₁₄R is expressed in various cells of the mucosa, including immune cells, mesenchymal cells, and neural cells in the gut. Although this point was also raised by the authors (e.g., Introduction) in their manuscript, the current form of the study did not address or discuss another possibility of the role of the P2Y₁₄R-mediated necroptosis regulation system on gut immune cells, mesenchymal cells, and /or neural cells. Considering the nature of inflammation, it is important address whether P2Y₁₄R positive immune cells, mesenchymal cells and/or neural cells are involved in the UC inflammation or not? Since the authors are focusing on the P2Y₁₄R epithelial cell mediated inflammation in UC, it would be useful to provide additional evidence supporting their view. What is the definitive evidence for the specific association of the P2Y₁₄R epithelial cell mediated inflammation with UC but not with other inflammatory diseases?

We appreciate your clear and detailed feedback. In the revised manuscript, we designed and supplemented relevant experiments to study the effect of P2Y₁₄R expressed by myeloid cells, mesenchymal cells as well as goblet cells in the pathogenesis of UC. Besides, we supplemented both the Chronic DSS colitis model and the TNBS-induced colitis model to further investigate the differential effect of IECs P2Y₁₄R in these different colitis models. The relevant results are presented in the following “point-to-point” responses.

In the remainder of this letter, we discuss each of your comments individually along with our corresponding responses. To facilitate this discussion, we first retype your comments and then present our responses to the comments.

Specific Comments

1. The author predicted that P2Y₁₄R in the intestinal epithelium plays a central role in the control of inflammation in UC based on the previous report indicating the association of the inflammatory score and the expressions of P2Y₁₄R in the mucosa. However, P2Y₁₄R expresses in the various cell types including macrophage, mesenchymal cells and neutrophils and these cells are also involved in the development of intestinal inflammation and considered as initial targets of IBD therapy. These important points should not be ignored and should experimentally addressed and discussed.

Response: Thank you for your constructive suggestion. To confirm the critical role of P2Y₁₄R expressed by IECs in the process of UC, we conducted the following experiments:

First, we induced experimental colitis in mice with myeloid cells-specific P2Y₁₄R knockout (P2Y₁₄R *Lyz2-cre*). The results showed that myeloid cells-specific P2Y₁₄R knockout did not lead to

significant improvements in DSS-induced weight loss, diarrhea, rectal bleeding, or colon shortening (**Fig #1b-c**). Additionally, histopathology examination also indicated that myeloid cells-specific P2Y₁₄R deficiency could not improve DSS-induced experimental colitis (**Fig #1d**). These findings demonstrated that P2Y₁₄R expressed by myeloid cells did not play a major role in DSS-induced inflammation. We have incorporated these results in the **Supplementary Fig. 1 e-h** of the revised manuscript. Besides, we have also included the related method in the revised manuscript's method section.

Fig #1. a. Experimental Flow Chart. **b.** Body weight change and DAI evaluation during the disease process (n = 6). **c.** The length of colons from P2Y₁₄R^{fl/fl} and P2Y₁₄R^{fl/fl} Lyz2-cre mice after DSS treatment. **d.** The H&E staining in colon tissues of DSS-treated mice (Scale bar = 200 μm). The data represent the mean ± SD, and statistical significance was determined by two-way analysis of variance (ANOVA) with Sidak's multiple comparisons test for **b**, and One-way analysis of variance (ANOVA) with Tukey multiple comparison test for **c**. #*P* < 0.05, ##*P* < 0.01, and ####*P* < 0.001 compared to P2Y₁₄R^{fl/fl}+H₂O group; **P* < 0.05 compared to P2Y₁₄R^{fl/fl}+DSS group.

Next, we investigate the role of P2Y₁₄R expressed by MSCs in the process of UC. Recently, the studies between mesenchymal cells and IBD mainly focus on the stem cell therapy in IBD (Panes J, et al. *Lancet* 2016; Molendijk I, et al. *Gastroenterology* 2015). Stem cell therapy has been reported to improve IBD through tissue repair, paracrine effects, and the promotion of angiogenesis, immune regulation, and anti-inflammatory effects (Tian CM, et al. *J Inflamm Res.* 2023). However, there are few studies investigating the role of endogenous mesenchymal cells in the pathogenesis of IBD. Thus, we isolated adipose derived mesenchymal stem cells (MSCs) from WT and P2Y₁₄R whole body knockout mice. Then we established DSS-induced experimental colitis in WT mice, and administrated WT or P2Y₁₄R^{-/-} MSCs (10⁶ cells every mouse) twice on days 3 and 5 to the mice through tail intravenous injection (**Fig #2a**). The results showed that MSCs administration could

significantly improve DSS-induced weight loss, disease activity index, colon shortening as well as pathological damage, whether expression P2Y₁₄R or not (**Fig #2b-d**), indicating that P2Y₁₄R expressed by MSCs did not participate in DSS-induced inflammation.

Fig #2. a. Experimental Flow Chart. **b.** Body weight change and DAI evaluation during the disease process (n = 6). **c.** The length of colons 7 days after DSS treatment. **d.** The H&E staining in colon tissues of DSS-treated mice (Scale bar = 200 μ m). The data represent the mean \pm SD, and statistical significance was determined by two-way analysis of variance (ANOVA) with Sidak's multiple comparisons test for **b**, and One-way analysis of variance (ANOVA) with Tukey multiple comparison test for **c**. [#]*P* < 0.05, ^{##}*P* < 0.01, and ^{###}*P* < 0.001 compared to P2Y₁₄R^{fl/fl}+H₂O group; **P* < 0.05 compared to P2Y₁₄R^{fl/fl}+DSS group.

2. It is critical to include the detailed data showing the molecular cellular mechanisms of upregulation of P2Y₁₄R expression on colonic epithelial cells. How is the receptor expression elevated in UC epithelium? What is molecular and cellular mechanisms for the receptor induction? Response: Thank you for this valuable feedback and constructive suggestions.

P2Y₁₄R, as a classical G-protein coupled receptor, could be up-regulated by its specific ligand (Tannenbaum GS, et al. *Neuroendocrinology*. 2001). Meanwhile, UDPG, a damage-associated molecular pattern (DAMP), was reported to be released by damaged cells and played a crucial role in the process of liver fibrosis and acute kidney injury by mediating intercellular communication. (Mederacke I et al., *Sci Transl Med*. 2022; Battistone MA, et al. *J Clin Invest*. 2020). Consistently, we detected an increase in P2Y₁₄R expression of HT-29 cells 12 hours after UDPG treatment (**Fig #3a**). Besides, the UDPG levels in the supernatant culture medium of TSZ-treated HT-29 cells showed a dramatic increase in UDPG concentration compared to the control group. Therefore, the upregulation of P2Y₁₄R in intestinal epithelia tissues could be induced by the UDPG released from

damaged cells. However, since the protein structure of P2Y₁₄R has not been analyzed yet, the exact mechanism of changes in the expression of P2Y₁₄R requires further investigation.

Fig #3. a. Western blot assay of P2Y₁₄R protein expression in HT-29 cells after 12 hours treated with PBS (Veh) or UDPG (n = 3). **d.** P2Y₁₄R siRNA or P2Y₁₄R plasmid transfected HT-29 cells were simulated with TSZ for 6 h, UDPG in supernatants was determined by ELISA (n = 4). The data represent the mean ± SD, and statistical significance was determined by Unpaired t test, **P* < 0.05, ***P* < 0.01, and ****P* < 0.001.

3. In related to above comment, the authors emphasized the correlation between the P2Y₁₄R expression and inflamed epithelial cells in UC. Does only the colon epithelium enhance the expression of P2Y₁₄R in the inflammatory condition? The expression of P2Y₁₄R in the colon mucosa of patients with Crohn's disease (CD) should be compared with UC. In addition to the data generated by the public depository (e.g., NCBI's Gene Expression Omnibus: GSE38713, GSE75214, GSE6879), this issue needs to be directly demonstrated by the related tissue specimens from different forms of inflammation.

Response: We appreciate your constructive comments. According to the research conducted by Rybaczyk L and colleagues, they analyzed the expression profiles of 22 purine genes and 36 probe-sets from NCBI GEO with the Comparative Analysis of Gene Expression and Selection (CAGES) method to investigate the purine gene expression patterns in IBD. The gene chip expression datasets generated from colonic mucosal biopsies or Peripheral Blood Mononuclear Cells (PBMCs) derived from IBD patients with Ulcerative Colitis (UC) or Crohn's Disease (CD) showed that the P2Y₁₄R gene was positively correlated with the inflammatory score in UC mucosal biopsies while negatively with the inflammatory score in CD mucosal biopsies. However, no expression differences were found between the PBMCs of healthy volunteers and IBD patients. (Rybaczyk L, et al. *Inflamm Bowel Dis.* 2009).

Image from Rybaczyk L, et al. *Inflamm Bowel Dis.* 2009

To further confirm these results, we performed an Immunofluorescence (IF) assay using anti-human P2Y₁₄R antibody with non-IBD colon tissues as well as inflamed colon tissues from UC and CD patients. The results showed that the expression of P2Y₁₄R protein in colon tissues was barely detectable in healthy controls and CD patients, but significantly higher in patients with UC (Fig #4). This finding is consistent with the RNA-seq results shown in Fig. 1a and the analysis conducted by Rybaczyk L and her colleagues. We have also incorporated these results in Fig. 1b of the revised manuscript. All the method and the information of primary antibody have been included in the method section of the revised manuscript.

Fig #4. Immunofluorescence (IF) assay with anti-human P2Y₁₄R antibody using colon tissues from healthy control and UC or CD patients. The immunofluorescent images of colon tissues from healthy control and UC or CD patients. The immunofluorescent images of colon tissues from healthy control and UC or CD patients.

control and UC or CD patients stained with P2Y₁₄R (red), EpCAM (green) and nuclear (DAPI, Blue) (Scale bar = 100 μm).

4. The authors somehow did not provide any data related to the phenotypical assessment of IEC-P2Y₁₄R deficient mice (P2Y₁₄R^{ΔIEC}) in the steady state. Is there a difference in epithelial cell integrity? It has been shown that goblet cells express functional P2Y₁₄R. It is also important to study whether the P2Y₁₄R^{ΔIEC} influence on goblets cells or not? If so, P2Y₁₄ deficiency could alter commensal bacteria and mucin secretion, which might influence the outcome of intestinal inflammation.

Response: Thank you for your valuable suggestion. First, we conducted an intestinal permeability assessment in P2Y₁₄R^{ΔIEC} and P2Y₁₄R^{fl/fl} mice using FITC-dextran to further assess the influence of IEC-P2Y₁₄R deficient on epithelial cell integrity. The results showed that IEC-specific P2Y₁₄R deficient did not influence the intestinal permeability in the steady state (Fig #5). We have incorporated these results in the Fig 1j of the revised manuscript.

Fig #5. Effects of IEC-specific P2Y₁₄R knockout on mice mucosal barrier function as measured by serum levels of FITC-dextran based on intestinal permeability methods. The data represent the mean ± SD, and One-way analysis of variance (ANOVA) with Tukey multiple comparison test. **P* < 0.05, ***P* < 0.01, and ****P* < 0.001.

Next, given that IEC-specific knockout mice established by hybridizing with Vil-cre mice can influence the expression of corresponding genes in goblet cells, we are very grateful to you for reminding us of this. To test if P2Y₁₄R on goblet cells is involved in regulating UC inflammation,

we detected the impact of P2Y₁₄R deficiency on MUC2 release in LS174T cells 30 minutes after treatment with ATP or PMA. Western blot analysis showed that P2Y₁₄R knockdown did not influence MUC2 release in LS174T cells (**Fig #6**), indicating that the improvement effect we found in P2Y₁₄R^{ΔIEC} mice was closely related to the downregulation of P2Y₁₄R in IECs.

Fig #6. MUC2 released levels in LS174T cell supernatant was analyzed by immunoblotting with corresponding antibodies (n = 3). The data represent the mean ± SD, and One-way analysis of variance (ANOVA) with Tukey multiple comparison test. **P* < 0.05, ***P* < 0.01, and ****P* < 0.001.

5. The authors emphasized the critical role of UDP-G-P2Y₁₄R mediated signaling cascade for the epithelia cell necroptosis and inflammation. It is useful to provide additional data related to how does the UDP-G involve in the epithelial integrity or necroptosis during inflammation?

Response: We appreciate your constructive comments. To investigate how UDPG was involved in the epithelial integrity or necroptosis, we first generated intestinal epithelial organoids from P2Y₁₄R^{fl/fl} or P2Y₁₄R^{ΔIEC} mice and pretreated the organoids with UDPG before stimulated with TSZ to induce necroptosis. In P2Y₁₄R^{fl/fl} organoids, the results of PI staining showed a dramatic increase of PI intensity in UDPG treated group. In contrast, UDPG treated in P2Y₁₄R^{ΔIEC} organoids did not display significant influence in TSZ-induced necroptosis (**Fig #7**). These results indicating that UDPG involved in TSZ-induced necroptosis rely on the activity of P2Y₁₄R. These results have been included in the **supplement Fig 3e** of the revised manuscript. All the method and reagents have been included in the method section of the revised manuscript.

Fig #7. Intestinal organoids from $P2Y_{14}R^{fl/fl}$ or $P2Y_{14}R^{\Delta IEC}$ mice treated as indicated with DMSO (Veh), TNF- α adding Smac mimetic and z-VAD (TSZ) or TSZ+UDPG for 12 h and stained with PI (red).

Next, We challenged $P2Y_{14}R^{fl/fl}$ or $P2Y_{14}R^{\Delta IEC}$ mice with 3% DSS and rectally administered the UDPG in sterile PBS daily (**Fig #8a**). Treatment with UDPG aggravate weight loss, diarrhea, rectal bleeding and inflammatory infiltration in $P2Y_{14}R^{fl/fl}$ mice. However, in $P2Y_{14}R^{\Delta IEC}$ mice there is still no significant different between mice with or with UDPG-treated (**Fig #8b-d**). All of these results concluded that UDPG involved in IECs necroptosis and DSS-induced inflammation through UPDG/ $P2Y_{14}R$ axis. These results have been included in the **supplement Fig 3a-d** of the revised manuscript. All the method and reagents have been included in the method section of the revised manuscript.

Fig #8. a. Experimental Flow Chart. **b.** Body weight change and DAI evaluation during the disease process (n = 6). **c.** The length of colons from P2Y₁₄R^{fl/fl} and P2Y₁₄R^{ΔIEC} Lyz2-cre mice after DSS treatment. **d.** The H&E staining in colon tissues of DSS-treated mice (Scale bar = 200 μm). The data represent the mean ± SD, and statistical significance was determined by One-way analysis of variance (ANOVA) with Tukey multiple comparison test for **d** and two-way analysis of variance (ANOVA) with Sidak's multiple comparisons test for **b**. #P < 0.05, ##P < 0.01, and ###P < 0.001 compared to P2Y₁₄R^{fl/fl}+H₂O group; *P < 0.05 compared to P2Y₁₄R^{fl/fl}+DSS group.

6. The author showed the expressions of Ugp2 and Gys1 in the mucosal compartment from the public data base and concluded the accumulation of UDP-G in patients with UC. This is very important point of the study and thus the authors should provide additional in vivo and/or in vitro data supporting their claim. In related to the issue, UDP- glucose is synthesized by UDP glucose pyrophosphorylase 2 (Ugp2) and then degraded by glycogen synthase 1 (Gys1) (Fig. 5a). As the authors recognized this conflicting result, this is an important part of the present study, however the authors somehow did not directly address and discuss the interpretation of simultaneous elevation of Ugp2 and Gys1 in intestinal inflammation.

Response: We are grateful to you for this constructive comment. In the process of glycogen metabolism, UDP-glucose is synthesized by UDP glucose pyrophosphorylase 2 (*Ugp2*) and then degraded by glycogen synthase 1 (*Gys1*).

After reviewing the GEO datasets and adding more relevant data, it became clear that the expression of *Ugp2* was found to decrease in the intestinal mucosa of patients with inflammatory bowel disease (IBD). However, the expression of *Gys1* in IBD patient's intestinal mucosa remains unclear. Furthermore, we tested the expression of *Ugp2* and *Gys1* in the intestinal mucosa of DSS-induced colitis mice. A simultaneous decrease in the expression of *Ugp2* and *Gys1* was observed after DSS treatment.

On the other hand, as a specific ligand for P2Y₁₄R, UDPG is released extracellularly to activate P2Y₁₄R and initiate downstream signaling pathways, suggesting that the activation of P2Y₁₄R might not be directly associated with the levels of intracellular UDPG. Notably, UDPG was reported to be released by necrotic hepatocytes, serving as a signaling molecule to promote the activation of hepatic stellate cells (Mederacke I, et al. *Sci Transl Med.* 2022). Similarly, ATP released by chemotherapy-induced dead tumor cells was found to activate P2X₄, which also belong to P2 purinergic receptor family, and mediated an mTOR-dependent pro-survival program in neighboring cancer cells (Schmitt, et al. *Nature*, 2022). It should be noted that in these two studies, the elevation of UDPG or ATP levels in the extracellular environment was not due to an increase in synthesis or a decrease in degradation but was caused by cell death instead.

Therefore, we supposed that, in this study, the elevation of UDPG in the inflammatory microenvironment of UC is mainly caused by dying IECs. Thank you again for your professional question, which led us to have a deeper thinking on this phenomenon.

Fig #9. a. *Ugp2* and *Gys1* expression in the colonic mucosa of healthy individuals or IBD patients from GEO database (using datasets GSE11223, G75214, GSE16879, GSE235236, GSE174159, GSE117993, GSE126123). **b.** RT-qPCR analysis of *Ugp2* and *Gys1* mRNA of intestinal epithelial cell in mice ($n = 3$). The data represent the mean \pm SD, and One-way analysis of variance (ANOVA) with Tukey multiple comparison test, * $P < 0.05$, ** $P < 0.01$, and *** $P < 0.001$.

7. The discovery of the novel inhibitor of P2Y14R “DL-16” is one of the strong advances of this manuscript. However, the current study is lacking the detailed pharmacological study including the detailed experimental protocols (e.g., concentration, administration route and frequency and the vehicle control). Administration of the inhibitor candidate to IEC P2Y14R deficient mice would strengthen the author’s conclusion.

Response: Thank you for point out this problem. We are so sorry for missing out the detailed experimental protocols of HDL-16 administration. In the pharmacological study, we administrated HDL-16 daily through rectal administration, and the solvent system of HDL-16 and PPTN was 1% DMSO. This part of protocol has been added in the method section in our revised manuscript. Thanks very much again for mention us this problem.

Next, as you suggested, we administered HDL-16 or PPTN to P2Y₁₄R^{ΔIEC} mice in order to further confirm that HDL-16 improved DSS-induced experimental colitis by targeting the P2Y₁₄R expressed by IECs. The results showed that daily administration with HDL-16 or PPTN showed barely improvement in DSS-fed P2Y₁₄R^{ΔIEC} mice. Specifically, treatment with HDL-16 or PPTN did not influence DSS-fed induced weight loss, diarrhea, rectal bleeding, mucosal barrier injury and inflammatory cell infiltration (**Fig #10 a-f**). Besides, western blot results showed that HDL-16 or PPTN treatment did not affect DSS-induced upregulation of p-MLKL in P2Y₁₄R^{ΔIEC} mice (**Fig #10 g**). These results indicated that the improvement effect of HDL-16 on DSS-induced experimental colitis relied on the targeted P2Y₁₄R of IECs. These results have been included in the **supplement Fig 4c-i** of the revised manuscript. All the method and reagents have been included in the method section of the revised manuscript.

Fig #10. a. Experimental Flow Chart, P2Y₁₄R^{ΔIEC} mice exposed to 3% DSS intraperitoneally received Low-dose HDL-16, High-dose HDL-16, or PPTN throughout the entire experimental period (n = 6). **b.** Body weight change and disease activity index evaluation of mice change during the disease process **c.** Effects of P2Y₁₄R inhibitors on P2Y₁₄R^{ΔIEC} mice mucosal barrier function as measured by serum levels of FITC-dextran based on intestinal permeability methods. **d.** The length of colons from mice 7 days after DSS treatment. **e.** The H&E staining in the colon tissues of DSS-treated mice (Scale bar = 200 μm). **f.** The immunofluorescent images of colon tissues stained with Claudin-1, Occludin and ZO-1, the principal components of tight junction (Scale bar = 200 μm). **g.**

Phosphorylation MLKL as well as its protein levels were analyzed by immunoblotting with corresponding antibodies in colon tissues (n = 5). The data represent the mean ± SD, and statistical significance was determined by One-way analysis of variance (ANOVA) with Tukey multiple comparison test for **c, d, g** and two-way analysis of variance (ANOVA) with Sidak's multiple comparisons test for **a, b**. **P* < 0.05, ***P* < 0.01, and ****P* < 0.001 compared to P2Y₁₄R^{fl/fl}+H₂O group.

8. The authors stated in their Introduction, “An epidemiological study purine gene dysregulation profiles in IBD showed that 59% of purine genes was dysregulation in IBD, but only the expression of P2Y₁₄R was positively correlated with acute inflammatory score in UC mucosal biopsies.” (lines 66-69). Based on this remark, it is important to elucidate and compare the P2Y₁₄R-mediated necroptosis regulation in the acute and chronic phases of the DSS induced inflammation using wild type and their unique P2Y₁₄RΔIEC mouse models.

Response: We appreciate your constructive comment. As your suggestion, we performed chronic DSS colitis model in P2Y₁₄R^{ΔIEC} or P2Y₁₄R^{fl/fl} mice (**Fig #11 a**). The results showed that IEC-P2Y₁₄R deficient also displayed significant improvement in weight loss, diarrhea, rectal bleeding as well as histologic damage in chronic DSS colitis model especially during feeding period. But the improvement effect of rectum shorten in chronic DSS colitis model was not as obvious as the acute DSS model (**Fig #11 b-d**), which probably because the intervene of adaptive immune system caused by recurrent inflammation (Wirtz S, et al. *Nat Protoc.* 2007.). In the future, P2Y₁₄R targeted therapy may mainly focus on patients with UC during the acute attack stage. These results have been included in the **supplement Fig 1a-d** of the revised manuscript. All the method and reagents have been included in the method section of the revised manuscript.

Fig #11. a. Experimental Flow Chart (n = 6). **b.** Body weight change and disease activity index evaluation of mice change during the disease process **c.** The length of colons from mice after DSS

treatment. **d.** The H&E staining in the colon tissues of DSS-treated mice (Scale bar = 200 μ m). The data represent the mean \pm SD, and statistical significance was determined by two-way analysis of variance (ANOVA) with Sidak's multiple comparisons test for **b**, and One-way analysis of variance (ANOVA) with Tukey multiple comparison test for **c**. # $P < 0.05$, ## $P < 0.01$, and ### $P < 0.001$ compared to P2Y₁₄R^{fl/fl}+H₂O group; * $P < 0.05$, ** $P < 0.01$, and *** $P < 0.001$ compared to P2Y₁₄R^{fl/fl}+DSS group.

9. The authors demonstrated the elevated P2Y₁₄R expression in UC specimens compared with those in healthy controls based on the public datasets (using 91 datasets from NCBI's Gene Expression Omnibus: GSE38713, GSE75214, GSE6879) (lines 89-91 and Figure 1a). It is useful to include actual clinical and histological data showing the increase in the receptor expression in the human colon epithelium of UC but not CD in addition to the murine data.

Response: We appreciate your constructive comment. According to your suggestion, we performed immunolocalization experiments on the inflamed colon tissues from UC and CD patients, the non-IBD colon tissues were used as healthy control. The results showed that the expression of P2Y₁₄R protein in colon tissues was barely detectable in healthy controls and CD patients, but significantly higher in patients with UC (**Fig #4**). This finding is consistent with the analysis results from public datasets shown in **Fig. 1a**. We have also incorporated these results in **Fig. 1b** of the revised manuscript. All the methods and the information of primary antibodies have been included in the method section of the revised manuscript.

Fig #4. Immunofluorescence (IF) assay with anti-human P2Y₁₄R antibody using colon tissues from healthy control and UC or CD patients. The immunofluorescent images of colon tissues from healthy control and UC or CD patients stained with P2Y₁₄R (red), EpCAM (green) and nuclear (DAPI, Blue) (Scale bar = 100 μ m).

10. To demonstrate the role of the P2Y₁₄R-mediated signaling cascade for necroptosis, the authors used HT-29 cells, a human intestinal epithelial cell line. Although HT-29 cells have been used extensively in in vitro study, unfortunately it is a carcinoma-derived cell line that may not reflect the actual situation in vivo. However, recent progress using human iPS cells and/or tissue-derived organoid and epithelial cells is now available, and thus the authors should consider the use of more physiologically relevant in vitro models.

Response: We appreciate your valid suggestion and have supplemented intestinal epithelial organoids experiments and HCT-116 cell to further verify our findings in HT-29 cells, the detailed results are summarized as follows.

First, we generated intestinal epithelial organoids from P2Y₁₄R^{fl/fl} or P2Y₁₄R^{ΔIEC} mice and then stimulated them with TSZ for 12 hours (Patankar JV, et al. *Nat Cell Biol.* 2021). We found that, similar to the results in HT-29 cells, P2Y₁₄R deficient significantly inhibited TSZ-induced necroptosis in P2Y₁₄R^{ΔIEC} organoids (**Fig. #12a**). Following, we pre-treated P2Y₁₄R^{ΔIEC} organoids with SQ22536, H-89 or 666-15 and then induced necroptosis. As expected, inhibition of cAMP/PKA/CREB pathway disrupted the protective effect of P2Y₁₄R deficiency in TSZ mediated necroptosis (**Fig. #12b**). Furthermore, we found UDPG treatment could increase TSZ induced necroptosis in organoids from P2Y₁₄R^{fl/fl} mice, while administration with Nec-1s and GSK'872 remarkably attenuated the UDP-glucose derived damage on TSZ-stimulated necroptosis (**Fig. #12c**). More importantly, administration with P2Y₁₄R inhibitor **HDL-16** also protected P2Y₁₄R^{fl/fl} organoids from TSZ induced necroptosis (**Fig. #12d**). These results were all included in the revised **Fig 2k, Fig 4l, Fig 5j and Fig 7m**. Related methods have been included in the method part of the revised manuscript.

Fig.#12 a. Intestinal organoids from $P2Y_{14}R^{fl/fl}$ or $P2Y_{14}R^{\Delta IEC}$ mice treated as indicated with DMSO (veh), or TNF- α adding Smac mimetic and z-VAD (TSZ) for 12 h and stained with PI (red). **b.** Intestinal organoids from $P2Y_{14}R^{\Delta IEC}$ mice treated as indicated with DMSO (veh), TSZ, TSZ adding SQ22536, TSZ adding H-89 or TSZ adding 666-15. **c.** Intestinal organoids from $P2Y_{14}R^{\Delta IEC}$ mice treated as indicated with DMSO (veh), TSZ, TSZ + UDPG, TSZ+UDPG+Nec-1s or TSZ+UDPG+GSK'872. **d.** Intestinal organoids from $P2Y_{14}R^{fl/fl}$ mice treated as indicated with DMSO (veh), TSZ, TSZ + HDL-16 or TSZ+PPTN.

Next, we repeated our *in vitro* experiments in HCT-116 cells. Similar with the result in HT-29 cells, $P2Y_{14}R$ silence inhibit TSZ-induced necroptosis according to the results of PI staining and western blot results of p-MLKL (**Fig. #13a-b**). The lack of $P2Y_{14}R$ in HCT-116 cells resulted in a decrease in the expression and mRNA level of *Ripk1*, while the overexpression of $P2Y_{14}R$ led to an upregulation of *Ripk1* under TSZ condition, indicating that $P2Y_{14}R$ participate in TSZ-induced necroptosis through influence the transcript of RIPK1 (**Fig. #13c-d**). The detection of intracellular cAMP, p-PKA and p-CREB indicating that cAMP/PKA/CREB pathway could involve in the regulation of $P2Y_{14}R$ on TSZ-induced necroptosis of HCT-116 cells (**Fig. #13e-f**). Then, we pre-treatment $P2Y_{14}R$ silence HCT-116 cells with SQ22536, H-89 and 666-15 to further confirm that $P2Y_{14}R$ regulated TSZ-induced necroptosis rely on cAMP/PKA/CREB pathway. The results showed that, pre-treatment with SQ22536, H-89 and 666-15 in HCT-116 cells disrupted the protective effect on TSZ-induced necroptosis and suppressed the transcript of *Ripk1* (**Fig. #13g-i**). Finally, we pre-administrated HCT-116 cells with UDPG for 12 h and found that UDPG exacerbated TSZ-induced necroptosis according to the upregulation of PI positive cells and p-

MLKL level. The necroptotic cell death was further confirmed using Nec-1s and GSK'872, where Nec-1s and GSK'872 treatment remarkably attenuated the UDP-glucose derived damage on TSZ-stimulated necroptosis (Fig. #13j-k). Besides, UDPG treatment aggravated the inhibition of TSZ on cAMP/PKA/CREB pathway according to the level of p-PKA, p-CREB, the mRNA level of *Ripk1* as well as the binding between CREB and *Ripk1* promoter (Fig. #13j-k). These results further demonstrated that UDP-glucose partly contribute to the activation of the P2Y₁₄R in regulating IECs necroptosis in IBD. These results were all included in the revised Fig 2i-j, Fig 3b and d, Fig 4c-d and i-k, Fig 5h-i and n-p. Related methods have been included in the method part of the revised manuscript. The primer sequence has been included in the supplementary file of the revised manuscript.

Fig#13 a. PI/Hoechst staining were used to analyzed PI positive cells in HCT-116 cells treated as indicated with veh or TSZ. **b.** Phosphorylation of MLKL as well as its protein levels were analyzed by immunoblotting with corresponding antibodies in HCT-116 cells. **c.** The expression of *Ripk1* were analyzed by immunoblotting with corresponding antibodies in HCT-116 cells. **d.** The mRNA level of *Ripk1* in siP2Y₁₄R or P2Y₁₄R-OE HCT-116 cells treated with Veh or TSZ was analyzed by RT-PCR. **e.** The intracellular cAMP level in HCT-116 cells. **f.** Phosphorylation of PKA and CREB

as well as their protein levels were analyzed by immunoblotting with corresponding antibodies in HCT-116 cells. **g.** PI/Hoechst staining were used to analyzed PI positive cells in siP2Y₁₄R HCT-116 cells treated as indicated with Veh, TSZ, TSZ+SQ22536, TSZ+H-89, TSZ+666-15. **h.** Phosphorylation of MLKL as well as its protein levels were analyzed by immunoblotting with corresponding antibodies in siP2Y₁₄R HCT-116 cells. **i.** The mRNA level of *Ripk1* in siP2Y₁₄R HCT-116 cells treated with Veh, TSZ, TSZ+SQ22536, TSZ+H-89, TSZ+666-15 was analyzed by RT-PCR. **j.** PI/Hoechst staining were used to analyzed PI positive cells in HCT-116 cells treated as indicated with Veh, TSZ, UDPG, UDPG+TSZ, UDPG+TSZ+Nec-1, UDPG+TSZ+GSK'872. **k.** Phosphorylation of MLKL as well as its protein levels were analyzed by immunoblotting with corresponding antibodies in HCT-116 cells. **l.** Phosphorylation of PKA and CREB as well as their protein levels were analyzed by immunoblotting with corresponding antibodies in HCT-116 cells. **m.** The mRNA level of *Ripk1* in HCT-116 cells treated with Veh, TSZ, UDPG, UDPG+TSZ, UDPG+TSZ+Nec-1, UDPG+TSZ+GSK'872 was analyzed by RT-PCR. **n.** ChIP with anti-CREB of the regions containing the CREB binding sites on the RIPK1 gene promoter in HCT-116 cells treated with Veh, TSZ, UDPG, UDPG+TSZ, UDPG+TSZ+Nec-1, UDPG+TSZ+GSK'872. The data represent the mean \pm SD, and statistical significance was determined by two-way analysis of variance (ANOVA) with Sidak's multiple comparisons test for **c, f** and **l**, and One-way analysis of variance (ANOVA) with Tukey multiple comparison test for **a-b, d-e, g-k, m-n**. * $P < 0.05$, ** $P < 0.01$, and *** $P < 0.001$.

11. The authors extensively investigated the molecular aspect of the P2Y₁₄R mediated signaling cascade using the HT-29 in vitro system and indicated that P2Y₁₄R regulated necroptosis in IECs at least in part by altering the activation of the cAMP / PKA / CREB pathway and further mediating the transcript of RIPK (lines 246-248). However, it was not clear how these in vitro data directly related and/or contributed to our understanding of pathological aspects of inflammation mediated by the P2Y₁₄R signaling cascade in the intestinal epithelium.

Response: Thank you a lot for your valuable suggestion. To prove the role of cAMP/PKA/CREB pathway, we first examined the influence of IEC-specific P2Y₁₄R knockout on the phosphorylation level of PKA and CREB in IECs of DSS-induced colitis mice. As expected, the activation of PKA/CREB pathway was suppressed by DSS feeding. However, IEC-specific P2Y₁₄R knockout partly upregulated the phosphorylation level of PKA and CREB (**Fig #14a**). Consistent with this,

administration of P2Y₁₄R antagonist HDL-16 significantly improved the activity of PKA and CREB in IECs of DSS-induced colitis mice (**Fig #14c**). More importantly, we further investigated the influence of IEC-specific P2Y₁₄R knockout and HDL-16 treatment on the binding between CREB and the promoter of *Ripk1*. The results of ChIP-PCR assay showed that DSS feeding suppressed the binding between CREB and the promoter of *Ripk1*, while IEC-specific P2Y₁₄R knockout and HDL-16 treatment promoted this binding (**Fig #14b** and **d**). We have incorporated these results into **Fig 3i**, **Fig 7i** and **k** in the revised manuscript.

Fig.#14 a. The phosphorylation of PKA and CREB as well as its protein levels were analyzed by immunoblotting with corresponding antibodies in the IECs from P2Y₁₄R^{fl/fl} and P2Y₁₄R^{ΔIEC} mice after DSS treatment (n = 6). **b.** ChIP with anti-CREB of the regions containing the CREB binding sites on the *Ripk1* gene promoter in DSS-treated P2Y₁₄R^{fl/fl} or P2Y₁₄R^{ΔIEC} intestinal epithelial cell (n = 3). **c.** The phosphorylation of Phosphorylation PKA, CREB as well as its protein levels were analyzed by immunoblotting with corresponding antibodies in DSS-fed mice colon tissues (n = 5). **d.** ChIP with anti-CREB of the regions containing the CREB binding sites on the *Ripk1* gene promoter in intestinal epithelial cell of HDL-16-L, HDL-H or PPTN administrated experimental colitis mice (n = 3). The data represent the mean ± SD, and statistical significance was determined by two-way analysis of variance (ANOVA) with Sidak's multiple comparisons test for **a** and **c**, and One-way analysis of variance (ANOVA) with Tukey multiple comparison test for **b** and **d**. **P* < 0.05, ***P* < 0.01, and ****P* < 0.001.

12. In terms of role of the UDP-glucose-P2Y₁₄R mediated promotion of necroptosis of IECs, the data were provided by the HT-29 in vitro system. Since the authors have established a unique in

vivo model of the P2Y₁₄R^{ΔIEC} mouse DSS model, it would be beneficial to adopt the system and compared to wild-type mice for a greater understanding of the UDP-glucose-P2Y₁₄R mediated regulation of inflammation in vivo.

Response: Thank you a lot for your valuable suggestion. As your suggestion, we challenged P2Y₁₄R^{fl/fl} and P2Y₁₄R^{ΔIEC} mice with 3% DSS, then rectal administration UDPG daily and monitored for weight and disease onset (Fig #8a). We found that administration of UDPG aggravated DSS-induced weight loss, diarrhea and rectal bleeding in P2Y₁₄R^{fl/fl} mice. However, in P2Y₁₄R^{ΔIEC} mice, administration of UDPG did not display obviously different with the mice administration vehicle (Fig #8b). The results of Colon length and histologic analysis also showed similar results (Fig #8c-d). Collectively, these results strongly suggested that UDPG-P2Y₁₄R axis was involved in the regulation of IBD. We have included these results in the supplement Fig 3a-d. Thanks again for your fabulous ideas helping us improve our research.

Fig #8. a. Experimental Flow Chart. b. Body weight change and DAI evaluation during the disease process (n = 6). c. The length of colons from P2Y₁₄R^{fl/fl} and P2Y₁₄R^{ΔIEC} mice after DSS treatment. d. The H&E staining in colon tissues of DSS-treated mice (Scale bar = 200 μm). The data represent the mean ± SD, and statistical significance was determined by two-way analysis of variance (ANOVA) with Sidak's multiple comparisons test for **b**, and One-way analysis of variance

(ANOVA) with Tukey multiple comparison test for **c**. # $P < 0.05$, ## $P < 0.01$, and ### $P < 0.001$ compared to P2Y₁₄R^{fl/fl}+H₂O group; * $P < 0.05$ compared to P2Y₁₄R^{fl/fl}+DSS group.

13. The authors' efforts on the discovery of P2Y₁₄R antagonist (e.g., DL-16) with their potent binding affinity to the receptor and their ability to inhibit necroptosis of epithelial cells both in vivo and in vitro must be recognized and congratulated. Based on the amount of and biomedical significance of the data, it might be better to publish as a separate study focusing on the newly identified P2Y₁₄R antagonists with additional pharmacological and clinically applicable data.

Response: Thank you for your valuable suggestion. HDL-16 as a self-developed P2Y₁₄R antagonist was introduced to our study to confirm the feasibility of P2Y₁₄R as a therapeutic target for UC. We aim to provide data supporting P2Y₁₄R-targeted therapy of UC by investigating the treatment effect of HDL-16 in a DSS-induced experimental colitis model. Fortunately, we found that targeting P2Y₁₄R with a small molecule antagonist could effectively alleviate UC, suggesting that P2Y₁₄R might serve as a promising target for UC treatment, which is the most significant contribution of our present study. On the other hand, according to your suggestion, we supplement several experiments to emphasize the critical role of P2Y₁₄R expressed by IECs in the process of UC. In future studies, we will conduct a comprehensive drug evaluation of HDL-16, including general pharmacology, safety, and potential clinical applications.

14. Related to the discovery of a P2Y₁₄R antagonist (e.g., DL-16), it is critical to elucidate whether the antagonist specifically controls epithelial cell inflammation associated with UC or other forms of intestinal inflammation? Is it specific for inflammation that occurs in colon epithelial cells? Further, it is also important to demonstrate whether the antagonist influence on other inflammatory cells such as macrophages, mesenchymal cells and neutrophils in UC and/or other forms of intestinal inflammation, since the authors' group have previously shown the role of P2Y₁₄R in regulating caspase-1-mediated pyroptosis in macrophages and NETosis of neutrophils.

Response: We appreciate your valuable comments.

P2Y₁₄R has been reported to have a functional expression in many types of cells, including macrophages, mesenchymal cells, and neutrophils. However, our supplement experiments have showed that the myeloid cell-specific P2Y₁₄R knockout did not improve DSS-induced experimental colitis. This indicates that P2Y₁₄R expressed by myeloid cell did not participate in the pathogenesis of ulcerative colitis. While WT and P2Y₁₄R^{-/-} MSCs administered showed similar therapeutic effects in DSS-fed mice, it suggested that P2Y₁₄R expressed by MSCs was not involved in the regulation of DSS-induced inflammation as well. Moreover, the significant improvement of IEC-specific

P2Y₁₄R knockout in DSS-induced acute colitis models strongly indicated the crucial role of P2Y₁₄R expressed by IECs.

As an antagonist of P2Y₁₄R, we believed that HDL-16 improved UC by blocking the UPDG/P2Y₁₄R axis. In our revised manuscript, we generated DSS-induced chronic colitis model and TNBS-induced colitis model in P2Y₁₄R^{fl/fl} and P2Y₁₄R^{ΔIEC} mice and found that IEC-specific P2Y₁₄R knockout showed a slight improvement in DSS-induced chronic colitis model, whereas it showed almost no improvement in TNBS-induced colitis (**Fig #16.**), which is usually used for the study of Crohn's disease. These results indicated that HDL-16 is more suitable to be the candidate compound for acute UC treatment. Furthermore, the ineffectiveness of HDL-16 in P2Y₁₄R^{ΔIEC} mice indicated that the treatment effect of HDL-16 on DSS-induced acute colitis relies on P2Y₁₄R expressed by IECs.

Fig #16. **a.** Experimental flow chart. **b.** Body weight change and DAI evaluation during the disease process (n = 6). **c.** The length of colons from P2Y₁₄R^{fl/fl} and P2Y₁₄R^{ΔIEC} mice after TNBS treatment. **d.** The H&E staining in colon tissues of DSS-treated mice (Scale bar = 200 μm). The data represent the mean ± SD, and statistical significance was determined by two-way analysis of variance (ANOVA) with Sidak's multiple comparisons test for **b**, and One-way analysis of variance (ANOVA) with Tukey multiple comparison test for **c**. #P < 0.05, ##P < 0.01, and ###P < 0.001 compared to P2Y₁₄R^{fl/fl}+ Veh group; *P < 0.05 compared to P2Y₁₄R^{fl/fl}+ TNBS group.

15. The authors emphasized the increased expression of P2Y₁₄R in the intestinal mucosa of UC patients but not CD patients. An obvious question is “why is such a specificity of P2Y₁₄R associated

with UC but not with CD inflammation?”, however, this important point has been neglected and seems to be a critical issue for the clinical application of the newly discovered P2Y₁₄R antagonist DL-16.

Response: We appreciate your question. We think the different expression of P2Y₁₄R on Ulcerative Colitis (UC) and Crohn’s disease (CD) perhaps induced by the different pathogenesis of these two diseases. UC is characterized by mucosal inflammation initiating in the rectum and extending proximally in the colon in a continuous fashion. At the beginning of the disease, an epithelial barrier defect is observed in UC. The disrupted of intestinal epithelium permit more microbiota to cross the barrier, activating macrophages and antigen-presenting cells (APCs) and resulting in the expression of chemokines that ultimately attract neutrophils. Obviously, innate immune system plays a critical role in pathogenesis of UC (Kobayashi T, et al. *Nat Rev Dis Primers*. 2020). In contrast, Crohn’s disease is believed to associate with genetic susceptibility, about 12% of patients have a family history of Crohn’s disease. Compared with UC, the onset of CD relies on the involvement of adaptive immune system. Furthermore, the pathological mechanisms of UC and CD are different. UC is characterized by mucosal inflammation initiating in the rectum and extending proximally in the colon in a continuous fashion. By contrast, inflammation in CD, the other type of IBD, demonstrates patchy lesions that are potentially scattered anywhere in the gastrointestinal tract. The inflammation in UC is typically limited to the mucosal layer, causing superficial damage of the bowel wall, whereas CD is characterized by transmural inflammation (involving all layers of the bowel wall) that leads to fibrosis, stricture and fistula (Torres J, et al. *Lancet*. 2017). We think these differences may be causing such a specificity of P2Y₁₄R associated with UC but not with CD inflammation.

Besides, as we mentioned above, we constructed TNBS-induced animal model, which is usually used for the study of Crohn's disease, in P2Y₁₄R^{fl/fl} and P2Y₁₄R^{ΔIEC} and monitored for daily weight and disease onset. The results showed that, IEC-specific P2Y₁₄R knockout showed barely improve in TNBS-induced weight loss, disease activity index, colon shortening as well as inflammatory infiltration (**Fig #16**).

Fig #16. a. Experimental flow chart. **b.** Body weight change and DAI evaluation during the disease process (n = 6). **c.** The length of colons from P2Y₁₄R^{fl/fl} and P2Y₁₄R^{ΔIEC} mice after TNBS treatment. **d.** The H&E staining in colon tissues of DSS-treated mice (Scale bar = 200 μm). The data represent the mean ± SD, and statistical significance was determined by two-way analysis of variance (ANOVA) with Sidak's multiple comparisons test for **b**, and One-way analysis of variance (ANOVA) with Tukey multiple comparison test for **c**. #P < 0.05, ##P < 0.01, and ####P < 0.001 compared to P2Y₁₄R^{fl/fl}+ Veh group; *P < 0.05 compared to P2Y₁₄R^{fl/fl}+ TNBS group.

Minor

16. Actual data from the evaluation of intestinal permeability assessment (e.g., FITC assay) should be shown in addition to the pictures of the mice to support the important findings of the authors (Figure 1).

Response: Thank you for your comment. We have added the intestinal permeability assessment of all the in vivo experiment by using FITC-dextran in the revised **Fig 1j**, **Fig 7e** and **supplement Fig 4e**.

Fig.#17 the evaluation of intestinal permeability assessment in DSS-induced experiment colitis. The data represent the mean \pm SD, and statistical significance was determined by One-way ANOVA with Tukey multiple comparison test. * $P < 0.05$, ** $P < 0.01$, and *** $P < 0.001$.

17. The reference(s) should be cited for the sentence 'Previous studies have found increased necroptosis in the colon tissues of patients with CD and UC' in the Introduction (lines 56-57), as it is critically related to the scope of this study.

Response: Thank you for your comment. As your suggestion, we have cited related reference in lines 56-57 and remarked this change in red in the revised manuscript.

18. For introducing one of the authors' rationales of the study, they have indicated that "IECs erosion and colitis induced with dextran sulfate sodium (DSS) are attenuated by knocking out MLKL, implicating IECs necroptosis as a key constituent of experimental colitis. Especially, necroptosis also actively participated in the inflammatory response by promoting cascade reactions through the leakage of cellular contents from damaged plasma membranes." (lines 57-62). It will be more persuasive if the authors can include its related human IBD cases.

Response: Thank you for your comment. As your suggestion, we have included related human IBD cases in lines 57-58 and remarked this change in red in the revised manuscript.

Reviewer #3 (Remarks to the Author):

In the manuscript entitled "Targeting P2Y₁₄R protects against necroptosis of intestinal epithelial cells through PKA/CREB/RIPK1 axis in ulcerative colitis", the authors present a finding that P2Y₁₄R deletion in UC can slow down intestinal damage. Then they investigated the mechanism of action and designed a new P2Y₁₄R antagonist with significant anti-UC effect. The novelty is good.

The reviewer asks to address the following issues:

(1) In Figure 1d, the histology suggests that no good care was taken as to where along the colon the samples were collected, which is very important when interpreting DSS colitis. The samples showing the control mice clearly show proximal colon. In contrast the DSS treated sample shows severe inflammation in the distal colon. As in the DSS-induced colitis model colitis is usually severe in the distal colon while much less activity occurs in the proximal colon, the authors must present samples from the distal colon.

Response: Thank you for your valuable comment. We have checked all the pathologic picture and immunohistochemical picture in our manuscript to make sure all the sample showed was the distal colon. Besides, we have changed the colon sample of Figure 1d, to ensure all sample were collected from the distal colon. However, for better display the expression change of P2Y₁₄R in IECs, we instead immunohistochemical with immunolocalization to characterize the P2Y₁₄R expression in IECs of DSS-treated mice. Thank you again for pointing out this problem for us.

(2) This manuscript shows that P2Y₁₄R may mediate DSS-induced colitis by affecting the death of IECs, and further studies have only clarified that it is not associated with apoptosis. The phenomena of cell membrane rupture and cell swelling also appeared when the cell pyroptosis. In addition, studies have shown that pyroptosis is associated with colitis, and P2Y₁₄R is involved in the regulation of pyroptosis. The authors need to show whether there is a relationship between the effect of P2Y₁₄R and pyroptosis in colitis.

Response: Thank you for this valuable feedback and constructive suggestions. As your suggestion, we detect the expression of GSDMD and cleaved caspase-1 to investigate the effect of IECs P2Y₁₄R on pyroptosis. However, the western blots showed that deficiency of P2Y₁₄R in IECs showed less influence on the expression of GSDMD and cleaved caspase-1, suggesting that IECs P2Y₁₄R did not affect the pyroptosis of IECs in DSS-induced experimental colitis. We have incorporated these

results into the revised **Fig 2b** and **Supplementary figure 4b**. All the method and the information of primary antibody have been included in the method section of the revised manuscript.

Fig #1. a. The expression of Bcl-2, Bax, Caspase1 p20 and GSDMD were analyzed by immunoblotting with corresponding antibodies in the IECs from P2Y₁₄R^{fl/fl} and P2Y₁₄R^{ΔIEC} mice after DSS treatment. **b.** The expression of Bcl-2, Bax, Caspase1 p20 and GSDMD were analyzed by immunoblotting with corresponding antibodies in the IECs from HDL-16-L, HDL-16-H and PPTN administrated mice after 7 days after DSS treatment. The data represent the mean ± SD, and statistical significance was determined by two-way analysis of variance (ANOVA) with Sidak's multiple comparisons test. **P* < 0.05, ***P* < 0.01, and ****P* < 0.001.

(3) There is a problem with Figure 2f, which is inconsistent with the description in the manuscript.

Response: Thank you for pointing out this error for us. We have made the necessary correction in the revised manuscript. We appreciate your kind reminder.

(4) There was no evidence that P2Y₁₄R was knocked down or overexpressed in HT-29 cells.

Response: Thank you for catching this lack of information. We have incorporated additional results into the revised **supplement Fig. 7** showing the reduce and increase levels of P2Y₁₄R in HT-29 cells after siRNA or over-expression plasmid treatment.

Fig.#7 The expression of P2Y₁₄R in HT-29 cells and HCT-116 cells after siRNA or over-expression plasmid treatment. The data represent the mean ± SD, and statistical significance was determined by One-way ANOVA with Tukey multiple comparison test. **P* < 0.05, ***P* < 0.01, and ****P* < 0.001.

(5) In Figure 7f-h, normal HT-29 cells need to be added as controls.

Response: Thank you for your comments. In Figure 7h, we used blank vehicle (0.1% DMSO) treated HT-29 cells as control. As your suggestion, we examined the influence of blank vehicle on necroptosis in HT-29 cells in addition, the results showed that treated HT-29 cells with blank vehicle did not affect the rate of PI positive cells.

Fig #2. PI/Hoechst staining were used to analyzed PI positive cells in normal HT-29 cells and blank vehicle treated HT-29 cells. The data represent the mean ± SD, and statistical significance was determined by Unpaired t test. **P* < 0.05, ***P* < 0.01, and ****P* < 0.001.

(6) This manuscript suggests that P2Y₁₄R prevents intestinal epithelial cell necrosis through the PKA/CREB/RIPK1 axis, but this has not been fully elucidated in experiments.

a) When P2Y₁₄R endogenous ligand UDP-glucose acts on HT-29 cells, do the contents of PKA, CREB and RIPK1 and their phosphorylation levels change?

Response: Thank you for your valuable comment. As suggestion, we have supplemented the results of phosphorylation level of CREB, PKA and mRNA level of *Ripk1* in UDPG treated HT-29 cells to further confirm the role of PKA/CREB/RIPK1 axis in P2Y₁₄R regulating TSZ-induced necroptosis. We have also incorporated these results into the **Fig. 5k-p** of the revised manuscript.

b) When treating DSS-induced colitis with P2Y₁₄R antagonist HDL-16, do the contents of PKA, CREB and RIPK1 and their phosphorylation levels change in colon?

Response: Thank you for your valuable comment. As suggestion, we have added the results of phosphorylation level of CREB, PKA and mRNA level of *Ripk1* in HDL-16 administrated DSS-induced colitis to further confirm the PKA/CREB/RIPK1 axis in P2Y₁₄R regulating DSS-induced colitis. We have also incorporated these results into the **Fig.5 k-p**, **Fig. 7i-k** of the revised manuscript.

Fig.#3 a. Phosphorylation of PKA and CREB as well as their protein levels were analyzed by immunoblotting with corresponding antibodies in HT-29 cells (n = 5). **b.** mRNA level of *Ripk1* in HT-29 cells treated with Veh, TSZ, UDPG, UDPG+TSZ, UDPG+TSZ+Nec-1, UDPG+TSZ+GSK'872. **c.** ChIP with anti-CREB of the regions containing the CREB binding sites on the *Ripk1* gene promoter in HT-29 cells treated with Veh, TSZ, UDPG, UDPG+TSZ, UDPG+TSZ+Nec-1, UDPG+TSZ+GSK'872. **d.** Phosphorylation of PKA and CREB as well as their protein levels were analyzed by immunoblotting with corresponding antibodies in HCT-116 cells. **e.** mRNA

level of *Ripk1* in HCT-116 cells treated with Veh, TSZ, UDPG, UDPG+TSZ, UDPG+ TSZ+Nec-1, UDPG+TSZ+GSK'872. **f.** ChIP with anti-CREB of the regions containing the CREB binding sites on the *Ripk1* gene promoter in HT-29 cells treated with Veh, TSZ, UDPG, UDPG+TSZ, UDPG+ TSZ+Nec-1, UDPG+TSZ+GSK'872. **g.** Phosphorylation of PKA and CREB as well as their protein levels were analyzed by immunoblotting with corresponding antibodies in HDL-16-L, HDL-H or PPTN administrated experimental colitis mice intestinal epithelial cell. **h.** mRNA level of *Ripk1* were analyzed by immunoblotting with corresponding antibodies in HDL-16-L, HDL-H or PPTN administrated experimental colitis mice intestinal epithelial cell. **i.** ChIP with anti-CREB of the regions containing the CREB binding sites on the *Ripk1* gene promoter in intestinal epithelial cell of HDL-16-L, HDL-H or PPTN administrated experimental colitis mice. The data represent the mean \pm SD, and statistical significance was determined by two-way analysis of variance (ANOVA) with Sidak's multiple comparisons test for **a, b** and **g**, and One-way analysis of variance (ANOVA) with Tukey multiple comparison test for **b-c, e-f, h-i**. * $P < 0.05$, ** $P < 0.01$, and *** $P < 0.001$.

(7) The P2Y₁₄R antagonist HDL-16 in this manuscript has excellent activity, but there are too few data to characterize the activity, only two methods (test cAMP production and CETSA) are used. GPCR is a kind of complex target, which needs more support of activity test data. In addition, P2Y₁₄R is a membrane protein, which does not seem to be suitable for direct testing by CETSA method. Whether special treatment methods have been adopted? The paper does not describe any methods for testing HDL-16 activity in vitro.

Response: Thank you for this valuable feedback and constructive comments. This cellular thermal shift assay (CETSA) is based on the biophysical principle of ligand-induced thermal stabilization of target proteins. This approach, termed thermal proteome profiling (TPP), was used to identify small molecule drug binding for target protein and to determine target occupancy at different drug concentrations. This method has been used in extensive research for testing the binding between protein and small molecule including some GPCR (Reinhard, et al. *Nat. Methods*, 2015; Ashok, et al. *Protein Eng.* 2015). Besides, the limitation of this method in ligand-binding receptors is caused by the use of detergent during the extraction of cells. However, it has been proved that extracted cells with 0.4% NP-40 do not affect the determination of accurate protein-drug affinity. Thus, we performed our CETSA experiments with this extraction method. We would also emphasize the point of the revised method (Bantscheff, M. et al. *Nat. Biotechnol.* 2011).

Anyway, we agree that it is not the best method to detect the interaction between GPCR and drugs. However, the protein structure of P2Y₁₄R has not been analyzed yet, so it is hard to performed

SPR, ITC or formation of protein compound eutectic. Nevertheless, we designed a HDL-16-FITC fluorescent conjugate probe suitable for imaging via confocal microscopy to confirm the interaction between **HDL-16** and P2Y₁₄R. Confocal-microscope images of HDL-16-FITC with HEK293 cells expressing P2Y₁₄R (P2Y₁₄-HEK293) showed localized membrane fluorescence (**Fig.#4a**). Besides, we design an **HDL-16**-biotin conjugate probe, which could use to pull down the protein interaction with **HDL-16**. The cell lysates of P2Y₁₄-HEK293 were incubated with 0.5 mM **HDL-16**-biotin probe, then the probe-protein complexes were collected with streptavidin-coupled Dynabeads. The P2Y₁₄R protein in probe-protein complexes was detected by western blot (**Fig.#4b**). The results showed that much more P2Y₁₄R protein was pulled down with HDL-16 than the control group. We have incorporated these results into the **Fig. 6f-g** of the revised manuscript. All the method including the structure and synthetic route of fluorescent conjugate probe and biotin conjugate probe have been included in supplement Scheme. 2 and 3. of the revised manuscript.

Fig.#4 a. HDL-16-FITC fluorescent conjugate probe specifically interacted with P2Y₁₄R in P2Y₁₄-HEK293 cells membrane, which was evaluated by confocal microscopy. **b.** HDL-16-biotin conjugate probe specifically interacted with P2Y₁₄R in P2Y₁₄-HEK293 cells, which was evaluated by a pull-down assay.

(8) The proposal is correct and the results are well-founded. There are some fragments of the Discussion that seem more like a summary. I suggest that the Discussion be rewritten to better reflect the quality of the work. The statement and description of future works are needed.

Response: Thank you for your advice. We have revised the discussion section and added a description of the limitations of our present study, as well as future research plans to better reflect the quality of the work. All the changes have been marked in red in the revised manuscript.

REVIEWER COMMENTS

Reviewer #1 (Remarks to the Author):

The manuscript by Liu et al. provides evidence for previously unknown function of the purinoceptor P2Y₁₄R in promoting necroptosis of intestinal epithelial cells and intestinal inflammation. They furthermore delineate the molecular mechanisms of this function and introduce a novel P2Y₁₄R antagonist HDL-16 capable of ameliorating colitis. The revised manuscript is now greatly improved with substantial additional data that support the author's conclusions. The overwhelming majority of my concerns have been adequately addressed and I have only few comments based on my previous report:

I had previously raised the concern that all experiments were performed in the HT29 cell line only. The authors now included some data performed in primary epithelial organoids (revised Fig 2k, Fig 4l, Fig 5j and Fig 7m) which they claim recapitulates their findings in cell lines. However, they show only some representative pictures of PI stained organoids, which this reviewer finds not very convincing. A quantification of organoid cell death is missing in these analyses.

The reviewer appreciates the addition of TUNEL and caspase-3 stainings of tissues to support the claim that necroptosis inhibition in vitro and colitis inhibition in vivo are mechanistically connected. However, the added pictures are difficult to interpret and therefore this connection is still a bit of a weak point.

Reviewer #3 (Remarks to the Author):

The manuscript by Liu and colleagues has been extensively revised with numerous improvements. My previous comments and concerns have all been addressed. One comment remains regarding data quality: the authors should also add the effect results of HDL-16-FITC in HEK293 cells to Figures 6f.

Reviewer #4 (Remarks to the Author):

In the present study, Liu et al., investigated the role of P2Y₁₄R in the pathophysiology of ulcerative colitis (UC), with particular regard for its ability to modulate intestinal epithelial cell necroptosis, hyperactivated in the presence of colitis. In addition, the authors examined the effects of a new

small-molecule P2Y 14 R antagonist, HDL-16, in counteracting necroptosis and, consequently in reducing intestinal injury associated with UC. In the paper, the authors demonstrated that P2Y 14 R activation is involved in the pathogenesis of UC by regulating necroptosis of intestinal epithelial cells via PKA/CREB/RIPK1 axis. In addition, P2Y 14 R-targeted intervention by a novel antagonist HDL6 exhibited notable therapeutic effects in counteracting DSS-induced colitis in mice, thus representing a promising therapeutic target for inflammatory bowel disease.

The manuscript is interesting, easy to read and presents points of extremely novelty. The execution of the experiments is accurate.

In my opinion, no further changes are needed

Reviewer #1 (Remarks to the Author):

The manuscript by Liu et al. provides evidence for previously unknown function of the purinoceptor P2Y₁₄R in promoting necroptosis of intestinal epithelial cells and intestinal inflammation. They furthermore delineate the molecular mechanisms of this function and introduce a novel P2Y₁₄R antagonist HDL-16 capable of ameliorating colitis. The revised manuscript is now greatly improved with substantial additional data that support the author's conclusions. The overwhelming majority of my concerns have been adequately addressed and I have only few comments based on my previous report:

I had previously raised the concern that all experiments were performed in the HT29 cell line only. The authors now included some data performed in primary epithelial organoids (revised Fig 2k, Fig 4l, Fig 5j and Fig 7m) which they claim recapitulates their findings in cell lines. However, they show only some representative pictures of PI stained organoids, which this reviewer finds not very convincing. A quantification of organoid cell death is missing in these analyses.

The reviewer appreciates the addition of TUNEL and caspase-3 stainings of tissues to support the claim that necroptosis inhibition in vitro and colitis inhibition in vivo are mechanistically connected. However, the added pictures are difficult to interpret and therefore this connection is still a bit of a weak point.

Dear reviewer,

Thank you very much for your comments and professional advice. These opinions help to improve the academic rigor of our manuscript. Based on your suggestion, we performed PI intensity analysis of organoid PI staining to show the changes of necroptosis in a better way. Besides, we repeated the TUNEL and cleaved caspase-3 staining in colon tissues to replace fig 2c and supplement fig 4a with clearer images. We also supplemented the western blot analysis of the cleaved caspase-3 (caspase-3 p17) to further interpret the relationship between P2Y₁₄R and apoptosis. The detailed responses are provided below.

Specific comments:

1. The authors now included some data performed in primary epithelial organoids (revised Fig 2k, Fig 4l, Fig 5j and Fig 7m) which they claim recapitulates their findings in cell lines. However, they

show only some representative pictures of PI stained organoids, which this reviewer finds not very convincing. A quantification of organoid cell death is missing in these analyses.

Response: We appreciate your valid suggestion and have performed quantitative analysis of organoid PI staining in Fig 2k, Fig 4l, Fig 5j, Fig 7m and Supplementary fig 3e of the revised manuscript. According to the results of the quantification of PI staining, P2Y₁₄R deficient significantly inhibited TSZ-induced necroptosis in P2Y₁₄R^{ΔIEC} organoids while inhibition of cAMP/PKA/CREB pathway with SQ22536, H-89 or 666-15 disrupted the protective effect of P2Y₁₄R deficiency in TSZ mediated necroptosis (Fig #1a, b). Meanwhile, UDPG treatment could increase TSZ induced necroptosis in organoids from P2Y₁₄R^{fl/fl} mice, while administration with Nec-1s and GSK'872 remarkably attenuated the UDP-glucose derived damage on TSZ-stimulated necroptosis (Fig #1c). Additionally, administration with P2Y₁₄R inhibitor HDL-16 also protected P2Y₁₄R^{fl/fl} organoids from TSZ induced necroptosis (Fig #1d). We have incorporated these results in Fig. 2k, Fig. 4l, Fig. 5j, Fig. 7l, and supplementary Fig. 3d of the revised manuscript. We also included all the raw data and original images used for PI intensity analysis in the “Source data” file. We sincerely thank you for your helpful feedback, which greatly contributes to improving the rigor of our research.

Fig#1 a The PI staining and quantification of intestinal organoids from P2Y₁₄R^{fl/fl} and P2Y₁₄R^{ΔIEC} mice treated as indicated with DMSO (veh) or TNF-α adding Smac mimetic and z-VAD (TSZ) for 12 h. **b** The PI staining and quantification of intestinal organoids from P2Y₁₄R^{ΔIEC} mice treated as indicated with DMSO (veh), TSZ, TSZ adding SQ22536, TSZ adding H-89 or TSZ adding 666-15. **c** The PI staining and quantification of intestinal organoids from P2Y₁₄R^{fl/fl} mice treated as indicated. **d** The PI staining and quantification of intestinal organoids from P2Y₁₄R^{fl/fl} mice treated as indicated. **e** The PI staining and quantification of intestinal organoids from P2Y₁₄R^{fl/fl} and P2Y₁₄R^{ΔIEC} mice treated as indicated with Veh, TSZ and TSZ + UDPG. The data represent the mean ± SD, and statistical significance was determined by One-way analysis of variance (ANOVA) with Tukey multiple comparison test. **P* < 0.05, ***P* < 0.01, and ****P* < 0.001.

2. The reviewer appreciates the addition of TUNEL and caspase-3 stainings of tissues to support the claim that necroptosis inhibition in vitro and colitis inhibition in vivo are mechanistically connected. However, the added pictures are difficult to interpret and therefore this connection is still a bit of a weak point.

Response: We appreciate your constructive suggestions. According to your suggestion, we repeated the staining of TUNEL and cleaved caspase-3 in fig 2c and supplement fig 4a. Besides, we added the western blot analysis of cleaved caspase-3 (caspase-3 p17) to investigate the effect of IEC-specific P2Y₁₄R knockout and HDL-16 treatment on apoptosis in DSS-induced experiment colitis. As the results showed, IEC-specific P2Y₁₄R knockout as well as HDL-16 treatment had minimal impact on apoptosis (**Fig #2**). We have also incorporated these results in **Fig. 2b, c** and **supplementary Fig. 4a, b** of the revised manuscript.

Fig#2 a The TUNEL staining of colon tissues from $P2Y_{14}R^{fl/fl}$ and $P2Y_{14}R^{\Delta IEC}$ mice after DSS treatment. The immunofluorescent images of colon tissues stained with cleave caspase-3 (Scale bar = 50 μ m). **b** The expression of Bcl-2, Bax, GSDMD-NT, Caspase-1 p20, Caspase-3 p17, MLKL and phosphorylation of MLKL were analyzed by immunoblotting in the IECs from $P2Y_{14}R^{fl/fl}$ and $P2Y_{14}R^{\Delta IEC}$ mice after DSS treatment. **c** The TUNEL staining of colon tissues from mice with HDL-16 or PPTN administration. The immunofluorescent images of colon tissues stained with cleave caspase-3 (Scale bar = 50 μ m). **d** The expression of Bax, Bcl-2, GSDMD, caspase-1 p20 as well as caspase-3 p17 were analyzed by immunoblotting with corresponding antibodies in the IECs from DSS treated mice with HDL-16 or PPTN administration. The data represent the mean \pm SD, and statistical significance was determined by two-way ANOVA with Sidak's multiple comparisons test. * $P < 0.05$, ** $P < 0.01$, and *** $P < 0.001$.

We would like to take this opportunity to thank you for all your time involved and this great opportunity for us to improve the manuscript. We hope you will find this revised version satisfactory. If you have any questions, please do not hesitate to contact us.

Thank you and best regards.

-----End of Reply to Reviewer #1-----

Reviewer #3 (Remarks to the Author):

The manuscript by Liu and colleagues has been extensively revised with numerous improvements. My previous comments and concerns have all been addressed. One comment remains regarding data quality: the authors should also add the effect results of HDL-16-FITC in HEK293 cells to Figures 6f.

Dear reviewer,

Thank you very much for your comments and professional advice. These opinions would help to improve the academic rigor of our manuscript. Based on your suggestion, we performed living cell image of **HDL-16-FITC** fluorescent conjugate probe in HEK293 cells. The results showed that, in contrast with the location in P2Y₁₄-HEK293 cells, the location of **HDL-16-FITC** in HEK293 cells was mainly in intercellular space (**Fig. #1**), indicating that the location of **HDL-16-FITC** in P2Y₁₄-HEK293 cells rely on the expression of P2Y₁₄R in membrane. We have also incorporated these results into **Fig. 6f** of the revised manuscript. Thank you very much for your attention and time again.

Fig.#1. Confocal imaging of **HDL-16-FITC** fluorescent conjugate probe in P2Y₁₄-HEK293 cells or HEK293 cells. We would like to take this opportunity to thank you for all your time involved and this great opportunity for us to improve the manuscript. We hope you will find this revised version satisfactory. If you have any questions, please do not hesitate to contact us.

Thank you and best regards.

-----End of Reply to Reviewer #3-----

REVIEWERS' COMMENTS

Reviewer #1 (Remarks to the Author):

My remaining comments were adequately addressed and the paper is now greatly improved

Reviewer #3 (Remarks to the Author):

This manuscript has been revised properly. I have no any more comments now.